# Transport meets Variational Inference: Controlled Monte Carlo Diffusions

**Francisco Vargas**[*], **Shreyas Padhy**[*]
University of Cambridge
Cambridge, UK
{fav25,sp2058}@cam.ac.uk

**Denis Blessing**
KIT
Karlsruhe, Germany
jl8142@kit.edu

**Nikolas Nüsken**[*]
Kings College London
London, UK
nik.nuesken@gmx.de

## Abstract

Connecting optimal transport and variational inference, we present a principled and systematic framework for sampling and generative modelling centred around divergences on path space. Our work culminates in the development of *Controlled Monte Carlo Diffusions* for sampling and inference, a score-based annealing technique that crucially adapts both forward and backward dynamics in a diffusion model. On the way, we clarify the relationship between the EM-algorithm and iterative proportional fitting (IPF) for Schrödinger bridges, providing a conceptual link between fields. Finally, we show that CMCD has a strong foundation in the Jarzinsky and Crooks identities from statistical physics, and that it convincingly outperforms competing approaches across a wide array of experiments.

## 1 Introduction

Optimal transport (Villani et al., 2009) and variational inference (Blei et al., 2017) have for a long time been separate fields of research. In recent years, many fruitful connections have been established (Liu et al., 2019), in particular based on dynamical formulations (Tzen & Raginsky, 2019a), and in conjunction with time reversals (Huang et al., 2021a; Song et al., 2021). The goal of this paper is twofold: In the first part, we enhance those relationships based on forward and reverse time diffusions, and associated Girsanov transformations, arriving at a unifying framework for generative modeling and sampling. In the second part, we build on this and develop a novel score-based scheme for sampling from unnormalised densities. To set the stage, we recall a classical approach (Kingma & Welling, 2014; Rezende & Mohamed, 2015) towards generating samples from a target distribution $\mu(\boldsymbol{x})$, which is the goal both in generative modelling and sampling:

**Generative processes, encoders and decoders.** We consider methodologies which can be implemented via the following generative process,

$$\boldsymbol{z} \sim \nu(\boldsymbol{z}), \qquad \boldsymbol{x}|\boldsymbol{z} \sim p^\theta(\boldsymbol{x}|\boldsymbol{z}), \tag{1}$$

transforming a sample $\boldsymbol{z} \sim \nu(\boldsymbol{z})$ into a sample $\boldsymbol{x} \sim \int p^\theta(\boldsymbol{x}|\boldsymbol{z})\nu(\mathrm{d}\boldsymbol{z})$. Traditionally, $\nu(\boldsymbol{z})$ is a simple auxiliary distribution, and the family of transitions $p^\theta(\boldsymbol{x}|\boldsymbol{z})$ is parameterised flexibly and in such a way that sampling according to (1) is tractable. Then we can frame the tasks of generative modelling and sampling as finding transition densities such that the marginal in $\boldsymbol{x}$ matches the target distribution,

$$\mu(\boldsymbol{x}) = \int p^\theta(\boldsymbol{x}|\boldsymbol{z})\nu(\mathrm{d}\boldsymbol{z}). \tag{2}$$

To learn such a transition, it is helpful to introduce a reversed process

$$\boldsymbol{x} \sim \mu(\boldsymbol{x}), \qquad \boldsymbol{z}|\boldsymbol{x} \sim q^\phi(\boldsymbol{z}|\boldsymbol{x}), \tag{3}$$

relying on an appropriately parameterised backward transition $q^\phi(\boldsymbol{z}|\boldsymbol{x})$. We will say that (1) and (3) are *reversals of each other* in the case when their joint distributions coincide, that is, when

$$q^\phi(\boldsymbol{z}|\boldsymbol{x})\mu(\boldsymbol{x}) = p^\theta(\boldsymbol{x}|\boldsymbol{z})\nu(\boldsymbol{z}). \tag{4}$$

To appreciate the significance of (3), notice that if (4) holds, then (2) is implied by integrating both sides with respect to $\boldsymbol{z}$. Building on this observation, it is natural to define the loss function

$$\mathcal{L}_D(\phi, \theta) := D\left(q^\phi(\boldsymbol{z}|\boldsymbol{x})\mu(\boldsymbol{x}) \,\big|\big|\, p^\theta(\boldsymbol{x}|\boldsymbol{z})\nu(\boldsymbol{z})\right), \tag{5}$$

---

[*]Equal contribution.

where $D$ is a divergence[1] between distributions yet to be specified. Along the lines of Bengio et al. (2021); Sohl-Dickstein et al. (2015); Wu et al. (2020); Liu et al. (b), we have now laid the foundations for algorithmic approaches that aim at sampling from $\mu(\boldsymbol{x})$ by minimising $\mathcal{L}_D(\phi, \theta)$:

**Framework 1.** Let $D$ be an arbitrary divergence, and assume that $\mathcal{L}_D(\phi, \theta) = 0$. Then we have

$$\mu(\boldsymbol{x}) = \int p^\theta(\boldsymbol{x}|\boldsymbol{z})\nu(\mathrm{d}\boldsymbol{z}) \quad \text{and} \quad \nu(\boldsymbol{z}) = \int q^\phi(\boldsymbol{z}|\boldsymbol{x})\mu(\mathrm{d}\boldsymbol{x}), \tag{6}$$

that is, $\nu(\boldsymbol{z})$ is transformed into $\mu(\boldsymbol{x})$ by $p^\theta(\boldsymbol{x}|\boldsymbol{z})$, and $\mu(\boldsymbol{x})$ is transformed into $\nu(\boldsymbol{z})$ by $q^\phi(\boldsymbol{z}|\boldsymbol{x})$.

**The sampling problem.** Let $\nu$ denote a probability density function on $\mathbb{R}^d$ of the form $\nu(\boldsymbol{z}) = \frac{\hat{\nu}(\boldsymbol{z})}{Z}$, $Z = \int_{\mathbb{R}^d} \hat{\nu}(\boldsymbol{z})\mathrm{d}\boldsymbol{z}$, where $\hat{\nu} : \mathbb{R}^d \to \mathbb{R}^+$ can be differentiated and evaluated pointwise but the normalizing constant $Z$ is intractable. We are interested in both estimating $Z$ and obtaining approximate samples from $\nu$ given we can sample from a more tractable density $\mu$. Framework 1 provides us with an objective to tackle the sampling problem as once $\mathcal{L}_D(\phi, \theta) = 0$, we can generate samples from $\nu(\boldsymbol{z})$ via the variational distribution $q^\phi(\boldsymbol{z}|\boldsymbol{x})$. Through variational inference and optimal transport, we discuss relationships to classical methods as well as shortcomings:

**KL-divergence, ELBO and variational inference.** Choosing $D = D_{\mathrm{KL}}$ in (5), variational inference (VI) and latent variable model based approaches (Dempster et al., 1977; Blei et al., 2017; Kingma & Welling, 2014) can elegantly be placed within Framework 1. Indeed, direct computation (see Appendix B) shows that $\mathcal{L}_{D_{\mathrm{KL}}}(\phi, \theta) = -\mathbb{E}_{\boldsymbol{x}\sim\mu(\boldsymbol{x})}[\mathrm{ELBO}_x(\phi, \theta)] + \mathbb{E}_{\boldsymbol{x}\sim\mu(\boldsymbol{x})}[\ln \mu(x)]$, so that minimising $\mathcal{L}_{D_{\mathrm{KL}}}(\phi, \theta)$ is equivalent to maximising the expected evidence lower bound (ELBO), also known as the negative free energy (Blei et al., 2017). This derivation is alternative to the standard approach via maximum likelihood and convex duality (or Jensen's inequality) (Kingma et al., 2021, Section 2.2), and directly accomodates various modifications by replacing the $D_{\mathrm{KL}}$-divergence (see Appendix B).

**Couplings, (optimal) transport and nonuniqueness.** Assuming (4) holds, it is natural to define the joint distribution $\pi(\boldsymbol{x}, \boldsymbol{z}) := q^\phi(\boldsymbol{z}|\boldsymbol{x})\mu(\boldsymbol{x}) = p^\theta(\boldsymbol{x}|\boldsymbol{z})\nu(\boldsymbol{z})$, which is a coupling between $\mu(\boldsymbol{x})$ and $\nu(\boldsymbol{z})$. Viewed from this angle, the set of minimisers of $\mathcal{L}(\phi, \theta)$ stands in one-to-one correspondence with the set of couplings between $\mu(\boldsymbol{x})$ and $\nu(\boldsymbol{z})$, provided that the parameterisations are chosen flexibly enough. Under the latter assumption, the objective in (5) admits an infinite number of minimisers, rendering algorithmic approaches solely based on Framework 1 potentially unstable and their output hard to interpret. In the language of *optimal transport* (Villani, 2003), minimising $\mathcal{L}(\phi, \theta)$ enforces the marginal (*'transport'*) constraints in (6) without a selection principle based on an appropriate cost function (*'optimal'*).

Methods such as VAEs (Kingma & Welling, 2014) parameterise $p^\theta(\boldsymbol{x}|\boldsymbol{z})$ and $q^\phi(\boldsymbol{z}|\boldsymbol{x})$ with a restricted family of distributions (such as Gaussians), thus restricting the set of couplings. Expectation maximisation (EM) minimises $\mathcal{L}(\phi, \theta)$ in a component-wise fashion, resolving nonquniqueness in a procedural manner (see Section 3.1). Common diffusion models fix either $p^\theta(\boldsymbol{x}|\boldsymbol{z})$ or $q^\phi(\boldsymbol{z}|\boldsymbol{x})$, and thus select a coupling (Section 2.2). In this paper, we argue that the full potential of diffusion models can be unleashed by training the forward and backward processes at the same time, but appropriate modifications that resolve the nonuniqueness inherent in Framework 1 need to be imposed. To develop principled approaches towards this, we proceed as follows:

**Outline and contributions.** In Section 2 we recall hierarchical VAEs (Rezende et al., 2014) and, following Tzen & Raginsky (2019a), proceed to the infinite-depth limit described by the SDEs in (12). Readers more familiar with VI and discrete time might want to take the development in Section 2.1 as an explanation of (12); readers with background in stochastic analysis might take Framework 1' as their starting point. In Proposition 2.2 we provide a generalised form of the Girsanov theorem for forward-reverse time SDEs, crucially incorporating the choice of a reference process that allows us to reason about sampling and generation in a systematic and principled way. We demonstrate that a range of widely used approaches, such as score-based diffusions and path integral samplers, among others, are special cases of our unifying framework (Section 2.2). Similarly in Section 3.1 we unify optimal transport (OT) and VI under our framework by establishing a correspondence between expectation-maximisation (EM) and iterative proportional fitting (IPF). Going further, we show that this framework allows us to derive new methods:

In Section 3.2, we derive a novel score-based annealed flow technique, the Controlled Monte Carlo Diffusion (CMCD) sampler, and show that it may be viewed as an infinitesimal analogue of the

---

[1]As usual, divergences are characterised by the requirement that $D(\alpha||\beta) \geq 0$, with equality iff $\alpha = \beta$.

method from Section 3.1. Finally, we connect CMCD to the foundational identities by Crooks and Jarzynki in statistical physics, and show that it empirically outperforms a range of state-of-the-art inference methods in sampling and estimating normalizing constants (Section 4).

## 2 FROM HIERARCHICAL VAES TO FORWARD-REVERSE TIME DIFFUSIONS

### 2.1 HIERARCHICAL VAES (REZENDE ET AL., 2014)

A particularly flexible choice of implicitly parameterising $p^\theta(\boldsymbol{x}|\boldsymbol{z})$ and $q^\phi(\boldsymbol{z}|\boldsymbol{x})$ can be achieved via a hierarchical model with intermediate latents: We identify $\boldsymbol{x} =: \boldsymbol{y}_0$ and $\boldsymbol{z} =: \boldsymbol{y}_L$ with the 'endpoints' of the layered augmentation $(\boldsymbol{y}_0, \boldsymbol{y}_1, \ldots, \boldsymbol{y}_{L-1}, \boldsymbol{y}_L) =: \boldsymbol{y}_{0:L}$, and define

$$q^\phi(\boldsymbol{y}_L, \ldots, \boldsymbol{y}_1|\boldsymbol{y}_0) := \prod_{l=1}^L q^{\phi_{l-1}}(\boldsymbol{y}_l|\boldsymbol{y}_{l-1}), \qquad p^\theta(\boldsymbol{y}_0, \ldots, \boldsymbol{y}_{L-1}|\boldsymbol{y}_L) := \prod_{l=1}^L p^{\theta_l}(\boldsymbol{y}_{l-1}|\boldsymbol{y}_l), \quad (7)$$

so that $q^\phi(\boldsymbol{z}|\boldsymbol{x})$ and $p^\theta(\boldsymbol{x}|\boldsymbol{z})$ can be obtained from (7) by marginalising over the auxiliary variables $\boldsymbol{y}_1, \ldots, \boldsymbol{y}_{L-1}$. Here, $\phi = (\phi_0, \ldots, \phi_{L-1})$ and $\theta = (\theta_1, \ldots, \theta_L)$ refer to sets of parameters to be specified in more detail below. Further introducing notation, we write $q^{\mu,\phi}(\boldsymbol{y}_{0:L}) := q^\phi(\boldsymbol{y}_{1:L}|\boldsymbol{y}_0)\mu(\boldsymbol{y}_0)$ as well as $p^{\nu,\theta}(\boldsymbol{y}_{0:L}) := p^\theta(\boldsymbol{y}_{0:L-1}|\boldsymbol{y}_L)\nu(\boldsymbol{y}_L)$ and think of those implied joint distributions as emanating from $\mu(\boldsymbol{x}) = \mu(\boldsymbol{y}_0)$ and $\nu(\boldsymbol{z}) = \nu(\boldsymbol{y}_L)$, respectively, moving 'forwards' or 'backwards' according to the specific choices for $\phi$ and $\theta$. In the regime when $L$ is large, the models in (7) are very expressive, even if the intermediate transition kernels are parameterised in a simple manner. We hence proceed by assuming Gaussian distributions,

$$q^{\phi_{l-1}}(\boldsymbol{y}_l|\boldsymbol{y}_{l-1}) = \mathcal{N}(\boldsymbol{y}_l|\boldsymbol{y}_{l-1} + \delta a_{l-1}^\phi(\boldsymbol{y}_{l-1}), \delta\sigma^2 I), \quad p^{\theta_l}(\boldsymbol{y}_{l-1}|\boldsymbol{y}_l) = \mathcal{N}(\boldsymbol{y}_{l-1}|\boldsymbol{y}_l + \delta b_l^\theta(\boldsymbol{y}_l), \delta\sigma^2 I), \quad (8)$$

where $\sigma > 0$ controls the standard deviation, and $\delta > 0$ is a small parameter, anticipating the limits $L \to \infty, \delta \to 0$ to be taken in Section 2.2 below. The vector fields $a_l^\phi(\boldsymbol{y}_l)$ and $b_l^\theta(\boldsymbol{y}_l)$ introduced in (8) should be thought of as parameterised by $\phi$ and $\theta$, but we will henceforth suppress this for brevity.

The models (7)-(8) could equivalently be defined via the Markov chains

$$\boldsymbol{y}_{l+1} = \boldsymbol{y}_l + \delta a_l(\boldsymbol{y}_l) + \sqrt{\delta}\sigma\xi_l, \qquad \boldsymbol{y}_0 \sim \mu \implies \boldsymbol{y}_{0:L} \sim q^{\mu,\phi}(\boldsymbol{y}_{0:L}), \quad (9a)$$

$$\boldsymbol{y}_{l-1} = \boldsymbol{y}_l + \delta b_l(\boldsymbol{y}_l) + \sqrt{\delta}\sigma\xi_l, \qquad \boldsymbol{y}_L \sim \nu \implies \boldsymbol{y}_{0:L} \sim p^{\nu,\theta}(\boldsymbol{y}_{0:L}), \quad (9b)$$

where $(\xi_l)_{l=1}^L$ is an iid sequence of standard Gaussian random variables. As indicated, the forward process in (9a) may serve to define the distribution $q^{\mu,\phi}(\boldsymbol{y}_{0:L})$, whilst the backward process in (9b) induces $p^{\nu,\theta}(\boldsymbol{y}_{0:L})$. Note that the transition densities $p^\theta(\boldsymbol{x}|\boldsymbol{z})$ and $q^\phi(\boldsymbol{z}|\boldsymbol{x})$ obtained as the marginals of (7) will in general not be available in closed form. However, generalising slightly from Framework 1, we may set out to minimise the extended loss

$$\mathcal{L}_D^{\text{ext}}(\phi, \theta) = D(q^{\mu,\phi}(\boldsymbol{y}_{0:L})||p^{\nu,\theta}(\boldsymbol{y}_{0:L})), \quad (10)$$

where $D$ refers to a divergence on the 'discrete path space' $\{\boldsymbol{y}_{0:L}\}$. Clearly, $\mathcal{L}_D^{\text{ext}}(\phi, \theta) = 0$ still implies (6), but is no longer equivalent. More specifically, in the case when $D = D_{\text{KL}}$, the data processing inequality yields

$$D_{\text{KL}}(q^{\mu,\phi}(\boldsymbol{y}_{0:L})||p^{\nu,\theta}(\boldsymbol{y}_{0:L})) \geq D_{\text{KL}}\left(q^\phi(\boldsymbol{z}|\boldsymbol{x})\mu(\boldsymbol{x})||p^\theta(\boldsymbol{x}|\boldsymbol{z})\nu(\boldsymbol{z})\right), \quad (11)$$

so that $\mathcal{L}_{D_{\text{KL}}}^{\text{ext}}(\phi, \theta)$ provides an upper bound for $\mathcal{L}_{D_{\text{KL}}}(\phi, \theta)$ as defined in (5).

### 2.2 DIFFUSION MODELS – HIERARCHICAL VAES IN THE INFINITE DEPTH LIMIT

Here we take inspiration from Section 2.1 and Tzen & Raginsky (2019a); Li et al. (2020); Huang et al. (2021a) to investigate the $L \to \infty$ limit, using stochastic differential equations (SDEs). To this end, we think of $l = 0, \ldots, L$ as discrete instances in a fixed time interval $[0, T]$, equidistant with time step $\delta$, that is, we set $\delta = TL^{-1}$. The discrete paths $\boldsymbol{y}_{0:L}$ give rise to continuous paths $(\boldsymbol{Y}_t)_{0 \leq t \leq T} \in C([0, T]; \mathbb{R}^d)$ by setting $\boldsymbol{Y}_{\delta l} = \boldsymbol{y}_l$ and linearly interpolating $\boldsymbol{Y}_{\delta l}$ and $\boldsymbol{Y}_{\delta(l+1)}$. To complete the set-up, we think of $a^\phi = (a_0^\phi, \ldots, a_{L-1}^\phi)$ and $b^\theta = (b_1^\theta, \ldots, b_L^\theta)$ in (8) as arising from time-dependent vector fields $a, b \in C^\infty([0, T] \times \mathbb{R}^d; \mathbb{R}^d)$ via $a_l^\phi(\boldsymbol{y}_l) = a_{t\delta^{-1}}(\boldsymbol{Y}_{\delta l})$ and $b_l^\theta(\boldsymbol{y}_l) = b_{t\delta^{-1}}(\boldsymbol{Y}_{\delta l})$.

Taking the limit $\delta \to 0$, while keeping $T > 0$ fixed, transforms the Markov chains in (9) into continuous-time dynamics described by the SDEs (Tzen & Raginsky, 2019a)

$$\mathrm{d}\boldsymbol{Y}_t = a_t(\boldsymbol{Y}_t)\,\mathrm{d}t + \sigma\,\overrightarrow{\mathrm{d}}\,\boldsymbol{W}_t, \quad \boldsymbol{Y}_0 \sim \mu \implies (\boldsymbol{Y}_t)_{0 \le t \le T} \sim \mathbb{Q}^{\mu,a} \equiv \overrightarrow{\mathbb{P}}^{\mu,a}, \tag{12a}$$

$$\mathrm{d}\boldsymbol{Y}_t = b_t(\boldsymbol{Y}_t)\,\mathrm{d}t + \sigma\,\overleftarrow{\mathrm{d}}\,\boldsymbol{W}_t, \quad \boldsymbol{Y}_T \sim \nu \implies (\boldsymbol{Y}_t)_{0 \le t \le T} \sim \mathbb{P}^{\nu,b} \equiv \overleftarrow{\mathbb{P}}^{\nu,b}, \tag{12b}$$

where $\overrightarrow{\mathrm{d}}$ and $\overleftarrow{\mathrm{d}}$ denote forward and backward Itô integration (see Appendix A for more details and remarks on the notation), and $(\boldsymbol{W}_t)_{0 \le t \le T}$ is a standard Brownian motion. In complete analogy with (9), the SDEs in (12) induce the distributions $\mathbb{Q}^{\mu,a}$ and $\mathbb{P}^{\nu,b}$ on the path space $C([0,T]; \mathbb{R}^d)$. Relating back to the discussion in the introduction, note that we maintain the relations $\boldsymbol{Y}_0 = \boldsymbol{x}$ and $\boldsymbol{Y}_T = \boldsymbol{z}$, and the transitions are parameterised by the vector fields $a, b$, in the sense that $p^\theta(\boldsymbol{x}|\boldsymbol{z}) = \mathbb{P}_0^{\nu,b^\theta}(\boldsymbol{x}|\boldsymbol{Y}_T = \boldsymbol{z}) = \mathbb{P}_0^{\delta_{\boldsymbol{z}},b^\theta}(\boldsymbol{x})$ and $q^\phi(\boldsymbol{z}|\boldsymbol{x}) = \mathbb{Q}_T^{\mu,a^\phi}(\boldsymbol{z}|\boldsymbol{Y}_0 = \boldsymbol{x}) = \mathbb{Q}_T^{\delta_{\boldsymbol{x}},a^\phi}(\boldsymbol{z})$.

The following well-known result (Anderson, 1982; Nelson, 1967) allows us to relate forward and backward path measures via a local (score-matching) condition for the reversal relation in (4). [2]

**Proposition 2.1** (Nelson's relation). *For $\mu$ and $a$ of sufficient regularity, denote the time-marginals of the corresponding path measure by $\overrightarrow{\mathbb{P}}_t^{\mu,a} =: \rho_t^{\mu,a}$. Then $\overrightarrow{\mathbb{P}}^{\mu,a} = \overleftarrow{\mathbb{P}}^{\nu,b}$ if and only if*

$$\nu = \overrightarrow{\mathbb{P}}_T^{\mu,a} \qquad and \qquad b_t = a_t - \sigma^2 \nabla \ln \rho_t^{\mu,a}, \qquad for \ all \ t \in (0,T]. \tag{13}$$

**Remark 1.** A similarly clean characterisation of equality between forward and backward path measures is not available for the discrete-time setting as presented in (9). In particular, Gaussianity of the intermediate transitions is not preserved under time-reversal.

A recurring theme in this work and related literature is the interplay between the score-matching condition in (13) and the global condition $D(\overrightarrow{\mathbb{P}}^{\mu,a}|\overleftarrow{\mathbb{P}}^{\nu,b}) = 0$, invoking Framework 1. To enable calculations involving the latter, we will rely on the following result:

**Proposition 2.2** (forward-backward Radon-Nikodym derivatives). *Let $\overrightarrow{\mathbb{P}}^{\Gamma_0,\gamma^+} = \overleftarrow{\mathbb{P}}^{\Gamma_T,\gamma^-}$ be a reference path measure (that is, $\Gamma_0$, $\Gamma_T$ and $\gamma^\pm$ define diffusions as in (12) and are related as in Proposition 2.1), absolutely continuous with respect to both $\overrightarrow{\mathbb{P}}^{\mu,a}$ and $\overleftarrow{\mathbb{P}}^{\nu,b}$. Then, $\overrightarrow{\mathbb{P}}^{\mu,a}$-almost surely, the corresponding Radon-Nikodym derivative (RND) can be expressed as follows,*

$$\ln\left(\frac{\mathrm{d}\overrightarrow{\mathbb{P}}^{\mu,a}}{\mathrm{d}\overleftarrow{\mathbb{P}}^{\nu,b}}\right)(\boldsymbol{Y}) = \ln\left(\frac{\mathrm{d}\mu}{\mathrm{d}\Gamma_0}\right)(\boldsymbol{Y}_0) - \ln\left(\frac{\mathrm{d}\nu}{\mathrm{d}\Gamma_T}\right)(\boldsymbol{Y}_T) \tag{14a}$$

$$+ \frac{1}{\sigma^2}\int_0^T \left(a_t - \gamma_t^+\right)(\boldsymbol{Y}_t)\cdot\left(\overrightarrow{\mathrm{d}}\,\boldsymbol{Y}_t - \frac{1}{2}\left(a_t + \gamma_t^+\right)(\boldsymbol{Y}_t)\,\mathrm{d}t\right) \tag{14b}$$

$$- \frac{1}{\sigma^2}\int_0^T \left(b_t - \gamma_t^-\right)(\boldsymbol{Y}_t)\cdot\left(\overleftarrow{\mathrm{d}}\,\boldsymbol{Y}_t - \frac{1}{2}\left(b_t + \gamma_t^-\right)(\boldsymbol{Y}_t)\,\mathrm{d}t\right). \tag{14c}$$

*Proof.* The proof relies on Girsanov's theorem (Üstünel & Zakai, 2013), using the reference to relate the forward and backward processes. For details, see Appendix E. □

**Remark 2** (Role of the reference process). According to Proposition 2.2, the Radon-Nikodym derivative between $\overrightarrow{\mathbb{P}}^{\mu,a}$ and $\overleftarrow{\mathbb{P}}^{\nu,b}$ can be decomposed into boundary terms (14a), as well as forward and backward path integrals (14b) and (14c). Since the left-hand side of (14a) does not depend on the reference $\Gamma_{0,T}, \gamma^\pm$, the expressions in (14) are in principle equivalent for all choices of reference. The freedom in $\Gamma_{0,T}$ and $\gamma^\pm$ allows us to 'reweight' between (14a), (14b) and (14c), or even cancel terms. A canonical choice is the Lebesgue measure for $\Gamma_0$ and $\Gamma_T$, and $\gamma^\pm = 0$, see Appendix C.1.

**Remark 3** (Discretisation and conversion formulae). The distinction between forward and backward integration in (14) is related to the time points at which the integrands $\left(a_t - \gamma_t^+\right)(\boldsymbol{Y}_t)$ and $\left(b_t - \gamma_t^-\right)(\boldsymbol{Y}_t)$ would be evaluated in discrete-time approximations, e.g.,

$$\int_0^T a_t(\boldsymbol{Y}_t)\cdot\overrightarrow{\mathrm{d}}\,\boldsymbol{Y}_t \approx \sum_i a_{t_i}(\boldsymbol{Y}_{t_i})\cdot(\boldsymbol{Y}_{t_{i+1}} - \boldsymbol{Y}_{t_i}), \quad \int_0^T a_t(\boldsymbol{Y}_t)\cdot\overleftarrow{\mathrm{d}}\,\boldsymbol{Y}_t \approx \sum_i a_{t_{i+1}}(\boldsymbol{Y}_{t_{i+1}})\cdot(\boldsymbol{Y}_{t_{i+1}} - \boldsymbol{Y}_{t_i}).$$

Alternatively, forward and backward integrals can be transformed into each other using the conversion

$$\int_0^T a_t(\boldsymbol{Y}_t)\cdot\overrightarrow{\mathrm{d}}\,\boldsymbol{Y}_t = \int_0^T a_t(\boldsymbol{Y}_t)\cdot\overleftarrow{\mathrm{d}}\,\boldsymbol{Y}_t - \sigma^2\int_0^T (\nabla\cdot a_t)(\boldsymbol{Y}_t)\,\mathrm{d}t. \tag{15}$$

We refer to Kunita (2019) and Appendix A for further details. In passing, we note that (15) allows us to eliminate the Hutchinson estimator (Hutchinson, 1989) from a variety of common score-matching objectives, potentially reducing the variance of gradient estimators, see Appendix C.1.

---

[2] The global condition $\overrightarrow{\mathbb{P}}^{\mu,a} = \overleftarrow{\mathbb{P}}^{\nu,b}$ is captured by the local condition (13) due to (12)'s Markovian nature.

Framework 1 can be translated into the setting of (12), noting that (11) continues to hold with appropriate modifications:

**Framework 1′.** For a divergence $D$ on path space, minimise $D(\overrightarrow{\mathbb{P}}^{\mu,a}|\overleftarrow{\mathbb{P}}^{\nu,b})$. If $D(\overrightarrow{\mathbb{P}}^{\mu,a}|\overleftarrow{\mathbb{P}}^{\nu,b}) = 0$, then (12a) transports $\mu$ to $\nu$, and (12b) transports $\nu$ to $\mu$. [3]

At optimality, $D(\overrightarrow{\mathbb{P}}^{\mu,a}|\overleftarrow{\mathbb{P}}^{\nu,b}) = 0$, Proposition 2.1 allows us to obtain the scores associated to the learned diffusion via $\sigma^2 \nabla \ln \rho_t^{\mu,a} = a_t - b_t$. In this way, Framework 1′ is closely connected to (and in some ways extends) score-matching ideas (Song & Ermon, 2019; Song et al., 2021). Indeed, recent approaches towards generative modeling and sampling can be recovered from Framework 1′ by making specific choices for the divergence $D$, the parameterisations for $a$ and $b$, as well as for the reference diffusion $\overrightarrow{\mathbb{P}}^{\Gamma_0,\gamma^+} = \overleftarrow{\mathbb{P}}^{\Gamma_T,\gamma^-}$ in Proposition 2.2:

**Score-based generative modeling:** Letting $\mu$ be the target and fixing the forward drift $a_t$, and, motivated by Proposition 2.1, parameterising the backward drift as $b_t = a_t - s_t$, we recover the SGM objectives in Hyvärinen & Dayan (2005); Song & Ermon (2019); Song et al. (2021) from $D = D_{\mathrm{KL}}$; when $\overrightarrow{\mathbb{P}}^{\mu,a} = \overleftarrow{\mathbb{P}}^{\nu,b}$, the variable drift component $s_t$ will represent the score $\sigma^2 \nabla \ln \rho_t^{\mu,a}$. Modifications can be obtained from the conversion formula (15), see Appendix C.2.

**Score-based sampling – ergodic drift:** In this setting, $\nu$ becomes the target and we fix $b_t$ to be the drift of an ergodic (backward) process. Then choosing $\Gamma_{0,T} = \mu$, $\gamma^{\pm} = b$ allows us to recover the approaches in Vargas et al. (2023a); Berner et al. (2022). Possible generalisations based on Framework 1′ include IWAE-type objectives, see Appendix C.3.

**Score-based sampling – Föllmer drift:** Finally choosing $b_t(x) = x/t$ we recover Föllmer sampling (Appendix C.3; Follmer, 1984; Vargas et al., 2023b; Zhang & Chen, 2022; Huang et al., 2021b).

## 3 LEARNING FORWARD AND BACKWARD TRANSITIONS SIMULTANEOUSLY

Recall from the introduction that complete flexibility in $a$ and $b$ will render the minima of $D(\overrightarrow{\mathbb{P}}^{\mu,a}|\overleftarrow{\mathbb{P}}^{\nu,b})$ highly nonunique. Furthermore, the approaches surveyed at the end of the previous section circumvent this problem by fixing either $\overrightarrow{\mathbb{P}}^{\mu,a}$ or $\overleftarrow{\mathbb{P}}^{\nu,b}$. However, to leverage the full power of diffusion models, both $\overrightarrow{\mathbb{P}}^{\mu,a}$ or $\overleftarrow{\mathbb{P}}^{\nu,b}$ should be adapted to the problem at hand. In this section, we explore models of this kind, by imposing additional constraints on $a$ and $b$. We end this section by presenting our new CMCD sampler connecting it to prior methodology within VI (Doucet et al., 2022b; Geffner & Domke, 2023; Papamakarios et al., 2017) and OT where we can view CMCD as an instance of entropy regularised OT in the infinite constraint limit (Bernton et al., 2019).

### 3.1 CONNECTION TO ENTROPIC OPTIMAL TRANSPORT

One way of selecting a particular transition between $\mu$ and $\nu$ is by imposing an entropic penalty, encouraging the dynamics to stay close to a prescribed, oftentimes physically or biologically motivated, reference process. Using the notation employed in Framework 1, the *static* Schrödinger problem (Schrödinger, 1931; Léonard, 2014a) is given by

$$\pi^*(\boldsymbol{x}, \boldsymbol{z}) \in \arg\min_{\pi(\boldsymbol{x}, \boldsymbol{z})} \left\{ D_{\mathrm{KL}}(\pi(\boldsymbol{x}, \boldsymbol{z}) || r(\boldsymbol{x}, \boldsymbol{z})) : \pi_{\boldsymbol{x}}(\boldsymbol{x}) = \mu(\boldsymbol{x}), \pi_{\boldsymbol{z}}(\boldsymbol{z}) = \nu(\boldsymbol{z}) \right\}, \quad (16)$$

where $r(\boldsymbol{x}, \boldsymbol{z})$ is the Schrödinger prior encoding additional domain-specific information. In an analogous way, we can introduce a regulariser to the path-space approach of Framework 1' to obtain the dynamic Schrödinger problem

$$\mathbb{P}^* \in \arg\min_{\overrightarrow{\mathbb{P}}^{\mu,a}_T = \nu} \mathbb{E}_{\boldsymbol{Y} \sim \overrightarrow{\mathbb{P}}^{\mu,a}} \left[ \frac{1}{2\sigma^2} \int_0^T \|a_t - f_t\|^2(\boldsymbol{Y}_t) \, \mathrm{d}t \right], \quad (17)$$

that is, the driving vector field $a_t$ determining $\mathbb{P}^*$ should be chosen in such a way that (i), the corresponding diffusion transitions from $\mu$ to $\nu$, and (ii), among such diffusions, the vector field $a_t$ remains close to the prescribed vector field $f_t$, in mean square sense. Under mild conditions, the solutions to (16) and (17) exist and are unique. Further, the static and dynamic viewpoints are related through a mixture-of-bridges construction (assuming that the conditionals $r(\boldsymbol{z}|\boldsymbol{x})$ correspond to the transitions induced by $f_t$), see (Léonard, 2014a, Section 2).

---

[3]Concurrently Richter & Berner (2024) propose an akin framework to ours.

**Iterative proportional fitting (IPF) and the EM algorithm.** It is well known that approximate solutions for $\pi^*(\boldsymbol{x}, \boldsymbol{z})$ and $\mathbb{P}^*$ can be obtained using alternating $D_{\mathrm{KL}}$-projections, keeping one of the marginals fixed in each iteration: Under mild conditions, the sequence defined by

$$\pi^{2n+1}(\boldsymbol{x}, \boldsymbol{z}) = \underset{\pi(\boldsymbol{x}, \boldsymbol{z})}{\arg\min} \left\{ D_{\mathrm{KL}}(\pi(\boldsymbol{x}, \boldsymbol{z}) || \pi^{2n}(\boldsymbol{x}, \boldsymbol{z})) : \ \pi_{\boldsymbol{x}}(\boldsymbol{x}) = \mu(\boldsymbol{x}) \right\}, \tag{18a}$$

$$\pi^{2n+2}(\boldsymbol{x}, \boldsymbol{z}) = \underset{\pi(\boldsymbol{x}, \boldsymbol{z})}{\arg\min} \left\{ D_{\mathrm{KL}}(\pi(\boldsymbol{x}, \boldsymbol{z}) || \pi^{2n+1}(\boldsymbol{x}, \boldsymbol{z})) : \ \pi_{\boldsymbol{z}}(\boldsymbol{z}) = \nu(\boldsymbol{z}) \right\}, \qquad n \geq 0, \tag{18b}$$

with initialisation $\pi^0(\boldsymbol{x}, \boldsymbol{z}) = r(\boldsymbol{x}, \boldsymbol{z})$, converges to $\pi^*(\boldsymbol{x}, \boldsymbol{z})$ as $n \to \infty$ (De Bortoli et al., 2021), and this procedure is commonly referred to as iterative proportional fitting (IPF) (Fortet, 1940; Kullback, 1968; Ruschendorf, 1995) or Sinkhorn updates (Cuturi, 2013). IPF can straightforwardly be modified to the path space setting of (17), and the resulting updates coincide with the Föllmer drift updates discussed in Section C.3, see (Vargas et al., 2021a) and Appendix E.4.

To further demonstrate the coverage of our framework, we establish a connection between IPF and expectation-maximisation (EM) (Dempster et al., 1977), originally devised for finding maximum likelihood estimates in models with latent (or hidden) variables. According to Neal & Hinton (1998), the EM-algorithm can be described in the setting from the introduction, and written in the form

$$\theta_{n+1} = \underset{\theta}{\arg\min} \, \mathcal{L}_{D_{\mathrm{KL}}}(\phi_n, \theta), \qquad \phi_{n+1} = \underset{\phi}{\arg\min} \, \mathcal{L}_{D_{\mathrm{KL}}}(\phi, \theta_{n+1}), \tag{19}$$

with $\mathcal{L}_{D_{\mathrm{KL}}}$ defined as in (5). If the initialisations are matched appropriately, the following result establishes an exact correspondence between the IPF updates in (18) and the EM updates in (19):

**Proposition 3.1** (EM $\iff$ IPF). *Assume that the transition densities $p^\theta(\boldsymbol{x}|\boldsymbol{z})$ and $q^\phi(\boldsymbol{z}|\boldsymbol{x})$ are parameterised with perfect flexibility,[4] and furthermore that the EM-scheme (19) is initialised at $\phi_0$ in such a way that $q^{\phi_0}(\boldsymbol{z}|\boldsymbol{x}) = r(\boldsymbol{z}|\boldsymbol{x})$. Then the IPF iterations in (18) agree with the EM iterations in (19) for all $n \geq 1$, in the sense that*

$$\pi^n(\boldsymbol{x}, \boldsymbol{z}) = q^{\phi_{(n-1)/2}}(\boldsymbol{z}|\boldsymbol{x})\mu(\boldsymbol{x}), \quad \text{for } n \text{ odd}, \quad \pi^n(\boldsymbol{x}, \boldsymbol{z}) = p^{\theta_{n/2}}(\boldsymbol{x}|\boldsymbol{z})\nu(\boldsymbol{z}), \quad \text{for } n \text{ even}. \tag{20}$$

From the proof (Appenix E), it is clear that flexibility of parameterisations is crucial, and thus EM $\iff$ IPF fails for classical VAEs, but holds up to a negligle error for the SDE-parameterisations from Section 2.2, see also Liu et al. (b). Under this assumption, the key observation is that replacing forward-$D_{\mathrm{KL}}$ by reverse-$D_{\mathrm{KL}}$ in one or both of (18a) and (18b) does not – in theory – change the sequence of minimisers.

In practice favoring the EM objectives over IPF can offer an advantage as optimizing with respect to forward-$D_{\mathrm{KL}}$ and backward-$D_{\mathrm{KL}}$ encourages moment-matching and mode-seeking behavior, respectively, and so an alternating scheme as defined in (19) might present a suitable compromise over optimizing a single direction of $D_{\mathrm{KL}}$'s, empirical exploration is left for future work.

Whilst EM and IPF might seem appealing for learning a sampler they both require sequentially solving a series of minimization problems, which we can only solve approximately; this is not only slow but also causes a sequential accumulation of errors arising from each iterate (Vargas et al., 2021a; Fernandes et al., 2021). In order to address both issues we will present a novel approach (CMCD) that similarly to IPF learns both the forward and backward processes whilst preserving the desired uniqueness property. However, in contrast to IPF it does so in an end-to-end fashion and performs updates simultaneously. As an alternative in Appendix E.5 we also discuss a regularised IPF objective and leave further empirical exploration for future work.

## 3.2 SCORE-BASED ANNEALING: THE CONTROLLED MONTE CARLO DIFFUSION SAMPLER

In this section, we fix a prescribed curve of distributions $(\pi_t)_{t \in [0,T]}$, whose scores $\nabla \ln \pi_t$ (and unnormalised densities $\hat{\pi}_t$) are assumed to be available in tractable form; this is the scenario typically encountered in annealed importance sampling (IS) and related approaches towards computing posterior expectations (Neal, 2001; Reich, 2011; Heng et al., 2021; 2020; Arbel et al., 2021; Doucet et al., 2022a). The *Controlled Monte Carlo Diffusion sampler* (CMCD) learns the vector field $\nabla\phi_t$ in

$$\mathrm{d}\boldsymbol{Y}_t = \left(\sigma^2 \nabla \ln \pi_t(\boldsymbol{Y}_t) + \nabla\phi_t(\boldsymbol{Y}_t)\right) \mathrm{d}t + \sigma\sqrt{2}\,\overrightarrow{\mathrm{d}}\,\boldsymbol{W}_t, \qquad \boldsymbol{Y}_0 \sim \pi_0, \tag{21}$$

---

[4]In precise terms, we assume that for any transition densities $p(\boldsymbol{x}|\boldsymbol{z})$ and $q(\boldsymbol{z}|\boldsymbol{x})$, there exist $\theta_*$ and $\phi_*$ such that $p(\boldsymbol{x}|\boldsymbol{z}) = p^{\theta_*}(\boldsymbol{x}|\boldsymbol{z})$ and $q(\boldsymbol{x}|\boldsymbol{z}) = q^{\phi_*}(\boldsymbol{x}|\boldsymbol{z})$.

---

**Algorithm 1** Controlled Monte Carlo Diffusions - Sampling and normalizing constant estimation

---

**Require:** $\pi_0, \pi_T, \pi_t, \sigma, K$ step-sizes $\Delta t_k$, network $f^\phi$ trained via minimising Eq 24
    $\boldsymbol{Y}_0 \sim \pi_0$
    $\ln \boldsymbol{W} = -\ln \pi_0(\boldsymbol{Y}_0)$
    **for** $k = 0$ to $K - 1$ **do**
        $\boldsymbol{Y}_{t_{k+1}} \sim \mathcal{N}\Big(\boldsymbol{Y}_{t_{k+1}}\big|\boldsymbol{Y}_{t_k} + (\sigma^2 \nabla \ln \pi_{t_k} + f^\phi_{t_k})(\boldsymbol{Y}_{t_k})\Delta t_k, 2\sigma^2 \Delta t_k\Big)$
        $\ln \boldsymbol{W} = \ln \boldsymbol{W} + \ln \dfrac{\mathcal{N}\Big(\boldsymbol{Y}_{t_k}\big|\boldsymbol{Y}_{t_{k+1}} + (\sigma^2 \nabla \ln \pi_{t_{k+1}} - f^\phi_{t_{k+1}})(\boldsymbol{Y}_{t_{k+1}})\Delta t_k, 2\sigma^2 \Delta t_k\Big)}{\mathcal{N}\Big(\boldsymbol{Y}_{t_{k+1}}\big|\boldsymbol{Y}_{t_k} + (\sigma^2 \nabla \ln \pi_{t_k} + f^\phi_{t_k})(\boldsymbol{Y}_{t_k})\Delta t_k, 2\sigma^2 \Delta t_k\Big)}$
    $\ln \boldsymbol{W} = \ln \boldsymbol{W} + \ln \pi_T(\boldsymbol{Y}_T)$
    **return** (Estimate of $\ln Z \approx \ln \boldsymbol{W}$, Approximate sample $\boldsymbol{Y}_T$)

---

so that (21) produces the interpolation from the prior $\pi_0$ to the posterior $\pi_T$, i.e., $\overrightarrow{\mathbb{P}}_t^{\pi_0, \sigma^2 \nabla \ln \pi + \nabla \phi} = \pi_t$, for all $t \in [0, T]$. Note that if $\pi_t$ were constant in time ($\pi_t = \pi_0$), then $\phi = 0$ would reduce (21) to equilibrium overdamped Langevin dynamics, preserving $\pi_0$. With $\pi_t$ varying in time, $\nabla \phi_t$ can be thought of as a control enabling transitions between neighbouring densities $\pi_t$ and $\pi_{t+\delta t}$.

To obtain $\nabla \phi_t$ we invoke Framework 1′, but restrict $\overleftarrow{\mathbb{P}}^{\pi_T, b}$ to retain uniqueness. Proposition 2.1 motivates the choice $b_t = (\sigma^2 \nabla \ln \pi_t + \nabla \phi_t) - 2\sigma^2 \nabla \ln \pi_t = -\sigma^2 \nabla \ln \pi_t + \nabla \phi_t$,[5] leading to

$$\mathcal{L}_D^{\mathrm{CMCD}}(\phi) := D\left(\overrightarrow{\mathbb{P}}^{\pi_0, \sigma^2 \nabla \ln \pi + \nabla \phi}, \overleftarrow{\mathbb{P}}^{\pi_T, -\sigma^2 \nabla \ln \pi + \nabla \phi}\right), \qquad (22)$$

which is valid for any choice of divergence $D$. The additional score constraint $b_t = a_t - 2\sigma^2 \nabla \ln \pi_t$ restores uniqueness in Framework 1′ (see Appendix D for a proof):

**Proposition 3.2** (Existence and uniqueness). *Under mild conditions on $(\pi_t)_{t \in [0,T]}$, (22) admits a ($\pi_t$-a.e.) unique minimiser $\phi^*$, up to additive constants, satisfying $\mathcal{L}_{\mathrm{CMCD}}(\phi^*) = 0$.*

Given the optimal vector field $\nabla \phi_t^*$, we can produce samples from $\pi_T$ by simulating (21). Following Zhang & Chen (2022); Vargas et al. (2023a),we can estimate $Z$ in $\pi_T = \hat{\pi}_T / Z_T$ unbiasedly via

$$Z = \mathbb{E}\left[\frac{\mathrm{d}\overleftarrow{\mathbb{P}}^{\hat{\pi}_T, -\sigma^2 \nabla \ln \pi + \nabla \phi}}{\mathrm{d}\overrightarrow{\mathbb{P}}^{\pi_0, \sigma^2 \nabla \ln \pi + \nabla \phi}}\right] = \frac{\mathrm{d}\overleftarrow{\mathbb{P}}^{\hat{\pi}_T, -\sigma^2 \nabla \ln \pi + \nabla \phi^*}}{\mathrm{d}\overrightarrow{\mathbb{P}}^{\pi_0, \sigma^2 \nabla \ln \pi + \nabla \phi^*}}(\boldsymbol{Y}), \qquad (23)$$

where the expectation is taken with respect to (21), and is valid for any (possibly suboptimal) $\nabla \phi_t$. The right-hand side, on the other hand, shows that optimality of $\nabla \phi_t^*$ yields a zero-variance estimator of $Z$, as the statement holds almost surely in $\boldsymbol{Y}$, without taking the expectation. To give a broader perspective, we give the following slight generalisation of a well-known result from statistical physics:

**Proposition 3.3** (Controlled Crooks' fluctuation theorem and Jarzynki's equality). *Following Jarzynski (1997); Chen et al. (2019), define* work *and* free energy *as* $\mathcal{W}_T(\boldsymbol{Y}) := -\int_0^T \sigma^2 \partial_t \ln \hat{\pi}_t(\boldsymbol{Y}_t)\,\mathrm{d}t$, $\mathcal{F}_t := -\sigma^2 \ln Z_t := \sigma^2 \ln(\hat{\pi}_t / \pi_t)$. *Then, we have the* controlled Crooks' identity,

$$\left(\frac{\mathrm{d}\overrightarrow{\mathbb{P}}^{\pi_0, \sigma^2 \nabla \ln \pi + \nabla \phi}}{\mathrm{d}\overleftarrow{\mathbb{P}}^{\pi_T, -\sigma^2 \nabla \ln \pi + \nabla \phi}}\right)(\boldsymbol{Y}) = \exp\left(-\tfrac{1}{\sigma^2}(\mathcal{F}_T - \mathcal{F}_0) + \tfrac{1}{\sigma^2}\mathcal{W}_T(\boldsymbol{Y}) + \mathcal{C}_T^\phi(\boldsymbol{Y})\right),$$

*where* $\mathcal{C}_T^\phi(\boldsymbol{Y}) := -\tfrac{1}{\sigma^2}\int_0^T \nabla \phi_t(\boldsymbol{Y}_t) \cdot \nabla \ln \pi_t(\boldsymbol{Y}_t)\,\mathrm{d}t - \int_0^T \Delta \phi_t(\boldsymbol{Y}_t)\mathrm{d}t$. *By taking expectations and $\phi = 0$, this implies* Jarzynski's equality $\mathbb{E}_{\overrightarrow{\mathbb{P}}^{\pi_0, \sigma^2 \nabla \ln \pi}}[\exp(-\tfrac{1}{\sigma^2}\mathcal{W}_T)] = Z_T / Z_0$.

The proof uses Proposition 2.2 to compute the RND $\mathrm{d}\overrightarrow{\mathbb{P}}^{\pi_0, \sigma^2 \nabla \ln \pi + \nabla \phi} / \mathrm{d}\overleftarrow{\mathbb{P}}^{\pi_T, -\sigma^2 \nabla \ln \pi + \nabla \phi}$, followed by applying Itô's formula to $t \mapsto \ln \hat{\pi}_t(\boldsymbol{Y}_t)$, see Appendix E.2. For $\phi = 0$, we recover Crooks fluctuation theorem (Crooks, 1999), but the additional control allows CMCD to suppress said fluctuations by adjusting the interaction term $\mathcal{C}_T^\phi(\boldsymbol{Y})$. Indeed, prior works (Neal, 2001; Chopin, 2002; Vaikuntanathan & Jarzynski, 2008; Hartmann et al., 2019; Zhang, 2021) have used the Jarzynski equality to estimate $Z$ via importance sampling, but this approach might suffer from high variance, see (Del Moral et al.,

---

[5]Note the additional factor of 2 in Nelson's relation due to the noise scaling $\sigma\sqrt{2}\,\overrightarrow{\mathrm{d}}\,\boldsymbol{W}_t$ in (21).

2006), (Stoltz et al., 2010, Section 4.1.4). In contrast, the CMCD estimator version of (23) achieves zero variance if trained to optimality (see Appendix E.1.1 for a convenient discretised version). Finally, we would like to highlight that Zhong et al. (2023) concurrently proposes this generalisation of Crook's identity using different techniques in their sketch.

Our next result connects CMCD to Section 3.1, showing that minimising (22) can be viewed as jointly solving an infinite number of Schrödinger problems on infinitesimal time intervals:

**Proposition 3.4** (infinitesimal Schrödinger problems). *The minimiser $\phi^*$ can be characterised as follows: For $N \in \mathbb{N}$, partition the interval $[0, T]$ into $N$ subintervals of length $T/N$, and on each subinterval $[(i-1)T/N, iT/N]$, solve the Schrödinger problem (17) with marginals $\mu = \pi_{(i-1)T/N}$, $\nu = \pi_{iT/N}$ and prior drift $f_t = \nabla \ln \pi_t$. Concatenate the solutions to obtain the drift $\nabla \phi^{(N)}$, defined on $[0, T]$. Then, $\nabla \phi^{(N)} \to \nabla \phi$ as $N \to \infty$ in the sense of $L^2([0, T] \times \mathbb{R}^d, \pi)$ (proof in Appendix D.4)*

Note the similarity of this interpretation to the sequential Schrödinger samplers of Bernton et al. (2019). Making specific choices for $D$ in 22, we establish further connections to other methods:

1. For $D = D_{\mathrm{KL}}$, direct computation (see Appendix D.1) based on Proposition 2.2 shows that

$$\mathcal{L}_{D_{\mathrm{KL}}}^{\mathrm{CMCD}}(\phi) = \mathbb{E}_{\boldsymbol{Y} \sim \overrightarrow{\mathbb{P}}^{\pi_0, \sigma^2 \nabla \ln \pi + \nabla \phi}} \left[ \ln \left( \frac{\mathrm{d} \overrightarrow{\mathbb{P}}^{\pi_0, \sigma^2 \nabla \ln \pi + \nabla \phi}}{\mathrm{d} \overleftarrow{\mathbb{P}}^{\pi_T, -\sigma^2 \nabla \ln \pi + \nabla \phi}} \right) (\boldsymbol{Y}) \right]$$

$$= \mathbb{E}\left[ \sigma^2 \int_0^T |\nabla \ln \pi_t(\boldsymbol{Y}_t)|^2 \mathrm{d}t + \frac{1}{\sigma\sqrt{2}} \int_0^T \left( \sigma^2 \nabla \ln \pi_t - \nabla \phi_t \right)(\boldsymbol{Y}_t) \cdot \overleftarrow{\mathrm{d}} \boldsymbol{W}_t - \ln \hat{\pi}_T(\boldsymbol{Y}_T) \right] + \mathrm{const.}$$

$$\approx \mathbb{E}\left[ \ln \frac{\pi_0(\boldsymbol{Y}_0)}{\hat{\pi}(\boldsymbol{Y}_T)} \prod_{k=0}^{K-1} \frac{\mathcal{N}(\boldsymbol{Y}_{t_{k+1}} | \boldsymbol{Y}_{t_k} + (\sigma^2 \nabla \ln \pi_{t_k} + \nabla \ln \phi_{t_k})(\boldsymbol{Y}_{t_k}) \Delta t_k, 2\sigma^2 \Delta t_k)}{\mathcal{N}(\boldsymbol{Y}_{t_k} | \boldsymbol{Y}_{t_{k+1}} + (\sigma^2 \nabla \ln \pi_{t_{k+1}} - \nabla \ln \phi_{t_{k+1}})(\boldsymbol{Y}_{t_{k+1}}) \Delta t_k, 2\sigma^2 \Delta t_k)} \right], \quad (24)$$

the time-discretisation in the third line is derived in Appendix D.5. Our goal is then to numerically minimize $\mathcal{L}_{D_{\mathrm{KL}}}^{\mathrm{CMCD}}(\phi)$ wrt to $\phi$ (for a numerical minimisation scheme see Algorithm 2). Note the first line in (24) is akin to the optimal control type objectives of Föllmer and DDS samplers recalled at the end of Section 2.2, see also (Berner et al., 2022). Setting $\phi = 0$ in the third line recovers Unadjusted Langevin Annealing (ULA), see, e.g., eq. (14) in (Thin et al., 2021) or eq. (21) in (Geffner & Domke, 2023); hence, we can view CMCD as a controlled version of ULA. Setting $\phi = 0$ *only in the numerator* is akin to Monte Carlo Diffusion (MCD), see Algorithm 1 and eq. (34) in (Doucet et al., 2022a). Finally, action matching (Neklyudov et al., 2023) can be recovered from $D = D_{\mathrm{KL}}$ and Framework 1' by choosing the reference $\overrightarrow{\mathbb{P}}^{\Gamma_0, \gamma^+} = \overleftarrow{\mathbb{P}}^{\Gamma_T, \gamma^-}$ in Proposition 2.2 appropriately, see Appendix C.4.

2. For the log-variance divergence $D_{\mathrm{Var}}(\mathbb{Q}, \mathbb{P}) = \mathrm{Var}\left(\ln \frac{\mathrm{d}\mathbb{Q}}{\mathrm{d}\mathbb{P}}\right)$, see Appendix B, we obtain

$$\mathcal{L}_{\mathrm{Var}}^{\mathrm{CMCD}}(\phi) = \mathrm{Var}\left( \ln \frac{\pi_T(\boldsymbol{Y}_T)}{\pi_0(\boldsymbol{Y}_0)} + \int_0^T \Delta \phi_t(\boldsymbol{Y}_t) \, \mathrm{d}t - \sigma\sqrt{2} \int_0^T \nabla \ln \pi_t(\boldsymbol{Y}_t) \circ \mathrm{d}\boldsymbol{W}_t - \sigma^2 \int_0^T |\nabla \ln \pi_t(\boldsymbol{Y}_t)|^2 \mathrm{d}t \right),$$

see Appendix D. Here, $\circ \mathrm{d}\boldsymbol{W}_t$ denotes Stratonovich integration, and the variance is taken with respect to samples from (21). In the limit $\sigma \to 0$, log-Var CMCD enforces an integrated version of the instantaneous change of density formula $\partial_t \ln \pi_t(\boldsymbol{Y}_t) = -\Delta \phi_t(\boldsymbol{Y}_t)$ for continuous-time normalising flows of the form $\dot{\boldsymbol{Y}}_t = \nabla \phi_t(\boldsymbol{Y}_t)$, (Papamakarios et al., 2021, Section 4).

**Remark 4** (Further related work). The task of learning the vector field $\nabla \phi_t$ so that (21) reproduces $(\pi_t)_{t \in [0,T]}$ has been approached from various directions. Reich (2011); Heng et al. (2021); Reich (2022); Vaikuntanathan & Jarzynski (2008) explore methodologies that exploit the characterisation of $\nabla \phi_t$ in terms of the elliptic PDE (50) in Appendix D.3. Arbel et al. (2021) propose to leverage normalising flows sequentially to minimise KL divergences between implied neighboring densities. In an appropriate limiting regime, they recover the SDE (21), see Remark 9. These approaches approximate $\nabla \phi_t$ sequentially in time, whilst CMCD learns $(\nabla \phi_t)_{t \in [0,T]}$ 'all-at-once'.

## 4 EXPERIMENTS

We now empirically demonstrate the performance of the proposed CMCD sampler (24) in both underdamped (detailed in Appendix D) and overdamped (CMCD (OD)), Appendix D.6) formulations on a series of sampling benchmarks. We first replicate the benchmarks from Geffner & Domke (2023) on 6 standard target benchmark distributions. Following the experimental methodology in Geffner &

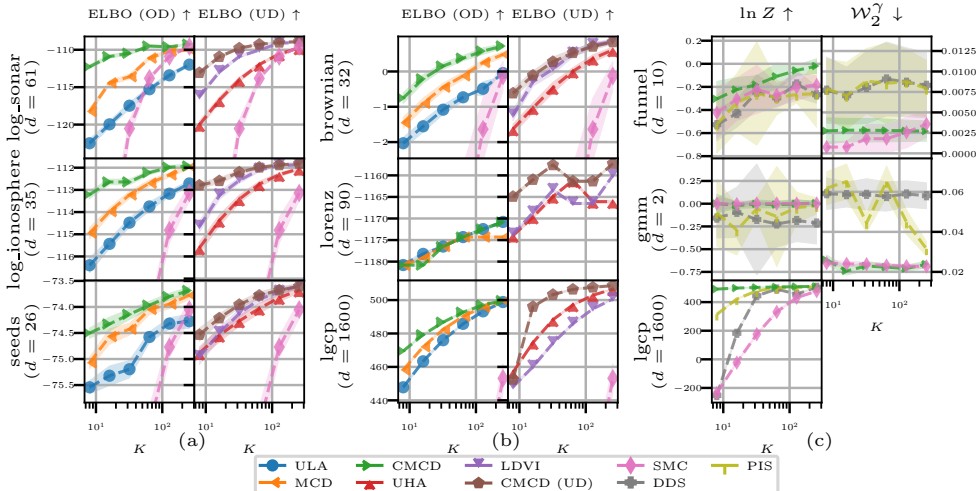

Figure 1: Figure panes a) and b) report ELBOs across methods and targets following the experimental setup in Geffner & Domke (2023), the (OD) and (UD) columns group over and under-damped methods seperately. Figure c) reports IS $\ln Z$ estimates and sample quality (where available) using eOT. Higher ELBO and $\ln Z$ denote better estimates, lower $\mathcal{W}_2^\gamma$ signifies better sample quality.

Domke (2023), we compare against two underdamped baselines, Unadjusted Langevin Annealing (ULA) (Wu et al., 2020; Thin et al., 2021) and Monte Carlo Diffusion (MCD) (Doucet et al., 2022b); and two overdamped baselines, Uncorrected Hamiltonian Annealing (UHA) (Geffner & Domke, 2021; Zhang et al., 2021) and Langevin Diffusion Variational Inference (LDVI) (Geffner & Domke, 2023). Furthermore, we include comparisons of $\ln Z$ estimation on two datasets with known partition function, the `funnel` and `gmm`, and compare against baselines from Vargas et al. (2023a), PIS (Barr et al., 2020; Vargas et al., 2023b; Zhang & Chen, 2022), DDS (Vargas et al., 2023a), and Sequential Monte Carlo Sampler (SMC) (Del Moral et al., 2006; Zhou et al., 2016).

We report the mean ELBO achieved by each method over 30 seeds of sampling, for Euler discretisation steps $K \in \{8, 16, 32, 64, 128, 256\}$, comparing the underdamped and overdamped baselines to their respective CMCD counterparts in Figure 1. We see that both overdamped and underdamped CMCD consistently outperform all baseline methods, especially at low $K$, and in fact, across most targets overdamped CMCD outperforms the underdamped baselines. Figure 1 also reports $\ln Z$ for two target distributions with known $Z$, comparing against PIS, DDS, and SMC. Again, CMCD recovers the log-partition more consistently, even at low $K$. Finally, as another measure of sample quality, we report the entropy-regularised OT distance ($\mathcal{W}_2^\gamma$) between obtained samples and samples from the target for `funnel` and `gmm`. Hyperparameter tuning and other experimental details can be found in Appendix F and we provide a GitHub repository to reproduce our results [6].

## 5 DISCUSSION

Overall we have successfully introduced a novel variational framework bridging VI and transport using modern advances in diffusion models and processes. In particular, we have shown that many existing diffusion-based methods for generative modelling and sampling can be viewed as special instances of our proposed framework. Building on this, we have developed novel objectives for dynamic entropy regularised transports (based on a relationship between the EM and IPF algorithms) and annealed flows (with connections to fluctuation theorems due to Crook and Jarzynski, rooted in statistical physics). Finally, we have explored the CMCD inference scheme obtaining state-of-the-art results across a suite of challenging inference benchmarks. We believe this experimental success is partly due to our approach striking a balance between parametrising a flexible family of distributions whilst being constrained enough such that learning the sampler is not overly expensive (Tzen & Raginsky, 2019b; Vargas et al., 2023c). Future directions can explore optimal schemes for the annealed flow $\pi_t$ (Goshtasbpour et al., 2023) and alternate divergences (Nüsken & Richter, 2021; Richter & Berner, 2024; Midgley et al., 2022).

---

[6] https://github.com/shreyaspadhy/CMCD

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

## A   STOCHASTIC ANALYSIS FOR BACKWARD PROCESSES

In this appendix, we briefly discuss background in stochastic analysis relevant to the SDEs in (12), here repeated for convenience:

$$d\boldsymbol{Y}_t = a_t(\boldsymbol{Y}_t)\, dt + \sigma \overrightarrow{d}\, \boldsymbol{W}_t, \qquad \boldsymbol{Y}_0 \sim \mu, \tag{25a}$$

$$d\boldsymbol{Y}_t = b_t(\boldsymbol{Y}_t)\, dt + \sigma \overleftarrow{d}\, \boldsymbol{W}_t, \qquad \boldsymbol{Y}_T \sim \nu. \tag{25b}$$

Recall that the forward Itô differential $\overrightarrow{d}$ in (25a) is far more commonly denoted simply[7] by d, and theory for the forward SDE (25a) is widely known (Karatzas et al., 1991; Øksendal, 2003). In contrast, reverse-time SDEs of the form (25b) are less common and there are fewer textbook accounts of their interactions with forward SDEs. We highlight Kunita (2019) for an in-depth treatment, and alert the reader to the fact that 'backward stochastic differential equations' as discussed in Zhang (2017); Chen et al. (2022), for instance, are largely unrelated. We therefore refer to (25b) as a 'reverse-time' SDE.

---

[7]...but in this paper we stick to the notation $\overrightarrow{d}$ to emphasise the symmetry of the setting.

**Remark 5** (Notation). We deliberately depart from some of the notation employed in the recent literature (see, for instance, Huang et al. (2021a); Liu et al. (b)) by using $\boldsymbol{Y}_t$ in both (25a) and (25b), and not introducing an auxiliary process capturing the reverse-time dynamics. From a technical perspective, this is justified since $(\boldsymbol{Y}_t)_{0 \leq t \leq T}$ merely represents a generic element in path space, and full information is encoded in the path measures $\mathbb{Q}^{\mu,a} \equiv \overrightarrow{\mathbb{P}}^{\mu,a}$ and $\mathbb{P}^{\nu,b} \equiv \overleftarrow{\mathbb{P}}^{\nu,b}$. Importantly, placing (25a) and (25b) on an equal footing seems essential for a convenient formulation of Proposition 2.2. Slightly departing from the VAE-inspired notation from Section 2.1, we equivalently refer to these path measures by $\overrightarrow{\mathbb{P}}^{\mu,a}$ and $\overleftarrow{\mathbb{P}}^{\nu,b}$, highlighting the symmetry of the setting in (25).

Intuitively, (25) can be viewed as continuous time limits of the Markov chains defined in (9), or, in other words, the Markov chains (9) are the Euler-Maruyama discretisations for (25), see Kloeden et al. (1992, Section 9.1). Throughout, we impose the following:

**Assumption A.1** (Smoothness and linear growth of vector fields). All (time-dependent) vector fields in this paper belong to the set

$$\mathcal{U} := \left\{ a \in C^\infty([0,T] \times \mathbb{R}^d; \mathbb{R}^d) : \quad \text{there exists a constant } C > 0 \right.$$

$$\left. \text{such that } \|a_t(\boldsymbol{x}) - a_t(\boldsymbol{y})\| \leq C \|\boldsymbol{x} - \boldsymbol{y}\|, \text{ for all } t \in [0,T], \ \boldsymbol{x}, \boldsymbol{y} \in \mathbb{R}^d \right\}.$$

The preceding assumption guarantees existence and uniqueness for (25a) and (25b), and it allows us to use Girsanov's theorem in the proof of Proposition 2.2 (Novikov's condition can be shown to be satisfied, see Øksendal (2003, Section 8.6)). Furthermore, Assumption A.1 is sufficient to conclude Nelson's relation (Proposition 2.1), see Haussmann & Pardoux (1985); Millet et al. (1989); Föllmer (2006b) and the discussion in Russo & Vallois (1996). Having said all that, it is possible to substantially weaken Assumption A.1 with more technical effort. Moreover, we can replace the constant $\sigma > 0$ by $\sigma : [0,T] \times \mathbb{R}^d \to \mathbb{R}^{d \times d}$ throughout, assuming sufficient regularity, growth and invertibility properties, and amending the formulas accordingly.

The precise meaning of (25) is given by the integrated formulations

$$\boldsymbol{Y}_t = \boldsymbol{Y}_0 + \int_0^t a_s(\boldsymbol{Y}_s)\,\mathrm{d}s + \int_0^t \sigma \overrightarrow{\mathrm{d}}\,\boldsymbol{W}_s, \qquad\qquad \boldsymbol{Y}_0 \sim \mu, \qquad (27a)$$

$$\boldsymbol{Y}_t = \boldsymbol{Y}_T - \int_t^T b_s(\boldsymbol{Y}_s)\,\mathrm{d}s - \int_t^T \sigma \overleftarrow{\mathrm{d}}\,\boldsymbol{W}_s, \qquad\qquad \boldsymbol{Y}_T \sim \nu, \qquad (27b)$$

where the forward and backward integrals need defining. Roughly speaking, we have

$$\int_{t_0}^{t_1} \boldsymbol{X}_s \cdot \overrightarrow{\mathrm{d}}\,\boldsymbol{Z}_s = \lim_{\text{'}\Delta t \to 0'} \sum_i \boldsymbol{X}_{t_i} \cdot (\boldsymbol{Z}_{t_{i+1}} - \boldsymbol{Z}_{t_i}), \qquad (28a)$$

$$\int_{t_0}^{t_1} \boldsymbol{X}_s \cdot \overleftarrow{\mathrm{d}}\,\boldsymbol{Z}_s = \lim_{\text{'}\Delta t \to 0'} \sum_i \boldsymbol{X}_{t_{i+1}} \cdot (\boldsymbol{Z}_{t_{i+1}} - \boldsymbol{Z}_{t_i}), \qquad (28b)$$

see Remark 3, for 'appropriate' processes $(\boldsymbol{X}_t)_{0 \leq t \leq T}$ and $(\boldsymbol{Z}_t)_{0 \leq t \leq T}$, and where the limit $\Delta t \to 0$ of vanishing step sizes needs careful analysis (see Remark 6 below). The most salient difference between (25a) and (25b) is the fact that $\boldsymbol{X}_{t_i}$ is replaced by $\boldsymbol{X}_{t_{i+1}}$ in (28b).

**Remark 6** (Convergence of the limits in (28)). If we only assume that $\boldsymbol{X}, \boldsymbol{Z} \in C([t_0, t_1]; \mathbb{R}^d)$, possibly pathwise, that is, deterministically, then the limits in (28) might not exist, or when they do, their values might depends on the specific sequence of mesh refinements. The following approaches are available to make the definitions (28) rigorous:

1. Itô calculus (see, for example, Revuz & Yor (2013, Chapter 9)) uses adaptedness and semimartingale properties for the forward integral in (28a), but note that the definition is not pathwise (that is, the limit (28a) is defined up to a set of measure zero). For the backward integrals in (28b) and, importantly for us, in (64), it can then be shown that the relevant processes are (continuous) reverse-time martingales (see Kunita (2019) for a discussion of the corresponding filtrations). The latter property is guaranteed under Assumption A.1, see the discussion around Theorem 2.3 in Russo & Vallois (1996).

2. Föllmer's 'Itô calculus without probabilities' (Föllmer, 2006a) is convenient, since it allows to us to perform calculations using (25) and Proposition 2.2 without introducing filtrations and related stochastic machinery. The caveat is that the results may in principle depend on the sequence of mesh refinements, but under Assumption A.1, those differences only appear on a set of measure zero, see Russo & Vallois (1995); Föllmer & Protter (2000).

3. Similarly, the integrals in (28) can be defined in a pathwise fashion using rough path techniques, see Friz & Hairer (2020, Section 5.4).

For the current paper, the following conversion formulas are crucial,

$$\int_0^t \boldsymbol{X}_s \cdot \overleftarrow{\mathrm{d}} \boldsymbol{Z}_s - \int_0^t \boldsymbol{X}_s \cdot \overrightarrow{\mathrm{d}} \boldsymbol{Z}_s = \langle \boldsymbol{X}, \boldsymbol{Z} \rangle_t, \tag{29a}$$

$$\int_0^t \boldsymbol{X}_s \cdot \overleftarrow{\mathrm{d}} \boldsymbol{Z}_s + \int_0^t \boldsymbol{X}_s \cdot \overrightarrow{\mathrm{d}} \boldsymbol{Z}_s = 2 \int_0^t \boldsymbol{X}_s \circ \boldsymbol{Z}_s, \tag{29b}$$

where $\langle \boldsymbol{X}, \boldsymbol{Z} \rangle$ is the quadratic variation process (if defined, see Russo & Vallois (1995); see in particular equations (3) and (4) therein), and $\circ$ denotes Stratonovich integration. For solutions to (25), we obtain (15) from (29a). In particular, we can often trade backward integrals for divergence terms (see Appendix C.1), using the (backward) martingale properties

$$\mathbb{E}\left[\int_0^t f_t(\boldsymbol{Y}_t) \cdot \overrightarrow{\mathrm{d}} \boldsymbol{W}_t\right] = 0, \quad \text{if } (\boldsymbol{Y}_t)_{0 \le t \le T} \text{ solves } (25a), \tag{30a}$$

$$\mathbb{E}\left[\int_t^T f_t(\boldsymbol{Y}_t) \cdot \overleftarrow{\mathrm{d}} \boldsymbol{W}_t\right] = 0, \quad \text{if } (\boldsymbol{Y}_t)_{0 \le t \le T} \text{ solves } (25b). \tag{30b}$$

## B  VARIATIONAL INFERENCE AND DIVERGENCES

Various concepts well-known in the variational inference community have direct counterparts in the diffusion setting. In this appendix we review a few that are directly relevant to this paper.

**Maximum likelihood.** Framework 1 with $D = D_{\mathrm{KL}}$ leads via direct calculations to

$$\mathcal{L}_{D_{\mathrm{KL}}}(\phi, \theta) = -\mathbb{E}_{\boldsymbol{x} \sim \mu(\boldsymbol{x})} \overbrace{\left[\int \ln \frac{p^\theta(\boldsymbol{x}|\boldsymbol{z})\nu(\boldsymbol{z})}{q^\phi(\boldsymbol{z}|\boldsymbol{x})} q^\phi(\mathrm{d}\boldsymbol{z}|\boldsymbol{x})\right]}^{=\mathrm{ELBO}_x(\phi,\theta)} + \int \ln \mu(\boldsymbol{x})\mu(\mathrm{d}\boldsymbol{x}), \tag{31}$$

so that maximising $\mathbb{E}_{\boldsymbol{x} \sim \mu(\boldsymbol{x})}[\mathrm{ELBO}_x(\phi, \theta)]$ is equivalent to minimising $\mathcal{L}_{D_{\mathrm{KL}}}(\phi, \theta)$.

However, the traditional approach (Blei et al., 2017; Kingma et al., 2019) towards the *evidence lower bound* (ELBO) in (31) is via maximum likelihood in latent variable models. Using the notation and set-up from the introduction, one can show using Jensen's inequality (or dual representations of the KL divergence), that

$$\ln \left(\int p_\theta(\boldsymbol{x}, \boldsymbol{z}) \,\mathrm{d}\boldsymbol{z}\right) = \ln p_\theta(\boldsymbol{x}) \ge \mathrm{ELBO}_{\boldsymbol{x}}(\phi, \theta), \tag{32}$$

with equality if and only if $q_\phi(\boldsymbol{z}|\boldsymbol{x}) = p_\theta(\boldsymbol{z}|\boldsymbol{x})$. The bound in (32) motivates maximising the (tractable) right-hand side, performing model selection (according to Bayesian evidence) and posterior approximation (in terms of the variational family $q_\phi(\boldsymbol{z}|\boldsymbol{x})$) at the same time. The calculation in (31) shows that this objective can equivalently be derived from Framework 1 and connected to the KL divergence between the joint distributions $q_\phi(\boldsymbol{x}, \boldsymbol{z})$ and $p_\theta(\boldsymbol{x}, \boldsymbol{z})$.

**Reparameterisation trick (Kingma & Welling, 2014; Rezende et al., 2014).** For optimising $\mathrm{ELBO}_{\boldsymbol{x}}(\phi, \theta)$, it is crucial to select efficient low-variance gradient estimators. In this context, it has been observed that reparameterising $\boldsymbol{z} \sim q_\phi(\boldsymbol{z}|\boldsymbol{x})$ in the form $\boldsymbol{z} = g(\epsilon, \phi, \boldsymbol{x})$, see Kingma et al. (2019, Section 2.4.1), substantially stabilises the training procedure. Here, $\epsilon$ is an auxiliary random variable with tractable 'base distribution' that is independent of $\phi$ and $\boldsymbol{x}$, and $g$ is a deterministic function (transforming $\epsilon$ into $\boldsymbol{z}$), parameterised by $\phi$ and $\boldsymbol{x}$. We would like to point out that many (although not all, see below) objectives in diffusion modelling are already reparameterised, since the SDEs (12) transform the 'auxiliary' variables $(\boldsymbol{W}_t)_{0 \le t \le T}$ into $(\boldsymbol{Y}_t)_{0 \le t \le T}$. With this viewpoint, the

vector field $a_t$ corresponds to the parameter $\phi$, $(\boldsymbol{W}_t)_{0 \leq t \leq T}$ corresponds to $\epsilon$, and $g$ corresponds to the solution map associated to the SDE (12a), sometimes referred to as the Itô map. In this sense, the objectives (74), (39) and (40) are reparameterised, but $\mathcal{L}_{\text{Var}}^{\text{CMCD}}$ from Section 3.2 is not if the gradients are detached as in (Nüsken & Richter, 2021; Richter et al., 2020; Richter & Berner, 2024). We mention in passing that *sticking the landing* (Roeder et al., 2017) offers a further variance reduction close to optimality, and that the same method can be employed for diffusion objectives, see Vargas et al. (2023b); Xu et al. (2021).

**Reinforce gradient estimators.** As an alternative to the KL-divergence, Nüsken & Richter (2021) investigated the family of *log-variance divergences*

$$D_{\text{Var}}^u(q||p) = \text{Var}_{\boldsymbol{x} \sim u}\left(\ln \frac{\mathrm{d}q}{\mathrm{d}p}(\boldsymbol{x})\right), \tag{33}$$

parameterised by an auxiliary distribution $u$, in order to connect variational inference to backward stochastic differential equations (Zhang, 2017). The fact that gradients of (33) do not have to be taken with respect to $\boldsymbol{x}$ (see Remark (Nüsken & Richter, 2021; Richter et al., 2020)) reduces the computational cost and provides additional flexibility in the choice of $u$, but the gradient estimates potentially suffer from higher variance since the reparameterisation trick is not available. The latter drawback is alleviated somewhat by the fact that particular choices of $u$ can be linked to control variate enhanced reinforce gradient estimators (Richter et al., 2020) that are particularly useful when reparameterisation is not available (such as in discrete models). We note that the same divergence has also been used as a variational inference objective in El Moselhy & Marzouk (2012).

**Importance weighted autoencoders (IWAE).** Burda et al. (2015) have developed a multi-sample version of $\text{ELBO}_{\boldsymbol{x}}(\phi, \theta)$ that achieves a tighter lower bound on the marginal log-likelihood in (32). To develop similar objectives in a diffusion setting, we observe that for each $K \geq 1$,

$$D_{\text{KL}}^{(K)}(q||p) = \mathbb{E}_{x_1,\ldots,x_K \overset{iid}{\sim} q}\left[\ln\left(\frac{1}{K}\sum_{i=1}^K \frac{\mathrm{d}q}{\mathrm{d}p}(x_i)\right)\right] \tag{34}$$

defines a generalised KL divergence[8] that reproduces the IWAE lower bound as per Framework 1, in the sense of equation (31). To the best of our knowledge, the precise formulation in (34) is new, but similar to the previous works Hernandez-Lobato et al. (2016); Li & Turner (2016); Daudel et al. (2022). We exhibit an example of (34) applied in a diffusion context in Section C.3, see Remark 7.

## C  CONNECTIONS TO PREVIOUS WORK

### C.1  DISCUSSION OF EQUIVALENT EXPRESSIONS FOR $D_{\text{KL}}(\overrightarrow{\mathbb{P}}^{\mu,a}||\overleftarrow{\mathbb{P}}^{\nu,b})$

Notice that we can realise samples from $\overleftarrow{\mathbb{P}}^{\nu,b}$ both via the reverse-time SDE in (12b) or via its time reversal given by the following forward SDE (Nelson, 1967; Anderson, 1982; Haussmann & Pardoux, 1985):

$$\mathrm{d}\widehat{\boldsymbol{Y}}_t = \left(b_{T-t}(\widehat{\boldsymbol{Y}}_t) - \sigma^2 \nabla \ln \overleftarrow{\rho}_{T-t}^{\nu,b}(\widehat{\boldsymbol{Y}}_t)\right)\mathrm{d}t + \sigma\,\overrightarrow{\mathrm{d}}\,\boldsymbol{W}_t, \qquad \widehat{\boldsymbol{Y}}_0 \sim \nu, \tag{35}$$

using $\widehat{\boldsymbol{Y}}_t := \widehat{\boldsymbol{Y}}_{T-t}$. This allows us to obtain an expression for $D_{\text{KL}}(\overrightarrow{\mathbb{P}}^{\mu,a}|\overleftarrow{\mathbb{P}}^{\nu,b})$ via Girsanov's theorem:

$$D_{\text{KL}}(\overrightarrow{\mathbb{P}}^{\mu,a}||\overleftarrow{\mathbb{P}}^{\nu,b}) = D_{\text{KL}}(\overleftarrow{\rho}_0^{\nu,b}||\nu) + \mathbb{E}\left[\frac{1}{2\sigma^2}\int_0^T \left|\left|a_t(\boldsymbol{Y}_t) - \left(b_t(\boldsymbol{Y}_t) - \sigma^2 \nabla \ln \overleftarrow{\rho}_t^{\nu,b}(\boldsymbol{Y}_t)\right)\right|\right|^2 \mathrm{d}t\right]. \tag{36}$$

However there are several terms here that we cannot estimate or realise in a tractable manner, one being the score $\nabla \ln \overleftarrow{\rho}_t^{\nu,b}$ and the other being sampling from the distribution $\overleftarrow{\rho}_0^{\nu,b}$.[9]

---

[8]Indeed, by Jensen's inequality, we have that $D_{\text{KL}}^{(K+1)}(q||p) \geq D_{\text{KL}}^{(K)}(q||p)$, so that in particular $q \neq p$ implies $D_{\text{KL}}^{(K)}(q||p) > 0$.

[9]When $\widehat{\boldsymbol{Y}}_t$ is an OU process and $\mu$ is Gaussian we are in the traditional DDPM setting (Song et al., 2021) and these two quantities admit the classical tractable score matching approximations

In order to circumvent the score term, the authors Vargas et al. (2021b); Chen et al. (2022) use the Fokker-Plank (FPK) equation and integration by parts, respectively, trading of the score with a divergence term, whilst Huang et al. (2021a) use a variant of the Feynman Kac formula to arrive at an equivalent solution. From Proposition 2.2, we can avoid the divergence entirely and replace it by a backwards integral (making use of the conversion formula (15) and the fact that the ensuing forward integral is zero in expectation). As hinted at in Remark 3, this replacement might have favourable variance-reducing properties, but numerical evidence would be necessary.

## C.2 SCORE-BASED GENERATIVE MODELING

Generative modeling is concerned with the scenario where $\mu(\boldsymbol{x})$ can be sampled from (but its density is unknown), and the goal is to learn a backward diffusion as in (12b) that allows us to generate further samples from $\mu(\boldsymbol{x})$, see Song et al. (2021). We may fix a reference forward drift $a_t$, and, motivated by Proposition 2.1, parameterise the backward drift as $b_t = a_t - s_t$, so that in the case when $\overrightarrow{\mathbb{P}}^{\mu,a} = \overleftarrow{\mathbb{P}}^{\nu,b}$, the variable drift component $s_t$ will represent the score $\sigma^2 \nabla \ln \rho_t^{\mu,a}$. When the diffusion associated to $a_t$ is ergodic and $T$ is large, $\overrightarrow{\mathbb{P}}^{\mu,a} = \overleftarrow{\mathbb{P}}^{\nu,b}$ requires that $\nu(\boldsymbol{z})$ is close to the corresponding invariant measure. Choosing $\gamma_t^- = a_t$, and, for simplicity $\sigma = 1$, direct calculations using Proposition 2.2 show that

$$\mathcal{L}_{\mathrm{ISM}}(s) := D_{\mathrm{KL}}(\overrightarrow{\mathbb{P}}^{\mu,a} || \overleftarrow{\mathbb{P}}^{\nu,a-s}) = \mathbb{E}_{\boldsymbol{Y} \sim \overrightarrow{\mathbb{P}}^{\mu,a}} \left[ \frac{1}{2} \int_0^T s_t^2(\boldsymbol{Y}_t) \, \mathrm{d}t + \int_0^T (\nabla \cdot s_t)(\boldsymbol{Y}_t) \, \mathrm{d}t \right] + \mathrm{const.}$$
(37)

recovers the implicit score matching objective (Hyvärinen & Dayan, 2005), up to a constant that does not depend on $s_t$.

*Proof.* We start by noticing that the contributions in (14a) and (14b) do not depend on $s_t$, and can therefore be absorbed in the constant in (37) Notice that the precise forms of $\Gamma_0$, $\Gamma_T$ and $\gamma^+$ are left unspecified or unknown, but this does not affect the argument. We find

$$D_{\mathrm{KL}}(\overrightarrow{\mathbb{P}}^{\mu,a} || \overleftarrow{\mathbb{P}}^{\nu,a-s}) = \mathbb{E}_{\boldsymbol{Y} \sim \overrightarrow{\mathbb{P}}^{\mu,a}} \left[ \int_0^T s_t(\boldsymbol{Y}_t) \cdot \left( \overleftarrow{\mathrm{d}} \, \boldsymbol{Y}_t - \frac{1}{2} \left( 2a_t - s_t \right) (\boldsymbol{Y}_t) \, \mathrm{d}t \right) \right] + \mathrm{const.}$$

$$= \mathbb{E} \left[ \int_0^T s_t(\boldsymbol{Y}_t) \cdot \left( \sigma \overleftarrow{\mathrm{d}} \, \boldsymbol{W}_t + \frac{1}{2} s_t(\boldsymbol{Y}_t) \, \mathrm{d}t \right) \right] + \mathrm{const.}$$

$$= \mathbb{E} \left[ \frac{1}{2} \int_0^T s_t^2(\boldsymbol{Y}_t) \, \mathrm{d}t + \int_0^T (\nabla \cdot s_t)(\boldsymbol{Y}_t) \, \mathrm{d}t \right] + \mathrm{const.},$$

where in the first line we use Proposition 2.2 together with $b_t = a_t - s_t$ and $\gamma_t^- = a_t$, and to proceed to the second line we substitute $\overleftarrow{\mathrm{d}} \, \boldsymbol{Y}_t$ using the SDE in (12a). The last equality follows from the conversion formula between forward and backward Itô integrals, see (15), and the fact that forward integrals with respect to Brownian motion have zero (forward) expectation, see (30a). □

Notice that the nonuniqueness in Framework 1′ has been circumvented by fixing the forward drift $a_t$; indeed $\mathcal{L}_{\mathrm{ISM}}$ is convex in $s$, confirming Note that using integration by parts, $\mathcal{L}_{\mathrm{ISM}}$ is equivalent to denoising score matching (Song et al., 2020; 2021):

$$D_{\mathrm{KL}}(\overrightarrow{\mathbb{P}}^{\mu,a} || \overleftarrow{\mathbb{P}}^{\nu,a-s}) = \mathbb{E}_{\boldsymbol{Y} \sim \overrightarrow{\mathbb{P}}^{\mu,a}} \left[ \frac{1}{2\sigma^2} \int_0^T \left\| s_t(\boldsymbol{Y}_t) - \nabla \ln \rho_{t|0}^{\mu,a}(\boldsymbol{Y}_t | \boldsymbol{Y}_0) \right\|^2 \, \mathrm{d}t \right] + \mathrm{const..} \quad (38)$$

Framework 1′ accommodates modifications of (37); in particular the divergence term in (37) can be replaced by a backward integral, see Appendix C.1 and Remark 3. Note that the settings discussed in this section are also akin to the formulations in Kingma et al. (2021); Huang et al. (2021a).

Finally, it is worth highlighting that this setting is not limited to ergodic models and can in fact accommodate finite time models in the exact same fashion as the Föllmer drift is used for sampling (Section C.3) by using a Doob's transform (Rogers & Williams, 2000) based SDE for $\overrightarrow{\mathbb{P}}^{\mu,a}$ as opposed to the classical VP-SDE see Example 2.4 in Ye et al. (2022).

### C.3  SCORE-BASED SAMPLING

Consider the setting when $\nu(z)$ is a target distribution that can be evaluated pointwise up to a normalisation constant. In order to construct a diffusion process that transports an appropriate auxiliary distribution $\mu(x)$ to $\nu(z)$, one approach is to fix a drift $b_t$ in the backward diffusion (12b), and then learn the corresponding forward diffusion (12a) by minimising $a \mapsto D(\overrightarrow{\mathbb{P}}^{\mu,a} | \overleftarrow{\mathbb{P}}^{\nu,b})$. Tractability of this objective requires that $\mu := \overleftarrow{\mathbb{P}}_0^{\nu,b}$ be known explicitly, at least approximately. In the following we review possible choices.

**The Föllmer drift.** Choosing $b_t(x) = x/t$, one can show using Doob's transform (Rogers & Williams, 2000, Theorem 40.3(iii)), that $\overleftarrow{\mathbb{P}}_0^{\nu,b}(x) = \delta(x)$, for any terminal distribution $\nu(z)$. Hence, minimising $a \mapsto D_{\mathrm{KL}}(\overrightarrow{\mathbb{P}}^{\delta_0,a} | \overleftarrow{\mathbb{P}}^{\nu,b})$ leads to a tractable objective. In particular consider the choice $\Gamma_0 = \delta_0$, $\gamma^+ = 0$, corresponding to a standard Brownian motion, then it follows that $\gamma^- = \frac{x}{t}$, $\Gamma_T = \mathcal{N}(0, T\sigma^2)$ and thus via Proposition 2.2:

$$D_{\mathrm{KL}}(\overrightarrow{\mathbb{P}}^{\delta_0,a} | \overleftarrow{\mathbb{P}}^{\nu,b}) = \mathbb{E}_{\boldsymbol{Y} \sim \overrightarrow{\mathbb{P}}^{\mu,a}} \left[ \frac{1}{\sigma^2} \int_0^T a^2(\boldsymbol{Y}_t)\, \mathrm{d}t + \ln\left( \frac{\mathrm{d}\mathcal{N}(0, T\sigma^2)}{\mathrm{d}\nu} \right)(\boldsymbol{Y}_T) \right] + \mathrm{const.}, \quad (39)$$

in accordance with (Dai Pra, 1991; Vargas et al., 2023b; Zhang & Chen, 2022). For further details, see Follmer (1984); Vargas et al. (2023b); Zhang & Chen (2022); Huang et al. (2021b). As hinted at in Appendix B, replacing $D_{\mathrm{KL}}$ in (39) by the log-variance divergence (33) leads to an objective that directly links to BSDEs, see (Nüsken & Richter, 2021, Section 3.2).

**Ergodic diffusions.** Vargas et al. (2023a); Berner et al. (2022) fix a backward drift $b_t$ that induces an ergodic backward diffusion, so that for large $T$, the marginal at initial time $\overleftarrow{\mathbb{P}}_{t=0}^{\nu,b}$ is close to the corresponding invariant distribution, and in particular (almost) independent of $\nu(z)$.[10] Defining $\mu := \overleftarrow{\mathbb{P}}_{t=0}^{\nu,b}$, Vargas et al. (2023a); Berner et al. (2022) set out to minimise the denoising diffusion sampler loss $\mathcal{L}_{\mathrm{DDS}}(f) := D_{\mathrm{KL}}(\overrightarrow{\mathbb{P}}^{\mu,b+\sigma^2 f} | \overleftarrow{\mathbb{P}}^{\nu,b})$. Choosing the reference process to be $\Gamma_{0,T} = \mu$, $\gamma^\pm = b$ (that is, the reference process is at stationarity, with invariant measure $\mu(z)$), direct calculation based on (14) shows that

$$\mathcal{L}_{\mathrm{DDS}}(f) = \mathbb{E}_{\boldsymbol{Y} \sim \overrightarrow{\mathbb{P}}^{\mu,b+\sigma^2 f}} \left[ \sigma^2 \int_0^T f^2(\boldsymbol{Y}_t)\, \mathrm{d}t + \ln\left( \frac{\mathrm{d}\Gamma_T}{\mathrm{d}\nu} \right)(\boldsymbol{Y}_T) \right], \quad (40)$$

**Remark 7** (IWAE-objective). In line with (34), we may also consider the multi-sample objective

$$\mathcal{L}_{\mathrm{DDS}}^{(K)}(f) := D_{\mathrm{KL}}^{(K)}(\overrightarrow{\mathbb{P}}^{\mu,b+\sigma^2 f} | \overleftarrow{\mathbb{P}}^{\nu,b})$$

$$= \mathbb{E}_{\boldsymbol{Y}^1,\dots,\boldsymbol{Y}^K \overset{iid}{\sim} \overrightarrow{\mathbb{P}}^{\mu,b+\sigma^2 f}} \left[ \ln\left( \frac{1}{K} \sum_{i=1}^K \exp\left( \sigma^2 \int_0^T f^2(\boldsymbol{Y}_t^i)\, \mathrm{d}t + \ln\left( \frac{\mathrm{d}\Gamma_T}{\mathrm{d}\nu} \right)(\boldsymbol{Y}_T^i) \right) \right) \right]$$

*Proof.* We start by noticing that the choice $\gamma_t^- = b_t$ cancels the terms in (14c), and the choice $\Gamma_0 = \mu$ cancels the first term in (14a). Using $a_t = b_t + \sigma^2 f_t$, we therefore obtain

$$\mathcal{L}_{\mathrm{DDS}}(f) = D_{\mathrm{KL}}(\overrightarrow{\mathbb{P}}^{\mu,b+\sigma^2 f} | \overleftarrow{\mathbb{P}}^{\nu,b}) \tag{41a}$$

$$= \mathbb{E}\left[ \sigma^2 \int_0^T f_t(\boldsymbol{Y}_t) \cdot \left( (b_t + f_t)(\boldsymbol{Y}_t)\, \mathrm{d}t - \tfrac{1}{2}(2b_t + f_t)(\boldsymbol{Y}_t)\, \mathrm{d}t \right) + \ln\left( \frac{\mathrm{d}\Gamma_T}{\mathrm{d}\nu} \right)(\boldsymbol{Y}_T) \right]$$

$$= \mathbb{E}\left[ \sigma^2 \int_0^T f_t^2(\boldsymbol{Y}_t)\, \mathrm{d}t + \ln\left( \frac{\mathrm{d}\Gamma_T}{\mathrm{d}\nu} \right)(\boldsymbol{Y}_T) \right]. \tag{41b}$$

As is implicit in Berner et al. (2022), it is also possible to choose $\gamma^\pm = 0$ for the reference process, with $\Gamma_0 = \Gamma_T = \mathrm{Leb}$, the Lebesgue measure on $\mathbb{R}^d$. We notice in passing that although the Lebesgue

---

[10]Vargas et al. (2023a) chose a (time-inhomoegenous) backward Ornstein-Uhlenbeck process, so that $\overleftarrow{\mathbb{P}}_{t=0}^{\nu,b}$ is close to a Gaussian, but generalisations are straightforward.

measure is not normalisable, it is invariant under Brownian motion (the forward and backward drifts are both zero), and the arguments can be made rigorous by a limiting argument (take Gaussians with diverging variances), or by using the techniques in Léonard (2014a, Appendix A.1). By similar calculations as above, we obtain

$$\mathcal{L}_{\text{DDS}}(f) = \mathbb{E}\left[\sigma^2 \int_0^T f_t^2(\boldsymbol{Y}_t)\,\mathrm{d}t - \tfrac{1}{\sigma}\int_0^T b_t(\boldsymbol{Y}_t)\cdot\overleftarrow{\mathrm{d}}\,\boldsymbol{W}_t + \ln\mu(\boldsymbol{Y}_0) - \ln\nu(\boldsymbol{Y}_T)\right] \tag{42a}$$

$$= \mathbb{E}\left[\sigma^2 \int_0^T f_t^2(\boldsymbol{Y}_t)\,\mathrm{d}t - \int_0^T (\nabla\cdot b_t)(\boldsymbol{Y}_t)\,\mathrm{d}t - \ln\nu(\boldsymbol{Y}_T)\right] + \text{const.}, \tag{42b}$$

where we overload notation and denote the Lebesgue densities of $\mu$ and $\nu$ with the same letters. In the second line we have used the conversion formula in (15), together with the fact that the forward Itô integrals are forward martingales (Kunita, 2019), and therefore have zero expectation. Comparing (40) and (42b), we notice the additional divergence term, due to the fact that the choice $\gamma^- = 0$ does not cancel the terms in (64). See also the discussion in Appendix C.1. $\qquad\square$

Finally we note that whilst the work in Berner et al. (2022) focuses on exploring a VP-SDE-based approach which is ergodic, their overarching framework generalises beyond ergodic settings, notice this objective is akin to the KL expressions in Vargas (2021, Proposition 1) and Liu et al. (a, Proposition 9).

## C.4 Action matching (Neklyudov et al., 2023)

Similar to our approach in Section 3.2, Neklyudov et al. (2023) fix a curve of distributions $(\pi_t)_{t\in[0,T]}$. In contrast to us, they assume that samples from $\pi_t$ are available, for each $t \in [0,T]$ (but scores and unnormalised densities are not). Still, we can use Framework $1'$ to rederive their objective:

Akin to the proof of Proposition 3.2, under mild conditions on $(\pi_t)_{t\in[0,T]}$, there exists a unique vector field $\nabla\phi_t^*$ that satisfies the Fokker-Planck equation

$$\partial_t \pi_t + \nabla\cdot(\pi_t \nabla\phi_t^*) = \tfrac{\sigma^2}{2}\Delta\pi_t. \tag{43}$$

We can now use the reference process $\overrightarrow{\mathbb{P}}^{\pi_0,\nabla\phi^*}$ (that is, $\Gamma_0 = \pi_0$, $\gamma_t^+ = \nabla\phi_t^*$, $\Gamma_T = \pi_T$, $\gamma_t^- = \nabla\phi_t^* - \sigma^2\nabla\ln\pi_t$) to compute the objective

$$\psi \mapsto D_{\text{KL}}\big(\overrightarrow{\mathbb{P}}^{\pi_0,\nabla\psi} || \overleftarrow{\mathbb{P}}^{\pi_T,\nabla\psi - \sigma^2\nabla\ln\pi}\big),$$

relying on the same calculational techniques as in Sections C.2 and C.3 (the particular choice of reference process cancels the terms in (14a)). Notice that the parameterisation in this objective constrains the target diffusion to have time-marginals $\pi_t$, just as in Section 3.2. By direct calculation, we obtain (up to a factor of $2/\sigma^2$) the action-gap in equation (5) in Neklyudov et al. (2023). Indeed, we see that

$$D_{\text{KL}}\big(\overrightarrow{\mathbb{P}}^{\pi_0,\nabla\psi} || \overleftarrow{\mathbb{P}}^{\pi_T,\nabla\psi - \sigma^2\nabla\ln\pi}\big) = \mathbb{E}_{\overrightarrow{\mathbb{P}}^{\pi_0,\nabla\psi}}\left[\ln\left(\frac{\mathrm{d}\overrightarrow{\mathbb{P}}^{\pi_0,\nabla\psi}}{\mathrm{d}\overleftarrow{\mathbb{P}}^{\nabla\psi - \sigma^2\nabla\ln\pi}}\right)\right]$$

$$= \mathbb{E}\left[\tfrac{1}{\sigma^2}\int_0^T (\nabla\psi_t - \nabla\phi_t^*)^2(\boldsymbol{Y}_t)\,\mathrm{d}t - \tfrac{1}{\sigma}\int_0^T (\nabla\psi_t - \nabla\phi_t^*)(\boldsymbol{Y}_t)\cdot\overleftarrow{\mathrm{d}}\,\boldsymbol{W}_t\right.$$

$$\left. - \int_0^T \nabla\ln\pi_t(\boldsymbol{Y}_t)\cdot(\nabla\psi_t - \nabla\phi_t^*)(\boldsymbol{Y}_t)\,\mathrm{d}t\right]$$

$$= \mathbb{E}\left[\tfrac{1}{\sigma^2}\int_0^T (\nabla\psi_t - \nabla\phi_t^*)^2(\boldsymbol{Y}_t)\,\mathrm{d}t\right],$$

where in the last line we have used the conversion formula (15) together with (30a) to compute

$$\mathbb{E}\left[\tfrac{1}{\sigma}\int_0^T (\nabla\psi_t - \nabla\phi_t^*)(\boldsymbol{Y}_t)\cdot\overleftarrow{\mathrm{d}}\,\boldsymbol{W}_t\right] = \mathbb{E}\left[\int_0^T (\nabla\cdot(\nabla\psi_t - \nabla\phi_t^*))(\boldsymbol{Y}_t)\,\mathrm{d}t\right]$$

$$= \int_0^T\int_{\mathbb{R}^d}(\nabla\cdot(\nabla\psi_t - \nabla\phi_t^*))(\boldsymbol{x})\pi_t(\mathrm{d}\boldsymbol{x})\,\mathrm{d}t = -\int_0^T\int_{\mathbb{R}^d}(\nabla\psi_t - \nabla\phi_t^*)(\boldsymbol{x})\cdot\nabla\ln\pi_t(\boldsymbol{x})\pi_t(\mathrm{d}\boldsymbol{x})\,\mathrm{d}t$$

$$= \mathbb{E}\left[\int_0^T \nabla\ln\pi_t(\boldsymbol{Y}_t)\cdot(\nabla\psi_t - \nabla\phi_t^*)(\boldsymbol{Y}_t)\,\mathrm{d}t\right]$$

and cancel the two last terms in the penultimate line.

# D    CONTROLLED MONTE CARLO DIFFUSIONS (SECTION 3.2)

## D.1    DERIVATION OF $\mathcal{L}_{D_{\mathrm{KL}}}^{\mathrm{CMCD}}$

The proof uses Proposition 2.2, choosing $\Gamma_0 = \Gamma_T$ to be the Lebesgue measure, with $\gamma^+ = \gamma^- = 0$ (but notice that $\sigma$ in (14) needs to be replaced by $\sigma\sqrt{2}$ due to the scaling in (21)). We compute

$$\mathcal{L}_{D_{\mathrm{KL}}}^{\mathrm{CMCD}}(\phi) = \mathbb{E}_{\boldsymbol{Y}\sim\overrightarrow{\mathbb{P}}^{\pi_0,\sigma^2\nabla\ln\pi+\nabla\phi}}\left[\ln\left(\frac{\mathrm{d}\,\overrightarrow{\mathbb{P}}^{\pi_0,\sigma^2\nabla\ln\pi+\nabla\phi}}{\mathrm{d}\,\overleftarrow{\mathbb{P}}^{\pi_T,-\sigma^2\nabla\ln\pi+\nabla\phi}}\right)(\boldsymbol{Y})\right]$$

$$=\mathbb{E}\left[\ln\pi_0(\boldsymbol{Y}_0) - \ln\pi_T(\boldsymbol{Y}_T)\right]$$

$$+ \mathbb{E}\left[\tfrac{1}{2\sigma^2}\int_0^T(\sigma^2\nabla\ln\pi_t + \nabla\phi_t)(\boldsymbol{Y}_t)\cdot\left(\overrightarrow{\mathrm{d}}\,\boldsymbol{Y}_t - \tfrac{1}{2}(\sigma^2\nabla\ln\pi_t + \nabla\phi_t)(\boldsymbol{Y}_t)\,\mathrm{d}t\right)\right]$$

$$- \mathbb{E}\left[\tfrac{1}{2\sigma^2}\int_0^T(-\sigma^2\nabla\ln\pi_t + \nabla\phi_t)(\boldsymbol{Y}_t)\cdot\left(\overleftarrow{\mathrm{d}}\,\boldsymbol{Y}_t - \tfrac{1}{2}(-\sigma^2\nabla\ln\pi_t + \nabla\phi_t)(\boldsymbol{Y}_t)\,\mathrm{d}t\right)\right]$$

$$=\mathbb{E}\left[\ln\pi_0(\boldsymbol{Y}_0) - \ln\pi_T(\boldsymbol{Y}_T)\right]$$

$$+ \mathbb{E}\left[\tfrac{1}{2\sigma^2}\int_0^T(\sigma^2\nabla\ln\pi_t + \nabla\phi_t)(\boldsymbol{Y}_t)\cdot\overrightarrow{\mathrm{d}}\,\boldsymbol{Y}_t\right] - \mathbb{E}\left[\tfrac{1}{2\sigma^2}\int_0^T(-\sigma^2\nabla\ln\pi_t + \nabla\phi_t)(\boldsymbol{Y}_t)\cdot\overleftarrow{\mathrm{d}}\,\boldsymbol{Y}_t\right]$$

$$- \tfrac{1}{\sigma^2}\mathbb{E}\left[\int_0^T(\sigma^2\nabla\ln\pi_t\cdot\nabla\phi_t)(\boldsymbol{Y}_t)\,\mathrm{d}t\right]$$

$$=\mathbb{E}\left[\ln\pi_0(\boldsymbol{Y}_0) - \ln\pi_T(\boldsymbol{Y}_T)\right]$$

$$+ \mathbb{E}\left[\sigma^2\int_0^T|\nabla\ln\pi_t(\boldsymbol{Y}_t)|^2\mathrm{d}t + \tfrac{1}{\sigma\sqrt{2}}\int_0^T\left(\sigma^2\nabla\ln\pi_t - \nabla\phi_t\right)(\boldsymbol{Y}_t)\cdot\overleftarrow{\mathrm{d}}\,\boldsymbol{W}_t\right],$$

where in the last equality we have inserted the dynamics (21) and used the martingale property (30a). Notice that the expectation of the backward integral is not zero, see Appendix A.

## D.2    DERIVATION OF $\mathcal{L}_{\mathrm{Var}}^{\mathrm{CMCD}}$

In this section, we first verify the expression for $\mathcal{L}_{\mathrm{Var}}^{\mathrm{CMCD}}$ in Section 3.2, using Proposition 2.2, and choosing $\Gamma_0 = \Gamma_T$ to be the Lebesgue measure, $\gamma^+ = \gamma^- = 0$. We recall that although the Lebesgue measure in not normalisable, the arguments can be made rigorous using the techniques in Léonard (2014a, Appendix A).

The Radon-Nikodym derivative (RND) along (21) reads

$$
\left( \ln \frac{\mathrm{d}\,\overrightarrow{\mathbb{P}}^{\,\pi_0,\sigma^2\nabla\ln\pi+\nabla\phi}}{\mathrm{d}\,\overleftarrow{\mathbb{P}}^{\,\pi_T,-\sigma^2\nabla\ln\pi+\nabla\phi}} \right)(\boldsymbol{Y}) = (\ln\pi_0)(\boldsymbol{Y}_0) - (\ln\pi_T)(\boldsymbol{Y}_T)
$$

$$
+ \tfrac{1}{2\sigma^2}\int_0^T (\sigma^2\nabla\ln\pi_t+\nabla\phi_t)(\boldsymbol{Y}_t)\Big((\sigma^2\nabla\ln\pi_t+\nabla\phi_t)(\boldsymbol{Y}_t)\,\mathrm{d}t + \sqrt{2}\sigma\,\overrightarrow{\mathrm{d}}\,\boldsymbol{W}_t - \tfrac{1}{2}(\sigma^2\nabla\ln\pi_t+\nabla\phi_t)(\boldsymbol{Y}_t)\,\mathrm{d}t\Big)
$$

$$
- \tfrac{1}{2\sigma^2}\int_0^T (-\sigma^2\nabla\ln\pi_t+\nabla\phi_t)(\boldsymbol{Y}_t)\Big((\sigma^2\nabla\ln\pi_t+\nabla\phi_t)(\boldsymbol{Y}_t)\,\mathrm{d}t + \sqrt{2}\sigma\,\overleftarrow{\mathrm{d}}\,\boldsymbol{W}_t - \tfrac{1}{2}(\nabla\phi_t-\sigma^2\nabla\ln\pi_t)(\boldsymbol{Y}_t)\,\mathrm{d}t\Big)
$$

$$
= (\ln\pi_0)(\boldsymbol{Y}_0) - (\ln\pi_T)(\boldsymbol{Y}_T) + \sigma^2\int_0^T |\nabla\ln\pi_t(\boldsymbol{Y}_t)|^2\,\mathrm{d}t
$$

$$
+ \tfrac{\sigma}{\sqrt{2}}\left(\int_0^T \nabla\ln\pi_t(\boldsymbol{Y}_t)\cdot\overrightarrow{\mathrm{d}}\,\boldsymbol{W}_t + \int_0^T \nabla\ln\pi_t(\boldsymbol{Y}_t)\cdot\overleftarrow{\mathrm{d}}\,\boldsymbol{W}_t\right)
$$

$$
+ \tfrac{1}{\sigma\sqrt{2}}\left(\int_0^T \nabla\phi(\boldsymbol{Y}_t)\cdot\overrightarrow{\mathrm{d}}\,\boldsymbol{W}_t - \int_0^T \nabla\phi(\boldsymbol{Y}_t)\cdot\overleftarrow{\mathrm{d}}\,\boldsymbol{W}_t\right).
$$

Using (29b), we obtain

$$
\tfrac{\sigma}{\sqrt{2}}\left(\int_0^T \nabla\ln\pi_t(\boldsymbol{Y}_t)\cdot\overrightarrow{\mathrm{d}}\,\boldsymbol{W}_t + \int_0^T \nabla\ln\pi_t(\boldsymbol{Y}_t)\cdot\overleftarrow{\mathrm{d}}\,\boldsymbol{W}_t\right) = \sqrt{2}\sigma\int_0^T \nabla\ln\pi_t(\boldsymbol{Y}_t)\circ\mathrm{d}\boldsymbol{W}_t.
$$

Furthermore, from (15) we see that

$$
\tfrac{1}{\sigma\sqrt{2}}\left(\int_0^T \nabla\phi(\boldsymbol{Y}_t)\cdot\overrightarrow{\mathrm{d}}\,\boldsymbol{W}_t - \int_0^T \nabla\phi(\boldsymbol{Y}_t)\cdot\overleftarrow{\mathrm{d}}\,\boldsymbol{W}_t\right) = -\int_0^T \Delta\phi_t(\boldsymbol{Y}_t)\,\mathrm{d}t, \tag{47}
$$

from which the claim follows.

**Remark 8** (Estimating $\mathcal{L}_{\mathrm{Var}}^{\mathrm{CMCD}}$ without second derivatives). Using (47), we can equivalently write the RND as

$$
\left( \ln \frac{\mathrm{d}\,\overrightarrow{\mathbb{P}}^{\,\pi_0,\sigma^2\nabla\ln\pi+\nabla\phi}}{\mathrm{d}\,\overleftarrow{\mathbb{P}}^{\,\pi_T,-\sigma^2\nabla\ln\pi+\nabla\phi}} \right)(\boldsymbol{Y}) = \ln\pi_T(\boldsymbol{Y}_T) - \ln\pi_0(\boldsymbol{Y}_0)
$$

$$
- \tfrac{1}{\sigma\sqrt{2}}\left(\int_0^T \nabla\phi(\boldsymbol{Y}_t)\cdot\overrightarrow{\mathrm{d}}\,\boldsymbol{W}_t - \int_0^T \nabla\phi(\boldsymbol{Y}_t)\cdot\overleftarrow{\mathrm{d}}\,\boldsymbol{W}_t\right)
$$

$$
- \sigma\sqrt{2}\int_0^T \nabla\ln\pi_t(\boldsymbol{Y}_t)\circ\mathrm{d}\boldsymbol{W}_t - \sigma^2\int_0^T |\nabla\ln\pi_t(\boldsymbol{Y}_t)|^2\,\mathrm{d}t,
$$

so that $\mathcal{L}_{\mathrm{Var}}^{\mathrm{CMCD}}$ can be estimated without the need to evaluate $\Delta\phi$. Note that the identity (47) is similar to a finite difference approximation of $\Delta\phi$ along the process $\boldsymbol{Y}_t$.

### D.3 Existence and uniqueness of the drift

Before proceeding to the proof of Propostion 3.2, we state the following assumption on the curve of distributions $(\pi_t)_{t\in[0,T]}$:

**Assumption D.1.** Assume that $\pi \in C^\infty([0,T]\times\mathbb{R}^d;\mathbb{R})$, and that for all $t\in[0,T]$

1. the time derivative $\partial_t\pi_t$ is square-integrable, that is, $\partial_t\pi_t(t,\cdot)\in L^2(\mathbb{R}^d)$,

2. $\pi_t$ satisfies a Poincaré inequality, that is, there exists a constant $C_t > 0$ such that

$$
\mathrm{Var}_{\pi_t}(f) \le C_t \int_{\mathbb{R}^d} |\nabla f|^2\mathrm{d}\pi_t, \tag{49}
$$

for all $f\in C_b^1(\mathbb{R}^d)$.

Note that at the boundary $\partial[0,T] = \{0,T\}$, we agree to denote by $\partial_t \pi_t$ the 'inward-pointing derivative' and interpret $C^\infty([0,T] \times \mathbb{R}^d; \mathbb{R})$ in that way. We remark that the Poincaré inequality (49) is satisfied under relatively mild conditions on the tails of $\pi_t$ (for instance, Gaussian tails) and control of its derivatives, see, e.g., Bakry et al. (2014, Chapter 4). Under Assumption D.1, we can prove Proposition 3.2 as follows:

*Proof of Proposition 3.2.* The Fokker-Planck equation associated to (21) is given by

$$\partial_t \pi_t + \nabla \cdot (\pi_t \nabla \phi_t) = 0. \tag{50}$$

The operator $\phi \mapsto -\nabla \cdot (\pi_t \nabla \phi)$ is essentially self-adjoint in $L^2(\mathbb{R}^d)$, and, by (49) coercive on $L_0^2(\mathbb{R}^d) := \{f \in L^2(\mathbb{R}^d) : \int f \mathrm{d}x = 0\}$. Therefore, there exists a unique solution $\phi_t^* \in L_0^2(\mathbb{R}^d)$ to (50), for any $t \in [0,T]$. This solution is smooth by elliptic regularity. By Proposition 2.1 and our general framework, any minimiser $\widetilde{\phi}$ of (22) necessarily satisfies (50) as well. We then obtain

$$\nabla \cdot (\pi_t \nabla (\phi_t - \widetilde{\phi}_t)) = 0.$$

Multiplying this equation by $\phi_t - \widetilde{\phi}_t$, integrating, and integrating by parts shows that $\int \|\nabla(\phi - \widetilde{\phi})\|^2 \, \mathrm{d}\pi_t = 0$, proving the claim. □

**Remark 9** (Relation to previous work). Note we can carry out a change of variables to equation (50),

$$\partial_t \ln \pi_t = -\pi_t^{-1} (\nabla \pi_t \cdot \nabla \phi_t + \pi_t \Delta \phi) = -\nabla \ln \pi_t \cdot \nabla \phi - \Delta \phi,$$

yielding the PDE

$$\partial_t \ln \pi_t + \nabla \ln \pi_t \cdot \nabla \phi + \Delta \phi = 0,$$

which when considered in terms of the unnormalised flow $\hat{\pi}_t = Z_t \pi_t$ coincides with PDE in Vaikuntanathan & Jarzynski (2008); Arbel et al. (2021):

$$\partial_t \ln \hat{\pi}_t + \nabla \ln \hat{\pi}_t \cdot \nabla \phi + \Delta \phi - \mathbb{E}_{\pi_t}[\partial_t \ln \hat{\pi}_t] = 0.$$

In particular, we note that the Markov chain proposed in Arbel et al. (2021) converges to our proposed parameterisation in equation (21) (see equation (12) in Arbel et al. (2021)).

### D.4 Infinitesimal Schrödinger bridges (proof of Proposition 3.4)

Throughout this proof, we assume that the Schrödinger problems on the intervals $[iT/N, (i+1)T/N]$, $i = 0, \ldots, N-1$ admit unique solutions, with drifts of regularity specified in Assumption A.1, see (Léonard, 2014a, Proposition 2.5) for sufficient conditions. We also work under Assumption D.1, so that the drift $\nabla \phi^*$ exists and is unique by Proposition 3.2.

Given the interpolation $(\pi_t)_{t \in [0,T]}$, we define the constraint sets

$$\mathcal{M}^N(\pi) := \left\{ a \in \mathcal{U}^N : \quad \overrightarrow{\mathbb{P}}_{t_i}^{\pi_0, \nabla \ln \pi + a} = \pi_{t_i} \quad \text{at times} \quad t_i = \frac{iT}{N}, \quad i = 0, \ldots, N \right\}, \tag{51}$$

as well as

$$\mathcal{M}^\infty(\pi) := \left\{ a \in \mathcal{U} : \quad \overrightarrow{\mathbb{P}}_t^{\pi_0, \nabla \ln \pi + a} = \pi_t \quad \text{for all} \quad t \in [0,T] \right\}. \tag{52a}$$

In (51), the set $\mathcal{U}^N$ is given by

$$\mathcal{U}^N := \left\{ a \in C([0,T] \times \mathbb{R}^d; \mathbb{R}^d) : \quad a \in C^\infty([\tfrac{iT}{N}, \tfrac{(i+1)T}{N}] \times \mathbb{R}^d; \mathbb{R}^d), \quad \text{for all } i = 0, \ldots, N-1, \right.$$

$$\left. \exists L > 0 \text{ such that } \|a_t(\boldsymbol{x}) - a_t(\boldsymbol{y})\| \leq L \|\boldsymbol{x} - \boldsymbol{y}\|, \text{ for all } t \in [0,T], \ \boldsymbol{x}, \boldsymbol{y} \in \mathbb{R}^d \right\},$$

and we recall that $\mathcal{U}$ has been defined in Assumption A.1.

---

**Algorithm 2** Controlled Monte Carlo Diffusions - Training

---

***Require:*** $\pi_0$, $\pi_T$, $\pi_t$, $\sigma$, *K step-sizes* $\Delta t_k$, *network* $f^\phi$
   **for** $i$ in epochs **do**
      $\ln \boldsymbol{W}_T, \boldsymbol{Y}_T \sim$ ***Algorithm 1****($\pi_0$, $\pi_T$, $\pi_t$, $\sigma$, $\{\Delta t_k\}_k$, $f^\phi$)*
      *Gradient descent step* $\nabla_\phi - \ln \boldsymbol{W}_T$
   **return** $f^\phi$

---

By the construction in Proposition 3.4, the drift $\nabla \phi^{(N)}$ can be characterised by

$$\nabla \phi^{(N)} \in \underset{a \in \mathcal{M}^N(\pi)}{\arg\min} \, \mathbb{E}_{\boldsymbol{Y} \sim \overrightarrow{\mathbb{P}}^{\pi_0, \nabla \ln \pi + a}} \left[ \frac{1}{2\sigma^2} \int_0^T \|a_t(\boldsymbol{Y})\|^2 \, \mathrm{d}t \right] \tag{54a}$$

$$= \underset{a \in \mathcal{M}^N(\pi)}{\arg\min} \, D_{\mathrm{KL}}(\overrightarrow{\mathbb{P}}^{\pi_0, \nabla \ln \pi + a} | \overrightarrow{\mathbb{P}}^{\pi_0, \nabla \ln \pi}), \tag{54b}$$

where the second line follows from Girsanov's theorem, see the proof of Proposition 2.2.

We now claim that the CMCD drift $\nabla \phi^*$, by definition the minimiser in (22), can be characterised in a similar way by

$$\nabla \phi^* \in \underset{a \in \mathcal{M}^\infty(\pi)}{\arg\min} \, \mathbb{E}_{\boldsymbol{Y} \sim \overrightarrow{\mathbb{P}}^{\pi_0, \nabla \ln \pi + a}} \left[ \frac{1}{2\sigma^2} \int_0^T \|a_t(\boldsymbol{Y})\|^2 \, \mathrm{d}t \right] \tag{55a}$$

$$= \underset{a \in \mathcal{M}^\infty(\pi)}{\arg\min} \, D_{\mathrm{KL}}(\overrightarrow{\mathbb{P}}^{\pi_0, \nabla \ln \pi + a} | \overrightarrow{\mathbb{P}}^{\pi_0, \nabla \ln \pi}). \tag{55b}$$

Indeed, the constraint $\overrightarrow{\mathbb{P}}_t^{\pi_0, \nabla \ln \pi + a} = \pi_t$ for all $t \in [0, T]$ implies that $a$ satisfies the Fokker-Planck equation $\partial_t \pi_t + \nabla \cdot (\pi_t a_t) = 0$. By the Helmholtz decomposition (Figalli & Glaudo, 2021, Section 2.5.4), minimisers of $a_t \mapsto \int a_t^2 \mathrm{d}\pi_t$ are of gradient form, thus (55a) holds.

Comparing (54) and (55a), it is plausible to infer the convergence $\nabla \phi^{(N)} \to \nabla \phi^*$, as the marginal constraints at the discrete time points $0, 1/T, 2/T, ..., T$ become dense and approach the continuous-time constraint in (52).

To make this more precise, we note that since $\mathcal{M}^\infty(\pi) \subset \mathcal{M}^N(\pi)$ for all $N \in \mathbb{N}$, we have that

$$D_{\mathrm{KL}}(\overrightarrow{\mathbb{P}}^{\pi_0, \nabla \ln \pi + \nabla \phi^{(N)}} | \overrightarrow{\mathbb{P}}^{\pi_0, \nabla \ln \pi}) \leq D_{\mathrm{KL}}(\overrightarrow{\mathbb{P}}^{\pi_0, \nabla \ln \pi + \nabla \phi^*} | \overrightarrow{\mathbb{P}}^{\pi_0, \nabla \ln \pi}), \tag{56}$$

for all $N \in \mathbb{N}$. Since $D_{\mathrm{KL}}(\cdot | \overrightarrow{\mathbb{P}}^{\pi_0, \nabla \ln \pi})$ has weakly compact sublevel sets (Dupuis & Ellis, 2011, Lemma 1.4.3), we can extract a subsequence $\overrightarrow{\mathbb{P}}^{\pi_0, \nabla \ln \pi + \nabla \phi^{(N_k)}}$ that converges weakly towards a path measure $\widetilde{\mathbb{P}} \in \mathcal{P}(C([0, T]; \mathbb{R}^d))$. To show that indeed $\widetilde{\mathbb{P}} = \overrightarrow{\mathbb{P}}^{\pi_0, \nabla \ln \pi + \nabla \phi^*}$, it is sufficient to note that by the constraints in (51) those measures necessarily have the same finite-dimensional marginals, and to combine this observation with the continuity statement of Theorem 2.7.3 in Billingsley (2013), as well as the uniqueness from Proposition 3.2. The convergence of the drifts in the sense of $L^2([0, T] \times \mathbb{R}^d; \mathbb{R}^d)$ now follows from the lower semicontinuity of $D_{\mathrm{KL}}$ in combination with Girsanov's theorem.

## D.5   DISCRETISATION AND OBJECTIVE

In the setting of CMCD with KL divergence we can use the EM approximations to the RND presented in Proposition E.1 to express the objective as:

$$\mathcal{L}_{D_{\mathrm{KL}}}^{\mathrm{CMCD}}(\phi) \approx \mathbb{E}\left[ \ln \frac{\pi_0(\boldsymbol{Y}_0)}{\hat{\pi}(\boldsymbol{Y}_T)} \prod_{k=0}^{K-1} \frac{\mathcal{N}(\boldsymbol{Y}_{t_{k+1}} | \boldsymbol{Y}_{t_k} + (\sigma^2 \nabla \ln \pi_{t_k} + \nabla \ln \phi_{t_k})(\boldsymbol{Y}_{t_k}) \Delta t_k, 2\sigma^2 \Delta t_k)}{\mathcal{N}(\boldsymbol{Y}_{t_k} | \boldsymbol{Y}_{t_{k+1}} + (\sigma^2 \nabla \ln \pi_{t_{k+1}} - \nabla \ln \phi_{t_{k+1}})(\boldsymbol{Y}_{t_{k+1}}) \Delta t_k, 2\sigma^2 \Delta t_k)} \right], \tag{57}$$

where the expectation is taken wrt to the EM approximation of the SDE in (21), that is:

$$\boldsymbol{Y}_{t_{k+1}} \sim \mathcal{N}(\boldsymbol{Y}_{t_k} + (\sigma^2 \nabla \ln \pi_{t_k} + \nabla \ln \phi_{t_k})(\boldsymbol{Y}_{t_k}) \Delta t_k, \, 2\sigma^2 \Delta t_k). \tag{58}$$

---

**Algorithm 3** Forward transition $F_{t_k}(\boldsymbol{Z}_{t_{k+1}}, \boldsymbol{Y}_{t_{k+1}} | \boldsymbol{Z}_{t_k}, \boldsymbol{Y}_{t_k})$

---

**Require:** $\boldsymbol{Z}_{t_k}, \boldsymbol{Y}_{t_k}$, step-size $\Delta t_k$

Re-sample momentum $\boldsymbol{Y}'_{t_k} \sim \mathcal{N}\left(\boldsymbol{Y}_{t_k}(1 - \sigma \Delta t_k) + \nabla \phi_{t_k}(\boldsymbol{Y}_{t_k}, \boldsymbol{Z}_{t_k})\Delta t_k, \ 2\sigma \Delta t_k I\right)$

$\left.\begin{array}{l}\text{Update } \boldsymbol{Y}''_{t_k} = \boldsymbol{Y}'_{t_k} + \frac{\Delta t_k}{2}\nabla \ln \pi_{t_k}(\boldsymbol{Z}_{t_k}) \\[4pt] \text{Update } \boldsymbol{Z}_{t_{k+1}} = \boldsymbol{Z}_{t_k} + \Delta t_k \boldsymbol{Y}''_{t_k} \\[4pt] \text{Update } \boldsymbol{Y}_{t_{k+1}} = \boldsymbol{Y}''_{t_k} + \frac{\Delta t_k}{2}\nabla \ln \pi_{t_k}(\boldsymbol{Z}_{t_{k+1}})\end{array}\right\}$ Leapfrog step $\Phi(\boldsymbol{Z}_{t_k}, \boldsymbol{Y}'_{t_k})$

**return** $(\boldsymbol{Z}_{t_{k+1}}, \boldsymbol{Y}_{t_{k+1}})$

---

**Algorithm 4** Backward transition $B_{t_k}(\boldsymbol{Z}_{t_k}, \boldsymbol{Y}_{t_k} | \boldsymbol{Z}_{t_{k+1}}, \boldsymbol{Y}_{t_{k+1}})$

---

**Require:** $\boldsymbol{Z}_{t_{k+1}}, \boldsymbol{Y}_{t_{k+1}}$, step-size $\Delta t_k$

$\left.\begin{array}{l}\text{Update } \boldsymbol{Y}''_{t_k} = \boldsymbol{Y}_{t_{k+1}} - \frac{\Delta t_k}{2}\nabla \ln \pi_{t_k}(\boldsymbol{Z}_{t_k}) \\[4pt] \text{Update } \boldsymbol{Z}_{t_k} = \boldsymbol{Z}_{t_{k+1}} - \Delta t_k \boldsymbol{Y}''_{t_k} \\[4pt] \text{Update } \boldsymbol{Y}'_{t_k} = \boldsymbol{Y}''_{t_k} - \frac{\Delta t_k}{2}\nabla \ln \pi_{t_k}(\boldsymbol{Z}_{t_{k+1}})\end{array}\right\}$ Inverse leapfrog $\Phi^{-1}(\boldsymbol{Z}_{t_{k+1}}, \boldsymbol{Y}_{t_{k+1}})$

Re-sample momentum $\boldsymbol{Y}_{t_k} \sim \mathcal{N}\left(\boldsymbol{Y}'_{t_k}(1 - \sigma \Delta t_k) - \nabla \phi_{t_k}(\boldsymbol{Y}'_{t_k}, \boldsymbol{Z}_{t_k})\Delta t_k, \ 2\sigma \Delta t_k I\right)$

**return** $(\boldsymbol{Z}_{t_k}, \boldsymbol{Y}_{t_k})$

---

### D.6 UNDERDAMPED LANGEVIN DYNAMICS

In this section, we motivate the underdamped generalisation of CMCD which is used across our experiments. This parameterisation is inspired by the underlying theory for the overdamped approach, and we leave a rigorous extension of those foundations for future work. However, we have found this heuristic parameterisation to perform very well empirically.

Following Geffner & Domke (2023) we parametrise as:

$$\begin{aligned} \boldsymbol{Y}_0, \boldsymbol{Z}_0 &\sim \mathcal{N}(0, I) \otimes \pi_0, \\ \mathrm{d}\boldsymbol{Z}_t &= \boldsymbol{Y}_t \mathrm{d}t, \\ \mathrm{d}\boldsymbol{Y}_t &= \left(\sigma^2 \nabla \ln \pi_t(\boldsymbol{Z}_t) - \sigma^2 \boldsymbol{Y}_t + \nabla \phi_t(\boldsymbol{Y}_t, \boldsymbol{Z}_t)\right)\mathrm{d}t + \sigma\sqrt{2}\,\overrightarrow{\mathrm{d}}\,\boldsymbol{W}_t. \end{aligned} \tag{59}$$

and it's time reversal as:

$$\begin{aligned} \boldsymbol{Y}_T, \boldsymbol{Z}_T &\sim \mathcal{N}(0, I) \otimes \pi_T, \\ \mathrm{d}\boldsymbol{Z}_t &= \boldsymbol{Y}_t \mathrm{d}t, \\ \mathrm{d}\boldsymbol{Y}_t &= \left(-\sigma^2 \nabla \ln \pi_t(\boldsymbol{Z}_t) + \sigma^2 \boldsymbol{Y}_t + \nabla \phi_t(\boldsymbol{Y}_t, \boldsymbol{Z}_t)\right)\mathrm{d}t + \sigma\sqrt{2}\,\overleftarrow{\mathrm{d}}\,\boldsymbol{W}_t. \end{aligned} \tag{60}$$

#### D.6.1 TIME DISCRTEISATION AND OBJECTIVE

To discretise the above processes we follow the exact same discretisation scheme carried out in Geffner & Domke (2023), however in this case we have to adapt the forward discretisation scheme to include the non-linear drift when carrying out the momentum re-sample step, specific details for this scheme can be found in Algorithms 3 and 4. This discretisation in turn allows us to compute the discrete RND between these two processes which we require for Framework 1'.

Now via Propostion 1 in (Geffner & Domke, 2023) it follows that

$$\begin{aligned} &\frac{\pi_0(\boldsymbol{Y}_0, \boldsymbol{Z}_0)}{\pi_T(\boldsymbol{Y}_T, \boldsymbol{Z}_T)} \prod_{k=0}^{K-1} \frac{F_{t_k}(\boldsymbol{Z}_{t_{k+1}}, \boldsymbol{Y}_{t_{k+1}} | \boldsymbol{Z}_{t_k}, \boldsymbol{Y}_{t_k})}{B_{t_k}(\boldsymbol{Z}_{t_k}, \boldsymbol{Y}_{t_k} | \boldsymbol{Z}_{t_{k+1}}, \boldsymbol{Y}_{t_{k+1}})} \\ &= \frac{\pi_0(\boldsymbol{Y}_0, \boldsymbol{Z}_0)}{\pi_T(\boldsymbol{Y}_T, \boldsymbol{Z}_T)} \prod_{k=0}^{K-1} \frac{\mathcal{N}\left(\boldsymbol{Y}'_{t_k} \mid \boldsymbol{Y}_{t_k}(1 - \sigma \Delta t_k) + \nabla \phi_{t_k}(\boldsymbol{Y}_{t_k}, \boldsymbol{Z}_{t_k})\Delta t_k, 2\sigma \Delta t_k I\right)}{\mathcal{N}\left(\boldsymbol{Y}_{t_k} \mid \boldsymbol{Y}'_{t_k}(1 - \sigma \Delta t_k) - \nabla \phi_{t_k}(\boldsymbol{Y}'_{t_k}, \boldsymbol{Z}_{t_k})\Delta t_k, 2\sigma \Delta t_k I\right)} \end{aligned} \tag{61}$$

then we can use the above discrete time RND to approximate the KL divergence between SDEs (59) and (60) yielding our objective for the under dampened setting:

$$\mathcal{L}_{D_{\mathrm{KL}}}^{\mathrm{CMCD-UD}}(\phi) \approx \mathbb{E}\left[\ln \frac{\pi_0(\boldsymbol{Y}_0, \boldsymbol{Z}_0)}{\pi_T(\boldsymbol{Y}_T, \boldsymbol{Z}_T)}\prod_{k=0}^{K-1}\frac{\mathcal{N}\left(\boldsymbol{Y}_{t_k}'|\boldsymbol{Y}_{t_k}(1-\sigma\Delta t_k)+\nabla\phi_{t_k}(\boldsymbol{Y}_{t_k},\boldsymbol{Z}_{t_k})\Delta t_k, 2\sigma\Delta t_k I\right)}{\mathcal{N}\left(\boldsymbol{Y}_{t_k}|\boldsymbol{Y}_{t_k}'(1-\sigma\Delta t_k)-\nabla\phi_{t_k}(\boldsymbol{Y}_{t_k}',\boldsymbol{Z}_{t_k})\Delta t_k, 2\sigma\Delta t_k I\right)}\right]$$
(62)

Where the expectation is taken with respect to the discrete-time process in Algorithm 3.

## E  PROOFS

### E.1  PROOF OF PROPOSITION 2.2 (FORWARD-BACKWARD RADON-NIKODYM DERIVATIVES)

*Proof.* We begin with the forward Radon-Nikodym derivative

$$\ln\left(\frac{\mathrm{d}\overrightarrow{\mathbb{P}}^{\mu,a}}{\mathrm{d}\overrightarrow{\mathbb{P}}^{\nu,b}}\right)(\boldsymbol{Y}) = \ln\left(\frac{\mathrm{d}\mu}{\mathrm{d}\nu}\right)(\boldsymbol{Y}_0) + \frac{1}{\sigma^2}\int_0^T (a_t-b_t)(\boldsymbol{Y}_t)\cdot\overrightarrow{\mathrm{d}}\,\boldsymbol{Y}_t + \frac{1}{2\sigma^2}\int_0^T\left(b_t^2-a_t^2\right)(\boldsymbol{Y}_t)\,\mathrm{d}t,$$
(63)

following from Girsanov's theorem (see, for instance, Nüsken & Richter (2021, Lemma A.1) and substitute $\sigma u = a - b$). To compute the backward Radon-Nikodym derivative, we temporarily introduce the time-reversal operator $\mathcal{R}$, acting as $(\mathcal{R}\boldsymbol{Y})_t := \boldsymbol{Y}_{T-t}$ on paths[11], and as $(\mathcal{R}a)_t(\boldsymbol{y}) := a_{T-t}(\boldsymbol{y})$ on vector fields. We then observe that

$$\ln\left(\frac{\mathrm{d}\overleftarrow{\mathbb{P}}^{\mu,\mathcal{R}a}}{\mathrm{d}\overleftarrow{\mathbb{P}}^{\nu,\mathcal{R}b}}\right)(\mathcal{R}\boldsymbol{Y}) = \ln\left(\frac{\mathrm{d}\overrightarrow{\mathbb{P}}^{\mu,a}}{\mathrm{d}\overrightarrow{\mathbb{P}}^{\nu,b}}\right)(\boldsymbol{Y}),$$

for instance by comparing the discrete-time processes in (9a) and (9b). Equivalently,

$$\ln\left(\frac{\mathrm{d}\overleftarrow{\mathbb{P}}^{\mu,a}}{\mathrm{d}\overleftarrow{\mathbb{P}}^{\nu,b}}\right)(\boldsymbol{Y}) = \ln\left(\frac{\mathrm{d}\overrightarrow{\mathbb{P}}^{\mu,\mathcal{R}a}}{\mathrm{d}\overrightarrow{\mathbb{P}}^{\nu,\mathcal{R}b}}\right)(\mathcal{R}\boldsymbol{Y}),$$

since $\mathcal{R}^2$ is the identity. Building on (63), the backward Radon-Nikodym derivative therefore reads

$$\ln\left(\frac{\mathrm{d}\overleftarrow{\mathbb{P}}^{\mu,a}}{\mathrm{d}\overleftarrow{\mathbb{P}}^{\nu,b}}\right)(\boldsymbol{Y}) = \ln\left(\frac{\mathrm{d}\mu}{\mathrm{d}\nu}\right)((\mathcal{R}\boldsymbol{Y})_0) + \frac{1}{\sigma^2}\int_0^T((\mathcal{R}a)_t-(\mathcal{R}b)_t)(\mathcal{R}\boldsymbol{Y}_t)\cdot\overrightarrow{\mathrm{d}}(\mathcal{R}\boldsymbol{Y})_t$$

$$+ \frac{1}{2\sigma^2}\int_0^T\left((\mathcal{R}b)_t^2-(\mathcal{R}a)_t^2\right)((\mathcal{R}\boldsymbol{Y})_t)\,\mathrm{d}t,$$

$$= \ln\left(\frac{\mathrm{d}\mu}{\mathrm{d}\nu}\right)(\boldsymbol{Y}_T) + \frac{1}{\sigma^2}\int_0^T(a_t-b_t)(\boldsymbol{Y}_t)\cdot\overleftarrow{\mathrm{d}}\,\boldsymbol{Y}_t + \frac{1}{2\sigma^2}\int_0^T\left(b_t^2-a_t^2\right)(\boldsymbol{Y}_t)\,\mathrm{d}t,$$
(64)

where the integrals have been transformed using the substitution $t \mapsto T - t$. The result in (14) now follows by writing

$$\ln\left(\frac{\mathrm{d}\overrightarrow{\mathbb{P}}^{\mu,a}}{\mathrm{d}\overleftarrow{\mathbb{P}}^{\nu,b}}\right)(\boldsymbol{Y}) = \ln\left(\frac{\mathrm{d}\overrightarrow{\mathbb{P}}^{\mu,a}}{\mathrm{d}\overrightarrow{\mathbb{P}}^{\Gamma_0,\gamma^+}}\right)(\boldsymbol{Y}) + \ln\left(\frac{\mathrm{d}\overleftarrow{\mathbb{P}}^{\Gamma_T,\gamma^-}}{\mathrm{d}\overleftarrow{\mathbb{P}}^{\nu,b}}\right)(\boldsymbol{Y}),$$

using the assumption $\overrightarrow{\mathbb{P}}^{\Gamma_0,\gamma^+} = \overleftarrow{\mathbb{P}}^{\Gamma_T,\gamma^-}$, and inserting (63) as well as (64). $\square$

### E.1.1  DISCRETISATION AND CONNECTION TO DNFs (DIFFUSION NORMALISING FLOWS)

In this section we derive the main discretisation formula used in our implementations for the forward-backwards Radon-Nikodym derivative (RND).

---

[11] Although pathwise definitions should be treated with care (because Itô integrals are defined only up to a set of measure zero), the arguments can be made rigorous using the machinery referred to in Appendix A.

**Proposition E.1.** *Letting $\Gamma_0 = \Gamma_T = \mathrm{Leb}$ and $\gamma^\pm = 0$, we have that the RND in (14) is given by*

$$\ln\left(\frac{\mathrm{d}\overrightarrow{\mathbb{P}}^{\mu,a}}{\mathrm{d}\overleftarrow{\mathbb{P}}^{\nu,b}}\right)(\boldsymbol{Y}) = \ln\mu(\boldsymbol{Y}_0) - \ln\nu(\boldsymbol{Y}_T) + \frac{1}{\sigma^2}\int_0^T a_t(\boldsymbol{Y}_t)\cdot\overrightarrow{\mathrm{d}}\,\boldsymbol{Y}_t - \frac{1}{2\sigma^2}\int_0^T ||a_t(\boldsymbol{Y}_t)||^2\,\mathrm{d}t$$

$$- \frac{1}{\sigma^2}\int_0^T b_t(\boldsymbol{Y}_t)\cdot\overleftarrow{\mathrm{d}}\,\boldsymbol{Y}_t + \frac{1}{2\sigma^2}\int_0^T ||b_t(\boldsymbol{Y}_t)|||^2\,\mathrm{d}t, \qquad \overrightarrow{\mathbb{P}}^{\mu,a}\text{-almost surely,}$$

*and admits the following discrete-time approximation up to constant terms in $a_t$ and $b_t$ (following Remark 3),*

$$\ln\left(\frac{\widehat{\mathrm{d}\overrightarrow{\mathbb{P}}^{\mu,a}}}{\mathrm{d}\overleftarrow{\mathbb{P}}^{\nu,b}}\right)(\boldsymbol{Y}) = -\ln\nu(\boldsymbol{Y}_T) + \sum_{i=0}^{K-1}\frac{1}{2\sigma^2(t_{i+1}-t_{i+1})}||\boldsymbol{Y}_{t_i}-\boldsymbol{Y}_{t_{i+1}}+b_{t_{i+1}}(\boldsymbol{Y}_{t_{i+1}})(t_{i+1}-t_{i+1})||^2 + \mathrm{const},$$

*when using the Euler-Maruyama discretisation:*

$$\boldsymbol{Y}_{t_{i+1}} = \boldsymbol{Y}_{t_i} + a_{t_i}(\boldsymbol{Y}_{t_i})(t_{i+1}-t_i) + \sqrt{(t_{i+1}-t_i)}\sigma\xi, \qquad \xi\sim\mathcal{N}(0,I).$$

*Proof.* The first part follows by direct computation.

From here on, we will use the notation $f_{t_i} = f_{t_i}(\boldsymbol{Y}_{t_i})$ for brevity. Following Remark 3 we have that

$$\ln\left(\frac{\mathrm{d}\overrightarrow{\mathbb{P}}^{\mu,a}}{\mathrm{d}\overleftarrow{\mathbb{P}}^{\nu,b}}\right)(\boldsymbol{Y}) \approx \ln\mu(\boldsymbol{Y}_0) - \ln\nu(\boldsymbol{Y}_T)$$

$$+ \frac{1}{\sigma^2}\sum_{i=0}^{K-1}a_{t_i}\cdot(\boldsymbol{Y}_{t_{i+1}}-\boldsymbol{Y}_{t_i}) - \frac{1}{2\sigma^2}\sum_{i=0}^{K-1}||a_{t_i}||^2(t_{i+1}-t_i)$$

$$- \frac{1}{\sigma^2}\sum_{i=0}^{K-1}b_{t_{i+1}}\cdot(\boldsymbol{Y}_{t_{i+1}}-\boldsymbol{Y}_{t_i}) + \frac{1}{2\sigma^2}\sum_{i=0}^{K-1}||b_{t_{i+1}}|||^2(t_{i+1}-t_i).$$

Adding and subtracting $||\boldsymbol{Y}_{t_{i+1}} - \boldsymbol{Y}_{t_i}||^2/(\sigma^2(t_{i+1}-t_i))$ allows us to complete the square in each sum, resulting in:

$$\ln\left(\frac{\mathrm{d}\overrightarrow{\mathbb{P}}^{\mu,a}}{\mathrm{d}\overleftarrow{\mathbb{P}}^{\nu,b}}\right)(\boldsymbol{Y}) \approx \ln\mu(\boldsymbol{Y}_0) - \ln\nu(\boldsymbol{Y}_T) - \sum_{i=0}^{K-1}\frac{1}{2\sigma^2(t_{i+1}-t_i)}||\boldsymbol{Y}_{t_{i+1}}-\boldsymbol{Y}_{t_i}-a_{t_i}(t_{i+1}-t_i)||^2$$

$$+ \sum_{i=0}^{K-1}\frac{1}{2\sigma^2(t_{i+1}-t_i)}||\boldsymbol{Y}_{t_i}-\boldsymbol{Y}_{t_{i+1}}+b_{t_{i+1}}(t_{i+1}-t_i)||^2. \tag{67}$$

Now notice that under the Euler-Maruyama discretisation $||\boldsymbol{Y}_{t_{i+1}} - \boldsymbol{Y}_{t_i} - a_{t_i}(t_{i+1}-t_i)||^2 = (t_{i+1}-t_i)\sigma^2||\xi||^2$ where $\xi\sim\mathcal{N}(0,I)$ does not depend on $a_t$ or $b_t$; in particular when using $D_{\mathrm{KL}}$ for the divergence we have that $\mathbb{E}_{\overrightarrow{\mathbb{P}}_{\mathrm{EM}}^{\mu,a}}||\boldsymbol{Y}_{t_{i+1}} - \boldsymbol{Y}_{t_i} - a_{t_i}(t_{i+1}-t_i)||^2 = \sigma^2$ and thus:

$$\ln\left(\frac{\widehat{\mathrm{d}\overrightarrow{\mathbb{P}}^{\mu,a}}}{\mathrm{d}\overleftarrow{\mathbb{P}}^{\nu,b}}\right)(\boldsymbol{Y}) \propto \ln\mu(\boldsymbol{Y}_0) - \ln\nu(\boldsymbol{Y}_T) + \sum_{i=0}^{K-1}\frac{1}{2\sigma^2(t_{i+1}-t_i)}||\boldsymbol{Y}_{t_i}-\boldsymbol{Y}_{t_{i+1}}+b_{t_{i+1}}(t_{i+1}-t_i)||^2.$$

$$\tag{68}$$

$\square$

Notice that in expectation (for computing $D_{\mathrm{KL}}$), equation (68) matches equation (15) in Zhang & Chen (2021) and thus provides a theoretical backing to the objective used in Zhang & Chen (2021). Resolving the term $\mathbb{E}_{\overrightarrow{\mathbb{P}}_{\mathrm{EM}}^{\mu,a}}||\boldsymbol{Y}_{t_{i+1}} - \boldsymbol{Y}_{t_i} - a_{t_i}(t_{i+1}-t_i)||^2$ analytically may offer a variance reduction similar to the analytic calculations in Sohl-Dickstein et al. (2015, Equation 14) and the Rao-Blackwelizations of $D_{\mathrm{KL}}$ in Ho et al. (2020).

**Remark 10.** The time discretised RND in equation (67) can be expressed as the ratio of the transition densities corresponding to two discrete-time Markov chains $\mu(\boldsymbol{y}_0)q^a(\boldsymbol{y}_{1:K}|\boldsymbol{y}_0)/p^b(\boldsymbol{y}_{0:K-1}|\boldsymbol{y}_K)\nu(\boldsymbol{y}_K)$ with $\boldsymbol{y}_{0:K} \sim q^a(\boldsymbol{y}_{1:K}|\boldsymbol{y}_0)\mu(\boldsymbol{y}_0)$. As a result considering $\nu(x) = \hat{\nu}(\boldsymbol{x})/Z$ and the IS estimator $\hat{Z} = p^b(\boldsymbol{y}_{0:K-1}|\boldsymbol{y}_K)\hat{\nu}(\boldsymbol{y}_K)/\mu(\boldsymbol{y}_0)q^a(\boldsymbol{y}_{1:K}|\boldsymbol{y}_0)$ it follows that $\mathbb{E}_{q^a(\boldsymbol{y}_{1:K}|\boldsymbol{y}_0)\mu(\boldsymbol{y}_0)}[\ln\hat{Z}]$ is an ELBO of $\hat{Z}$ (e.g. $\mathbb{E}_{q^a(\boldsymbol{y}_{1:K}|\boldsymbol{y}_0)\mu(\boldsymbol{y}_0)}[\ln\hat{Z}] \leq \ln Z$).

Whilst superficially simple, Remark 10 guarantees that normalizing constant estimators arising from our discretisation do not overestimate the true normalizing constant. This result is beneficial in practice as it allows us to compare estimators possessing this property by selecting the one with the largest value. As highlighted in Vargas et al. (2023a) many SDE discretisations can result in estimators that do not yield an ELBO: for example, the estimators used in Berner et al. (2022) can result in overestimating the normalising constant. Note similar remarks have been established in the context of free energy computation and the Jarzynski equality see Stoltz et al. (2010, Remark 4.5).

### E.2 Proof of Proposition 3.3 (Controlled Crooks' fluctuation theorem and the Jarzinky equality)

*Proof.* Following the computations in Appendix D.1 and using the formulae (29), we compute

$$\ln\left(\frac{\mathrm{d}\overrightarrow{\mathbb{P}}^{\mu,\sigma^2\nabla\ln\pi+\nabla\phi}}{\mathrm{d}\overleftarrow{\mathbb{P}}^{\nu,-\sigma^2\nabla\ln\pi+\nabla\phi}}\right)(\boldsymbol{Y}) = \ln\mu(\boldsymbol{Y}_0) - \ln\nu(\boldsymbol{Y}_T)$$

$$+ \int_0^T \nabla\ln\pi_t(\boldsymbol{Y}_t)\circ\mathrm{d}\boldsymbol{Y}_t - \tfrac{1}{\sigma^2}\int_0^T \nabla\phi_t(\boldsymbol{Y}_t)\cdot\nabla\ln\pi_t(\boldsymbol{Y}_t)\,\mathrm{d}t - \int_0^T \Delta\phi(\boldsymbol{Y}_t)\,\mathrm{d}t.$$

Then via Itô's lemma applied to the unnormalised annealed log target $\ln\hat{\pi}_t = \ln\pi_t - \ln\mathcal{Z}_t$ we have

$$\ln\hat{\pi}_T(\boldsymbol{Y}_T) - \ln\hat{\pi}_0(\boldsymbol{Y}_0) - \int_0^T \partial_t\ln\hat{\pi}_t(\boldsymbol{Y}_t)\,\mathrm{d}t = \int_0^T \nabla\ln\hat{\pi}_t(\boldsymbol{Y}_t)\circ\mathrm{d}\boldsymbol{Y}_t = \int_0^T \nabla\ln\pi_t(\boldsymbol{Y}_t)\circ\mathrm{d}\boldsymbol{Y}_t,$$

thus we arrive at

$$\ln\left(\frac{\mathrm{d}\overrightarrow{\mathbb{P}}^{\mu,\sigma^2\nabla\ln\pi+\nabla\phi}}{\mathrm{d}\overleftarrow{\mathbb{P}}^{\nu,-\sigma^2\nabla\ln\pi+\nabla\phi}}\right)(\boldsymbol{Y}) = \ln\mu(\boldsymbol{Y}_0) - \ln\nu(\boldsymbol{Y}_T) + \ln\hat{\pi}_T(\boldsymbol{Y}_T) - \ln\hat{\pi}_0(\boldsymbol{Y}_0)$$

$$- \int_0^T \partial_t\ln\hat{\pi}_t(\boldsymbol{Y}_t)\,\mathrm{d}t - \tfrac{1}{\sigma^2}\int_0^T \nabla\phi_t(\boldsymbol{Y}_t)\cdot\nabla\ln\pi_t(\boldsymbol{Y}_t)\,\mathrm{d}t - \int_0^T \Delta\phi_t(\boldsymbol{Y}_t)\,\mathrm{d}t.,$$

for arbitrary initial and final densities $\mu$ and $\nu$. Crooks' generalised fluctuation theorem (Crooks, 1999) now follows from taking $\phi = 0$, and the controlled version in Proposition 3.3 follows from $\mu = \pi_0$ and $\nu = \pi_T$. Finally notice that:

$$1 = \mathbb{E}_{\overrightarrow{\mathbb{P}}^{\mu,\sigma^2\nabla\ln\pi}}\left[\left(\frac{\mathrm{d}\overrightarrow{\mathbb{P}}^{\mu,\sigma^2\nabla\ln\pi}}{\mathrm{d}\overleftarrow{\mathbb{P}}^{\nu,-\sigma^2\nabla\ln\pi}}\right)^{-1}\right]$$

$$= \mathbb{E}_{\overrightarrow{\mathbb{P}}^{\mu,\sigma^2\nabla\ln\pi}}\left[\exp\left(-\ln\mu(\boldsymbol{Y}_0) + \ln\nu(\boldsymbol{Y}_T) - \ln\hat{\pi}_T(\boldsymbol{Y}_T) + \ln\hat{\pi}_0(\boldsymbol{Y}_0) + \int_0^T \partial_t\ln\hat{\pi}_t(\boldsymbol{Y}_t)\,\mathrm{d}t\right)\right],$$

which implies the Jarzynski equality when considering the boundaries $\mu = \pi_0$ and $\nu = \pi_T$, resulting in:

$$\mathbb{E}_{\overrightarrow{\mathbb{P}}^{\pi_0,\sigma^2\nabla\ln\pi}}\left[\exp\left(\int_0^T \partial_t\ln\hat{\pi}_t(\boldsymbol{Y}_t)\,\mathrm{d}t\right)\right] = e^{-(\ln\mathcal{Z}_0-\ln\mathcal{Z}_T)}.$$

$\square$

We want to highlight that in (Vaikuntanathan & Jarzynski, 2008, Equations 10-14) ; we can see a similar formulation to our proposed generalised Crooks' fluctuation theorem, that said Vaikuntanathan & Jarzynski (2008) seems to pose this as a conjecture providing no rigorous proof. Furthermore, unlike our work, they do not formulate this result through SDEs, which we believe we are the first to do. In short, our work and concurrently Zhong et al. (2023) are the first to provide a rigorous treatment in establishing the escorted version of Crooks' fluctuation theorem.

### E.3 Proof of Proposition 3.1: EM $\iff$ IPF

In applications, IPF is faced with the following challenges:

1. The sequential nature of IPF, coupled with the need for each iteration to undergo comprehensive training as outlined in Section C.3, results in significant computational demands.

2. The reference distribution $r(\boldsymbol{x}, \boldsymbol{z})$ (or the reference vector field $f_t$) enters the iterations in (18) only through the initialisation. As a consequence, numerical errors accumulate, and it is often observed that the Schrödinger prior is 'forgotten' as IPF proceeds (Vargas et al., 2021a; Fernandes et al., 2021; Shi et al., 2023).

Thus to address these challenges this section will focus on establishing the connection between EM and IPF which in turn will provide us with a family of algorithms that circumvent the sequential nature of IPF, further bridging variational inference and entropic optimal transport.

*Proof.* The proof proceeds by induction.

To begin with, the update formula in (18a) implies that

$$\pi^1(\boldsymbol{x}, \boldsymbol{z}) = \underset{\pi(\boldsymbol{x}, \boldsymbol{z})}{\arg\min} \left\{ D_{\mathrm{KL}}(\pi(\boldsymbol{x}, \boldsymbol{z}) || r(\boldsymbol{x}, \boldsymbol{z})) : \pi_{\boldsymbol{x}}(\boldsymbol{x}) = \mu(\boldsymbol{x}) \right\},$$

recalling the initialisation $\pi(\boldsymbol{x}, \boldsymbol{z}) = r(\boldsymbol{x}, \boldsymbol{z})$. To take account of the marginal constraint, we may write $\pi(\boldsymbol{x}, \boldsymbol{z}) = \mu(\boldsymbol{x})\pi(\boldsymbol{z}|\boldsymbol{x})$ and vary over the conditionals $\pi(\boldsymbol{z}|\boldsymbol{x})$. By the chain rule for $D_{\mathrm{KL}}$, we see that

$$D_{\mathrm{KL}}(\mu(\boldsymbol{x})\pi(\boldsymbol{z}|\boldsymbol{x}) || r(\boldsymbol{x}, \boldsymbol{z})) = D_{\mathrm{KL}}(\mu(\boldsymbol{x}) || r(\boldsymbol{x})) + \mathbb{E}_{\boldsymbol{x} \sim \mu(\boldsymbol{x})}[D_{\mathrm{KL}}(\pi(\boldsymbol{z}|\boldsymbol{x}) || r(\boldsymbol{z}|\boldsymbol{x}))], \quad (70)$$

which is minimised at $\pi(\boldsymbol{z}|\boldsymbol{x}) = r(\boldsymbol{z}|\boldsymbol{x})$. From this, it follows that $\pi^1(\boldsymbol{x}, \boldsymbol{z}) = \mu(\boldsymbol{x})r(\boldsymbol{z}|\boldsymbol{x})$ for the first IPF iterate. By assumption, the EM iteration is initialised in such a way that $q^{\phi_0}(\boldsymbol{z}|\boldsymbol{x}) = r(\boldsymbol{z}|\boldsymbol{x})$, so that indeed $\pi^1(\boldsymbol{x}, \boldsymbol{z}) = q^{\phi_0}(\boldsymbol{z}|\boldsymbol{x})\mu(\boldsymbol{x})$.

The induction step is split (depending on whether $n$ is odd or even):

1.) First assume that the first line of (20) holds for a fixed odd $n \geq 1$. Our aim is to show that this implies that

$$\pi^{n+1}(\boldsymbol{x}, \boldsymbol{z}) = p^{\theta_{(n+1)/2}}(\boldsymbol{x}|\boldsymbol{z})\nu(\boldsymbol{z}), \quad (71)$$

that is, the second line of (20) with $n$ replaced by $n + 1$. From (18b), we see that

$$\pi^{n+1}(\boldsymbol{x}, \boldsymbol{z}) = \underset{\pi(\boldsymbol{x}, \boldsymbol{z})}{\arg\min} \left\{ D_{\mathrm{KL}}(\pi(\boldsymbol{x}, \boldsymbol{z}) || \pi^n(\boldsymbol{x}, \boldsymbol{z})) : \pi_{\boldsymbol{z}}(\boldsymbol{z}) = \nu(\boldsymbol{z}) \right\}.$$

Again, we enforce the marginal constraint by setting $\pi(\boldsymbol{x}, \boldsymbol{z}) = \pi(\boldsymbol{z}|\boldsymbol{x})\nu(\boldsymbol{z})$ and proceed as in (70) to obtain $\pi^{n+1}(\boldsymbol{x}, \boldsymbol{z}) = \pi^n(\boldsymbol{x}|\boldsymbol{z})\nu(\boldsymbol{z})$. The statement in (71) is therefore equivalent to $\pi^n(\boldsymbol{x}|\boldsymbol{z}) = p^{\theta_{(n+1)/2}}(\boldsymbol{x}|\boldsymbol{z})$. To show this, we observe from the EM-scheme in (19) that

$$\theta_{(n+1)/2} = \underset{\theta}{\arg\min} \, \mathcal{L}_{D_{\mathrm{KL}}}(\phi_{(n-1)/2}, \theta).$$

In combination with the second line of (20) and the definition of $\mathcal{L}_D(\phi, \theta)$ in (5), we obtain

$$\theta_{(n+1)/2} = \underset{\theta}{\arg\min} \, D_{\mathrm{KL}}(\pi^n(\boldsymbol{x}, \boldsymbol{z}) || p^\theta(\boldsymbol{x}|\boldsymbol{z})\nu(\boldsymbol{z})) = \underset{\theta}{\arg\min} \, \mathbb{E}_{\boldsymbol{z} \sim \pi^n_{\boldsymbol{z}}(\boldsymbol{z})} \left[ D_{\mathrm{KL}}(\pi^n(\boldsymbol{x}|\boldsymbol{z}) || p^\theta(\boldsymbol{x}|\boldsymbol{z})) \right],$$

where the second equality follows from the chain rule for $D_{\mathrm{KL}}$ as in (70). Since by assumption the parameterisation of $p^\theta(\boldsymbol{x}|\boldsymbol{z})$ is flexible, we indeed conclude that $\pi^n(\boldsymbol{x}|\boldsymbol{z}) = p^{\theta_{(n+1)/2}}(\boldsymbol{x}|\boldsymbol{z})$.

2.) Assume now that the second line of (20) holds for a fixed even $n \geq 2$. We need to show that the first line holds with $n$ replaced by $n + 1$, that is,

$$\pi^{n+1}(\boldsymbol{x}, \boldsymbol{z}) = q^{\phi_{n/2}}(\boldsymbol{z}|\boldsymbol{x})\mu(\boldsymbol{x}).$$

Using similar arguments as before, we see that $\pi^{n+1}(\boldsymbol{x}, \boldsymbol{z}) = \pi^n(\boldsymbol{x}|\boldsymbol{z})\mu(\boldsymbol{x})$, so that it is left to show that $\pi^n(\boldsymbol{x}|\boldsymbol{z}) = q^{\phi_{n/2}}(\boldsymbol{z}|\boldsymbol{x})$. Along the same lines as in 1.), we obtain

$$\phi_{n/2} = \underset{\phi}{\arg\min} \, \mathcal{L}_{D_{\mathrm{KL}}}(\phi, \theta_{n/2}) = \underset{\phi}{\arg\min} \, D_{\mathrm{KL}}(q^\phi(\boldsymbol{z}|\boldsymbol{x})\mu(\boldsymbol{x}) || \pi^n(\boldsymbol{x}, \boldsymbol{z}))$$

$$= \underset{\phi}{\arg\min} \, \mathbb{E}_{\boldsymbol{x} \sim \mu(\boldsymbol{x})} \left[ q^\phi(\boldsymbol{z}|\boldsymbol{x}) || \pi^n(\boldsymbol{z}|\boldsymbol{x}) \right].$$

Again, this allows us to conclude, since the parameterisation in $q^\phi(\boldsymbol{z}|\boldsymbol{x})$ is assumed to be flexible enough to allow for $q^{\phi_{n/2}}(\boldsymbol{z}|\boldsymbol{x}) = \pi^n(\boldsymbol{x}|\boldsymbol{z})$.

The proof for the path space IPF scheme is verbatim the same after adjusting the notation. For completeness, we consider a drift-wise version below. □

**Remark 11** (Extension to $f$-divergences). The proof does not make use of specific properties of $D_{\mathrm{KL}}$, other than that it satisfies the chain rule. As a consequence, the statement of Proposition 3.1 straightforwardly extends to other divergences with this property, in particular to $f$-divergences, see Proposition 6 in (Baudoin, 2002).

### E.4 DRIFT BASED EM

As remarked in the previous subsection, the proof of the equivalence between IPF and EM in path space follows the exact same lines, replacing the chain rule of $D_{\mathrm{KL}}$ with the (slightly more general) disintegration theorem (Léonard, 2014b). In this section, we provide a direct extension to the control setting, yielding yet another IPF-type algorithm and motivating certain design choices for the family of methods we study.

**Corollary E.2** (Path space EM). *For the initialisation $\phi_0 = 0$ the alternating scheme*

$$
\theta_{n+1} = \arg\min_\theta D_{\mathrm{KL}}(\overrightarrow{\mathbb{P}}^{\mu, f + \sigma^2 \nabla \phi_n}, \overleftarrow{\mathbb{P}}^{\nu, f + \sigma^2 \nabla \theta}),
$$
$$
\phi_{n+1} = \arg\min_\phi D_{\mathrm{KL}}(\overrightarrow{\mathbb{P}}^{\mu, f + \sigma^2 \nabla \phi}, \overleftarrow{\mathbb{P}}^{\nu, f + \sigma^2 \nabla \theta_{n+1}}) \tag{73}
$$

*agrees with the path space IPF iterations in (Bernton et al., 2019; Vargas et al., 2021a; De Bortoli et al., 2021).*

*Proof.* For brevity let $\mathcal{L}_{\mathrm{FB}}(\phi, \theta) := D_{\mathrm{KL}}(\overrightarrow{\mathbb{P}}^{\mu, f + \sigma^2 \nabla \phi}, \overleftarrow{\mathbb{P}}^{\nu, f + \sigma^2 \nabla \theta})$. Additionally, we parameterise the forwards and backwards SDEs with respective path distributions $\overrightarrow{\mathbb{P}}^{\mu, f + \sigma^2 \nabla \phi}$, $\overleftarrow{\mathbb{P}}^{\nu, f + \sigma^2 \nabla \theta}$ as:

$$
\mathrm{d}\boldsymbol{Y}_t = f_t(\boldsymbol{Y}_t)\, \mathrm{d}t + \sigma^2 \nabla \phi_t(\boldsymbol{Y}_t)\, \mathrm{d}t + \sigma\, \overrightarrow{\mathrm{d}}\, \boldsymbol{W}_t, \quad \boldsymbol{Y}_0 \sim \mu,
$$
$$
\mathrm{d}\boldsymbol{Y}_t = f_t(\boldsymbol{Y}_t)\, \mathrm{d}t + \sigma^2 \nabla \theta_t(\boldsymbol{Y}_t)\, \mathrm{d}t + \sigma\, \overleftarrow{\mathrm{d}}\, \boldsymbol{W}_t, \quad \boldsymbol{Y}_T \sim \nu.
$$

The proof will proceed quite similarly, so instead we will consider just the inductive step for the odd half bridge:

$$
\theta_n = \arg\min_\theta \mathcal{L}_{\mathrm{FB}}(\phi_{n-1}, \theta).
$$

We can show via the $D_{\mathrm{KL}}$ chain rule and the disintegration theorem (Léonard, 2014b) that the above is minimised when $\theta$ satisfies $\overleftarrow{\mathbb{P}}^{\nu, f + \sigma^2 \nabla \theta} = \overrightarrow{\mathbb{P}}^{\mu, f + \sigma^2 \nabla \phi_{n-1}} \frac{\mathrm{d}\nu}{\mathrm{d}\rho_T^{\mu, f + \sigma^2 \nabla \phi_{n-1}}}$ which corresponds to

$\nabla \theta_n = \sigma^2 \nabla \phi_{n-1} - \sigma^2 \nabla \ln \rho_t^{\mu, f + \sigma^2 \nabla \phi_{n-1}}$ following Observation 1 in Vargas et al. (2021a). Similarly as per Proposition 3.1 the results will follow for the even half bridges.

$\qquad\qquad\qquad\qquad\qquad\qquad\qquad\qquad\qquad\qquad\qquad\qquad\qquad\qquad\qquad\qquad\qquad\qquad\qquad\quad\square$

**EM initialisation:** The above corollary provides us with convergence guarantees when performing coordinate descent on $D_{\mathrm{KL}}(\overrightarrow{\mathbb{P}}^{\mu, f + \sigma^2 \nabla \phi}, \overleftarrow{\mathbb{P}}^{\nu, f + \sigma^2 \nabla \theta})$ subject to initialising $\phi_0 = 0$. n practice, this indicates that the way of initialising $\phi$ has a major impact on which bridge we converge to.

Thus as a rule of thumb we propose initialising $\phi_0 = 0$ such that we initialise at the Schrödinger prior: then one may carry out joint updates as an alternate heuristic, we call this approach DNF (EM Init), as it is effectively a clever initialisation of DNF inspired by the relationship between IPF and EM.

### E.5 HJB-REGULARIZERS

As per Section 3.1, IPF resolves the nonquniqueness in minimising $\mathcal{L}_D(\phi, \theta)$ by performing the coordinate-wise updates (19) starting from an initialisation informed by the Schrödinger prior. On the basis of this observation, the joint updates $(\phi_{n+1}, \theta_{n+1}) \leftarrow (\phi_n, \theta_n) - h \nabla_{\phi, \theta} \mathcal{L}_D(\phi, \theta)$ suggest themselves, in the spirit of VAEs (Kingma et al., 2019) and as already proposed in this setting by Neal & Hinton (1998). However, as is clear from the introduction, the limit $\lim_{n \to \infty}(\phi_n, \theta_n)$, can merely be expected to respect the marginals in (6), and no optimality in the sense of (17) is expected. As a remedy, we present the following result:

**Proposition E.3.** *For $\lambda > 0$, a divergence $D$ on path space, and $\phi, \psi \in C^{1,2}([0,T] \times \mathbb{R}^d; \mathbb{R})$, let*

$$\mathcal{L}^{\text{Schr}}(\phi, \theta) := D(\overrightarrow{\mathbb{P}}^{\mu, f + \sigma^2 \nabla \phi}, \overleftarrow{\mathbb{P}}^{\nu, f + \sigma^2 \nabla \theta}) + \lambda \text{Reg}(\phi), \tag{74}$$

*where* $\text{Reg}(\phi) = 0$ *if and only if the HJB-equation* $\partial_t \phi + f \cdot \nabla \phi + \frac{\sigma^2}{2} \Delta \phi + \frac{\sigma^2}{2} |\nabla \phi|^2 = 0$ *holds. Then* $\mathcal{L}_{\text{Schr}}(\phi, \theta) = 0$ *implies that the drift* $a_t := \sigma^2 \nabla \phi_t$ *solves (17).*

The proof rests on an optimal control reformulation of the Schrödinger problem (see Appendix E), identifying the HJB-equation as the missing link that renders joint minimisation of (74) theoretically sound for solving (17). The loss in (74) has two important benefits compared to standard IPF. First, it circumvents the need for the sequential updates used in IPF, thereby simplifying and speeding up the optimisation procedure. Second, it enforces the Schrödinger prior drift $f$ directly, rather than recursively via eq. (18a), (18b). This prevents the prior from being forgotten, as is usually the case in regular IPF. In Appendix E.5, we detail possible constructions of $\text{Reg}(\phi)$, discuss relationships to previous work, and evaluate the performance of the suggested approach in numerical experiments.

This result can be found in Chen et al. (2021, Proposition 5.1), for instance, but since it is relevant to the connections pointed out in Remark 13 below, we present an independent proof:

**Proposition E.4** (Mean-field game formulation)**.** *Assume that* $\phi \in C^{1,2}([0,T] \times \mathbb{R}^d; \mathbb{R})$ *satisfies the conditions:*

1. *The forward SDE*

   $$\mathrm{d}\boldsymbol{Y}_t = f_t(\boldsymbol{Y}_t)\,\mathrm{d}t + \sigma^2 \nabla \phi_t(\boldsymbol{Y}_t)\,\mathrm{d}t + \sigma \,\overrightarrow{\mathrm{d}\boldsymbol{W}}_t, \ \boldsymbol{Y}_0 \sim \mu \tag{75}$$

   *admits a unique strong solution on $[0,T]$, satisfying moreover the terminal constraint* $\boldsymbol{Y}_T \sim \nu$.

2. *The Hamilton-Jacobi-Bellmann (HJB) equation*

   $$\partial_t \phi + f \cdot \nabla \phi + \frac{\sigma^2}{2} \Delta \phi + \frac{\sigma^2}{2} |\nabla \phi|^2 = 0 \tag{76}$$

   *holds for all $(t,x) \in [0,T] \times \mathbb{R}^d$.*

*Then $a = \sigma^2 \nabla \phi$ provides the unique solution to the dynamical Schrödinger problem as posed in (17).*

*Proof.* We denote the path measures associated to the SDE

$$\mathrm{d}\boldsymbol{Y}_t = f_t(\boldsymbol{Y}_t)\,\mathrm{d}t + \sigma\,\mathrm{d}\boldsymbol{W}_t \tag{77}$$

by $\mathbb{P}$ and the SDE (75) by $\mathbb{P}^\phi$, respectively. According to Girsanov's theorem, the Radon-Nikodym derivative satisfies

$$\frac{\mathrm{d}\mathbb{P}^\phi}{\mathrm{d}\mathbb{P}} = \exp\left(\sigma \int_0^T \nabla \phi_t(\boldsymbol{Y}_t) \cdot \mathrm{d}\boldsymbol{W}_t - \frac{\sigma^2}{2} \int_0^T |\nabla \phi_t|^2(\boldsymbol{Y}_t)\,\mathrm{d}t\right), \tag{78}$$

provided that the marginals agree at initial time, $\mathbb{P}_0 = \mathbb{P}_0^\phi$. Along solutions of (77), we have by Itô's formula

$$\phi_T(\boldsymbol{Y}_T) - \phi_0(\boldsymbol{Y}_0) = \int_0^T \partial_t \phi_t(\boldsymbol{Y}_t)\,\mathrm{d}t + \int_0^T (f_t \cdot \nabla \phi_t)(\boldsymbol{Y}_t)\,\mathrm{d}t + \frac{\sigma^2}{2} \int_0^T \Delta \phi_t(\boldsymbol{Y}_t)\,\mathrm{d}t + \sigma \int_0^T \nabla \phi_t(\boldsymbol{Y}_t) \cdot \mathrm{d}\boldsymbol{W}_t$$

$$= -\frac{\sigma^2}{2} \int_0^T |\nabla \phi_t|^2(\boldsymbol{Y}_t)\,\mathrm{d}t + \sigma \int_0^T \nabla \phi_t(\boldsymbol{Y}_t) \cdot \mathrm{d}\boldsymbol{W}_t = \ln\left(\frac{\mathrm{d}\mathbb{P}^\phi}{\mathrm{d}\mathbb{P}}\right), \tag{79}$$

where we have used the HJB-equation (76) in the second line. Combining this with (78), we see that

$$\frac{\mathrm{d}\mathbb{P}^\phi}{\mathrm{d}\mathbb{P}}(\boldsymbol{Y}) = \exp\left(-\phi_0(\boldsymbol{Y}_0)\right) \exp\left(\phi_T(\boldsymbol{Y}_T)\right). \tag{80}$$

The claim now follows, since the unique solution to the Schrödinger problem is characterised by the product-form expression in (80, see Léonard (2014a, Section 2), together with the marginal constraints $\mathbb{P}_0^\phi = \mu$ and $\mathbb{P}_T^\phi = \nu$, which are satisfied by assumption. $\square$

**Remark 12** (Summarised relationship to previous work). For $\lambda = 0$, coordinate-wise updates of $\mathcal{L}_{\mathrm{Schr}}(\phi, \theta)$ recover the IPF updates from De Bortoli et al. (2021); Vargas et al. (2021a) according to Corollary 3.1. Note that $\mathcal{L}_{\mathrm{Schr}}$ is an unconstrained objective, in contrast to (17); previous works (Koshizuka & Sato, 2023; Zhang & Katsoulakis, 2023) have suggested incorporating the marginal constraints softly by adding penalising terms to the running cost in (17). Those approaches require a limiting argument (from an algorithmic standpoint, adaptive tuning of a weight parameter) to recover the solution to (17). In contrast, the conclusion of Proposition E.3 holds for arbitrary $\lambda > 0$. Shi et al. (2023); Peluchetti (2023) suggest an algorithm involving reciprocal projections onto the reciprocal class associated to $f_t$. From Clark (1991); Thieullen (2002); Rœlly (2013), the HJB-equation (76) is a local characteristic ($\mathrm{Reg}(\phi) = 0$ forces (75) to be in the reciprocal class); hence $\mathrm{Reg}(\phi)$ in (74) plays a similar role as the reciprocal projection (Shi et al., 2023, Definition 3), see Remark 13. Liu et al. (a) suggest an iterative IPF-like scheme involving a temporal difference term (Sutton & Barto, 2018, Chapter 6). As in Nüsken & Richter (2023), this is a an HJB-regulariser in the sense of Proposition E.3, see Remark 13. Finally, Albergo et al. (2023, Theorem 5.3) and Gushchin et al. (2022) develop saddle-point objectives for (17).

**Remark 13** (Connection to *reciprocal classes* (Shi et al., 2023; Peluchetti, 2023) and *TD learning* (Liu et al., a)). The calculation in equation (79) makes the relationship between the HJB equation (76) and reciprocal classes manifest (since reciprocal classes can essentially be defined through the relationship (80), see Léonard et al. (2014); Rœlly (2013)). Moreover, equation (79) showcases the relationship between TD learning (Sutton & Barto, 2018, Chapter 6) as suggested in Liu et al. (a) and HJB regularisation. Indeed,

$$\mathrm{Reg}_{\mathrm{BSDE}}(\phi) := \mathrm{Var}\left(\phi_T(\boldsymbol{Y}_T) - \phi_0(\boldsymbol{Y}_0) + \tfrac{\sigma^2}{2}\int_0^T |\nabla\phi_t|^2(\boldsymbol{Y}_t)\,\mathrm{d}t - \sigma\int_0^T \nabla\phi_t(\boldsymbol{Y}_t)\cdot\mathrm{d}\boldsymbol{W}_t\right),$$
(81)

where the variance is taken with respect to the path measure induced by (77), is a valid HJB-regulariser in the sense of Proposition E.3. The equivalence between $\mathrm{Reg}_{\mathrm{BSDE}}(\phi) = 0$ and the HJB equation (76) follows from the theory of backward stochastic differential equations (BSDEs)[12], see, for example, the proof of Proposition 3.4 in Nüsken & Richter (2023) and the discussion in Nüsken & Richter (2021, Section 3.2).

In the following, we present an analogue of Proposition E.4 involving the backward drift (Chen et al., 2019):

**Proposition E.5.** *Assume that* $\theta \in C^{1,2}([0, T] \times \mathbb{R}^d; \mathbb{R})$ *satisfies the following two conditions:*

1. *The backward SDE*

$$\mathrm{d}\boldsymbol{Y}_t = f_t(\boldsymbol{Y}_t)\,\mathrm{d}t + \sigma^2\nabla\theta_t(\boldsymbol{Y}_t)\,\mathrm{d}t + \sigma\,\overset{\leftarrow}{\mathrm{d}}\,\boldsymbol{W}_t, \qquad \boldsymbol{Y}_T \sim \nu$$
(82)

*admits a unique strong solution on* $[0, T]$*, satisfying moreover the initial constraint* $\boldsymbol{Y}_0 \sim \mu$.

2. *The Hamilton-Jacobi-Bellmann (HJB) equation*

$$\partial_t\theta + f\cdot\nabla\theta - \frac{\sigma^2}{2}\Delta\theta + \tfrac{\sigma^2}{2}|\nabla\theta|^2 - \nabla\cdot f = 0$$
(83)

*holds for all* $(t, x) \in [0, T] \times \mathbb{R}^d$.

*Assuming furthermore that the solution to (82) admits a smooth positive density* $\rho$*, we have that* $a_t = \nabla\theta_t + \sigma^2\nabla\ln\rho_t$ *provides the unique solution to the Schrödinger problem as posed in (17).*

**Remark 14.** As opposed to Chen et al. (2016, equation (41)), the HJB-equation (83) does not involve the time reversal of the Schrödinger prior; the form of the HJB equations is not uniquely determined. On the other hand, (83) contains the divergence term $\nabla \cdot f$, which discourages us from enforcing this constraint in the same way as (76). An akin result can be found in Liu et al. (a) stated in terms of BSDEs.

---

[12]... not to be confused with reverse-time SDEs as in (12).

*Proof of Corollary E.5.* Using the forward-backward Radon-Nikodym derivative in (14), we compute

$$\ln\left(\frac{\mathrm{d}\,\overrightarrow{\mathbb{P}}^{\,\mu,f}}{\mathrm{d}\,\overleftarrow{\mathbb{P}}^{\,\nu,f+\sigma^2\nabla\psi}}\right)(\boldsymbol{Y}) = \ln\left(\frac{\mathrm{d}\mu}{\mathrm{d}\mathrm{Leb}}\right) - \ln\left(\frac{\mathrm{d}\nu}{\mathrm{d}\mathrm{Leb}}\right) + \sigma\int_0^T f_t(\boldsymbol{Y}_t)\cdot\mathrm{d}\boldsymbol{W}_t - \sigma\int_0^T f_t(\boldsymbol{Y}_t)\cdot\overleftarrow{\mathrm{d}}\,\boldsymbol{W}_t$$

$$-\sigma\int_0^T\nabla\theta_t(\boldsymbol{Y}_t)\cdot\overleftarrow{\mathrm{d}}\,\boldsymbol{W}_t + \frac{\sigma^2}{2}\int_0^T|\nabla\theta_t|^2(\boldsymbol{Y}_t)\,\mathrm{d}t$$

$$= \ln\left(\frac{\mathrm{d}\mu}{\mathrm{d}\mathrm{Leb}}\right) - \ln\left(\frac{\mathrm{d}\nu}{\mathrm{d}\mathrm{Leb}}\right) - \sigma\int_0^T(\nabla\cdot f_t)(\boldsymbol{Y}_t)\,\mathrm{d}t - \sigma\int_0^T\nabla\theta_t(\boldsymbol{Y}_t)\cdot\overleftarrow{\mathrm{d}}\,\boldsymbol{W}_t + \frac{\sigma^2}{2}\int_0^T|\nabla\theta_t|^2(\boldsymbol{Y}_t)\,\mathrm{d}t.$$

Here we have chosen $\overrightarrow{\gamma} = \overleftarrow{\gamma} = 0$, and $\Gamma_0 = \Gamma_T = \mathrm{Leb}$. The initial measure for the Schrödinger prior is $\mu$, but the argument is unaffected by this choice (as the solution is independent of this). We now use the (backward) Itô formula along the Schrödinger prior,

$$\theta_t(\boldsymbol{Y}_T) - \theta_0(\boldsymbol{Y}_0) = \int_0^T\partial_t\theta_t(\boldsymbol{Y}_t)\,\mathrm{d}t + \int_0^T\nabla\theta_t(\boldsymbol{W}_t)\cdot\overleftarrow{\mathrm{d}}\,\boldsymbol{W}_t + \int_0^T\nabla\theta_t(\boldsymbol{Y}_t)\cdot f_t(\boldsymbol{Y}_t)\,\mathrm{d}t - \frac{1}{2}\int_0^T\Delta\theta_t(\boldsymbol{Y}_t)\,\mathrm{d}t.$$

Using the HJB-equation (83), we see that

$$\ln\left(\frac{\mathrm{d}\,\overrightarrow{\mathbb{P}}^{\,\mu,f}}{\mathrm{d}\,\overleftarrow{\mathbb{P}}^{\,\nu,f+\sigma^2\nabla\theta}}\right)(\boldsymbol{Y}) = \ln\left(\frac{\mathrm{d}\mu}{\mathrm{d}\mathrm{Leb}}\right) - \ln\left(\frac{\mathrm{d}\nu}{\mathrm{d}\mathrm{Leb}}\right) - \theta_t(\boldsymbol{Y}_T) + \theta_0(\boldsymbol{Y}_0), \qquad (84)$$

and we can conclude as in the proof of Proposition E.4. $\qquad\square$

# F    CMCD EXPERIMENTS

In this section, we will cover further details pertaining to our experimental setup.

## F.1    ELBO EXPERIMENTS AND COMPARISON TO GEFFNER & DOMKE (2023)

We compare our underdamped and overdamped CMCD variants against 5 datasets from Geffner & Domke (2023), which we describe in further detail below.

- log_sonar ($d = 61$) and log_ionosphere ($d = 35$) are Bayesian logistic regression models: $x \sim \mathcal{N}(0, \sigma_w^2 I), y_i \sim \mathrm{Bernoulli}(\mathrm{sigmoid}(x^\top u_i))$ with posteriors conditioned on the *sonar* and *ionosphere* datasets respectively.

- brownian ($d = 32$) corresponds to the time discretisation of a Brownian motion:

$$\begin{aligned}
\alpha_{\mathrm{inn}} &\sim \mathrm{LogNormal}(0, 2),\\
\alpha_{\mathrm{obs}} &\sim \mathrm{LogNormal}(0, 2),\\
x_1 &\sim \mathcal{N}(0, \alpha_{\mathrm{inn}}),\\
x_i &\sim \mathcal{N}(x_{i-1}, \alpha_{\mathrm{inn}}),\quad i = 2,\ldots 20,\\
y_i &\sim \mathcal{N}(x_i, \alpha_{\mathrm{obs}}),\quad i = 1,\ldots 30.
\end{aligned}$$

  inference is performed over the variables $\alpha_{\mathrm{inn}}, \alpha_{\mathrm{obs}}$ and $\{x_i\}_{i=1}^{30}$ given the observations $\{y_i\}_{i=1}^{10} \cup \{y_i\}_{i=20}^{30}$.

- lorenz ($d = 90$) is the discretisation of a highly stiff 3-dimensional SDE that models atmospheric convection:

$$\begin{aligned}
x_1 &\sim \mathcal{N}(\,\mathrm{loc} = 0,\ \mathrm{scale} = 1)\\
y_1 &\sim \mathcal{N}(\,\mathrm{loc} = 0,\ \mathrm{scale} = 1)\\
z_1 &\sim \mathcal{N}(\,\mathrm{loc} = 0,\ \mathrm{scale} = 1)\\
x_i &\sim \mathcal{N}(\,\mathrm{loc} = 10\,(y_{i-1} - x_{i-1}),\ \mathrm{scale} = \alpha_{\mathrm{inn}})& i = 2,\ldots,30\\
y_i &\sim \mathcal{N}(\,\mathrm{loc} = x_{i-1}(28 - z_{i-1}) - y_{i-1}),\ \mathrm{scale} = \alpha_{\mathrm{inn}})& i = 2,\ldots,30\\
z_i &\sim \mathcal{N}(\,\mathrm{loc} = x_{i-1}y_{i-1} - \tfrac{8}{3}z_{i-1},\ \mathrm{scale} = \alpha_{\mathrm{inn}})& i = 2,\ldots,30,\\
o_i &\sim \mathcal{N}(\,\mathrm{loc} = x_i,\ \mathrm{scale} = 1)& i = 2,\ldots,30
\end{aligned}$$

  where $\alpha_{\mathrm{inn}} = 0.1$ (determined by the discretization step-size used for the original SDE). The goal is to do inference over $x_i, y_i, z_i$ for $i = 1,\ldots,30$, given observed values $o_i$ for $i \in \{1,\ldots,10\} \cup \{20,\ldots,30\}$.

- `seeds` ($d = 26$) is a random effect regression model trained on the *seeds* dataset:

$$\tau \sim \text{Gamma}(0.01, 0.01)$$
$$a_0 \sim \mathcal{N}(0, 10)$$
$$a_1 \sim \mathcal{N}(0, 10)$$
$$a_2 \sim \mathcal{N}(0, 10)$$
$$a_{12} \sim \mathcal{N}(0, 10)$$
$$b_i \sim \mathcal{N}\left(0, \frac{1}{\sqrt{\tau}}\right)$$
$$i = 1, \ldots, 21$$
$$\text{logits}_i = a_0 + a_1 x_i + a_2 y_i + a_{12} x_i y_i + b_1$$
$$i = 1, \ldots, 21$$
$$r_i \sim \text{Binomial}(\text{logits}_i, N_i)$$
$$i = 1, \ldots, 21.$$

The goal is to do inference over the variables $\tau, a_0, a_1, a_2, a_{12}$ and $b_i$ for $i = 1, \ldots, 21$, given observed values for $x_i, y_i$ and $N_i$.

For all target distributions, we follow the hyperparameter setup from Geffner & Domke (2023) from their code repository[13] for all baseline methods (ULA, MCD, UHA, and LDVI) as well as our overdamped and underdamped variants. We first pretrain the source distribution to a mean-field Gaussian distribution trained for $150,000$ steps with ADAM and a learning rate of $10^{-2}$. We then train for $150000$ iterations with a batch size of 5, tuning learning rate between $[10^{-5}, 10^{-4}, 10^{-3}]$ picking the best one based on mean ELBO after training. For all methods, during training the mean-field source distribution is continued to be trained, as well as the discretisation step size and $\epsilon = \delta t \sigma$. For the underdamped methods we also train the damping coefficient $\gamma$, and for methods involving a score network, i.e. MCD, LDVI, CMCD and CMCD (UD), we train the networks which are chosen to be fully-connected residual networks with layer sizes of $[20, 20]$. In order to report the mean ELBO after training, we obtain 500 samples with 30 seeds and report an averaged value over them.

### F.2 $\ln Z$, SAMPLE QUALITY EXPERIMENTS AND COMPARISON TO (ZHANG & CHEN, 2022; VARGAS ET AL., 2023A)

Furthermore, we also include comparisons to a large-dimensional target distribution and two standard distributions with known $\ln Z$ replicated from Vargas et al. (2023a), which we summarise below.

- `lgcp` ($d = 1600$) is a high-dimensional Log Gaussian Cox process popular in spatial statistics (Møller et al., 1998). Using a $d = M \times M = 1600$ grid, we obtain the unnormalised target density $\mathcal{N}(x; \mu, K) \prod_{i \in [1:M]^2} \exp(x_i y_i - a \exp(x_i))$.

- `funnel` ($d = 10$) is a challenging distribution given by $\pi_T(x_{1:10}) = \mathcal{N}(x_1; 0, \sigma_f^2) \mathcal{N}(x_{2:10}; 0, \exp(x_1)I)$, with $\sigma_f^2 = 9$ (Neal, 2003).

- `gmm` ($d = 2$) is a two-dimensional Gaussian mixture model with three modes, given by the following target distribution

$$\pi_T(x) = \frac{1}{3}\mathcal{N}\left(x; \begin{bmatrix} 3 \\ 0 \end{bmatrix}, \begin{bmatrix} 0.7 & 0 \\ 0 & 0.05 \end{bmatrix}\right) + \frac{1}{3}\mathcal{N}\left(x; \begin{bmatrix} -2.5 \\ 0 \end{bmatrix}, \begin{bmatrix} 0.7 & 0 \\ 0 & 0.05 \end{bmatrix}\right)$$
$$+ \mathcal{N}\left(x; \begin{bmatrix} 2 \\ 3 \end{bmatrix}, \begin{bmatrix} 1 & 0.95 \\ 0.95 & 1 \end{bmatrix}\right)$$

For these target distributions, we follow the hyperparameter setup from Vargas et al. (2023a) from their code repository[14]for the baseline methods of DDS and PIS, and replicate them as closely as

---

[13]https://github.com/tomsons22/LDVI
[14]https://github.com/franciscovargas/denoising_diffusion_samplers

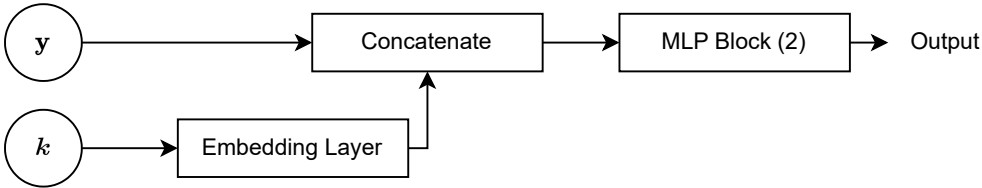

Figure 2: Architecture from (Geffner & Domke, 2023) used across experiments for our CMCD drift network. Softplus activations are used.

possible for CMCD. Unlike the previous, we don't pretrain the mean-field Gaussian source distribution $\mathcal{N}(0, \sigma_{\text{init}}^2 I)$. We select the optimal learning rate in $[10^{-3}, 10^{-4}, 10^{-5}]$, the optimal standard deviation of the source distribution $\sigma_{\text{init}}$ in $[1, 2, 3, 4, 5]$ and the optimal $\alpha$ in $[0.1, 0.5, 1, 1.5, 2]$. Instead of training $\epsilon = \delta t \sigma$, we sweep over an optimal value in $[10^{-2}, 10^{-1}, 1]$. The models are trained with a batch size of 300 for 11000 steps, where we keep the source distribution parameters fixed, as well as $\epsilon$. For evaluation, we use 30 seeds with a batch size of 2000, and report average performance over the seeds. DDS and PIS use a 128-dimensional positional embedding, along with an additional network for the time parameters, however MCMD uses a regular score network. In order to make exact comparisons, we select differing network architecture sizes that result in an equivalent number of parameters for `funnel` and `gmm`. For `lgcp`, due to the high dimensionality of the dataset, we choose a small network for CMCD. We summarise these below. For `gmm` and `funnel`, it is possible to sample from the target distribution, and we report an OT-regularised distance $(\mathcal{W}_2^\gamma)$ with a regularisation $\gamma = 10^{-2}$. Similar to the mean ELBO, we draw 2000 samples from the models and the targets, and average $\mathcal{W}_2^\gamma$ over 30 seeds. We use the Python Optimal Transport[15] library's default implementation of entropy-regularised distance. Results for comparisons to DNF can be found in Table 5.

Table 1: **Network Sizes for comparison.** Note that CMCD has less parameters for the despite the Funnel target despite the larger drift due to the PIS and DDS networks having an additional grad network.

|       | GMM        | LGCP       | FUNNEL       |
|-------|------------|------------|--------------|
| DDS   | $[10, 10]$ | $[64, 64]$ | $[64, 64]$   |
| PIS   | $[10, 10]$ | $[64, 64]$ | $[64, 64]$   |
| CMCD  | $[38, 38]$ | $[64, 64]$ | $[110, 110]$ |

### F.3 COMPARISONS WITH THE LOG-VARIANCE LOSS - MODE COLLAPSE FAILURE MODE

Here, we report performance using the log-variance divergence-based loss (Nüsken & Richter, 2021) introduced at the end of Section 3.2,

$$\mathcal{L}_{\text{Var}}^{\text{CMCD}}(\phi) \approx \text{Var}\left[\ln \frac{\pi_0(\boldsymbol{Y}_0)}{\hat{\pi}(\boldsymbol{Y}_T)} \prod_{k=0}^{K-1} \frac{\mathcal{N}(\boldsymbol{Y}_{t_{k+1}} | \boldsymbol{Y}_{t_k} + (\nabla \ln \pi_{t_k} + \nabla \ln \phi_{t_k})(\boldsymbol{Y}_{t_k}) \Delta t_k, 2\sigma^2 \Delta t_k)}{\mathcal{N}(\boldsymbol{Y}_{t_k} | \boldsymbol{Y}_{t_{k+1}} + (\nabla \ln \pi_{t_{k+1}} - \nabla \ln \phi_{t_{k+1}})(\boldsymbol{Y}_{t_{k+1}}) \Delta t_k, 2\sigma^2 \Delta t_k)}\right], \tag{24}$$

A careful reader will note this loss simply consists of replacing the expectation in the KL loss with a variance. A major computational advantage of this loss is that the measure that the expectations are taken with respect to can be any measure and is not restricted to the forward or backward SDEs like in KL (Richter & Berner, 2024; Richter et al., 2020; Nüsken & Richter, 2021), this allows us to detach the samples and thus accommodating for a much more computational objective.

which we find performs quite well compared to our default loss function, especially for multimodal target distributions. We consider the very multi-modal mixture of Gaussian target distribution from Midgley et al. (2022), and report the ELBO and $\ln Z$ numbers in the table below. For this experiment, we use a batch size of 2000 and train neural networks with a size $[130, 130]$ for $150k$ iterations.

---

[15]https://pythonot.github.io/

Table 2: **ELBO and** $\ln Z$ **on 40-GMM**

|  | ELBO | $\ln Z$ | $\mathcal{W}_2$ |
|---|---|---|---|
| *log-variance* LOSS | $-1.279 \pm 0.096$ | $-0.065 \pm 0.101$ | $0.0143 \pm 0.001$ |
| KL LOSS | $-2.286 \pm 0.1109$ | $-0.244 \pm 0.3309$ | $0.0441 \pm 0.012$ |

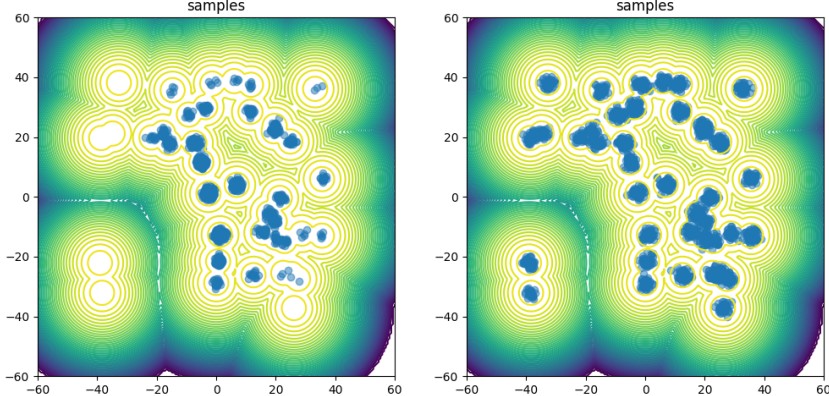

Figure 3: (left) 2000 samples drawn from the CMCD algorithm trained with the default loss function, and (right) 2000 samples drawn from the algorithm trained with the log-variance divergence-based loss. We can see that the default loss function misses many modes in the target distribution, whereas the log-variance loss has not missed any modes. We report final results after sweeping over $\Delta_{t_k}$ and learning rates for both methods, picking the one with the lowest training loss. We highlight that concurrent work by Richter & Berner (2024) explores the log variance divergence in more detail and proposes an akin general framework for diffusion-based sampling.

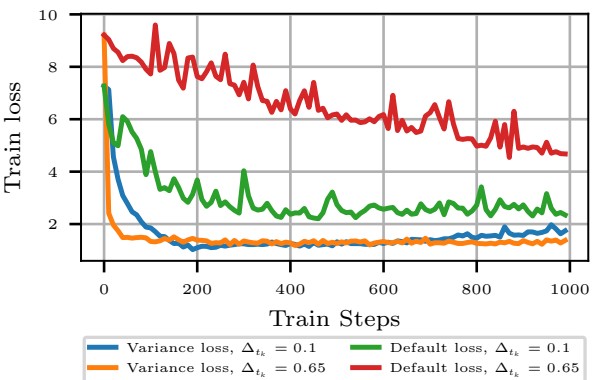

Figure 4: Plots showing training loss curves for the log-variance loss and the default loss for different values of $\Delta_{t_k}$. We find that a low value of $\Delta_{t_k} = 0.1$ is needed in order to obtain a low training loss for the default loss, whereas the log-variance loss is much more robust to different values of $\Delta_{t_k}$. The x-axis reports an evaluation every 150 steps of training

### F.4 SPECIFICATION AND TUNING: SMC, AFT, AND NF-VI

We adopted the implementations[16] provided by the studies in Arbel et al. (2021); Matthews et al. (2022) and initialized them with default hyperparameters before fine-tuning.

---

[16]https://github.com/google-deepmind/annealed_flow_transport

**Sequential Monte Carlo (SMC).** For SMC, we utilized 2000 particles sampled from a zero-mean unit Gaussian distribution, implementing re-sampling if the effective sample size (ESS) fell below 0.3. We employed Hamiltonian Monte-Carlo (HMC) for particle mutation, executing one Markov Chain Monte Carlo (MCMC) step after each annealing step. The number of leapfrog steps was fixed at 10, and an extensive grid search over different step sizes was conducted, consistent with Arbel et al. (2021). This search spanned four different step sizes, contingent on the temperature, resulting in a grid search over 256 parameters. The finalized values are presented in Table 3. For SMC, $K$ is defined by the number of temperatures.

Furthermore, we report the results for SMC in Table 3 for the tuned hyperparameters used for each target. Please note that we were able to obtain similar $\ln Z$ values as in Vargas et al. (2023a) suggesting SMC was well-tuned. Finally, for each result, we report the mean and standard deviations across 30 different seeds, results can be seen in Table 3.

**Annealed Flow Transport Monte Carlo (AFT).** We maintained a similar setup to SMC, with a few adjustments: using 500 particles for training and 2000 for evaluation to accommodate the added complexity from the normalizing flows. We also decreased the number of temperatures and increased the number of MCMC steps to mitigate memory requirements from the flows. $K$ is defined as the number of temperatures $\times$ MCMC steps, with the latter fixed at 4, resulting in a maximum of 64 flows trained simultaneously. Inverse autoregressive flows (IAFs) were employed in all experiments except for *lgcp*, using a neural network with one hidden layer whose dimension matches the problem's dimensionality. For *lgcp*, a diagonal affine flow was used due to memory constraints arising from the high dimensionality. AFT flows were trained for 300 iterations until convergence.

**Variational Inference with Normalizing Flows (VI-NF).** We utilized the same flows as for AFT. In this case, $K$ denotes the number of flows to stack. The flows were trained over a total of 2000 iterations with a batch size of 500. For some targets

Table 3: **Tuned MCMC Step Sizes.**

| | GMM | LGCP | LORENZ | BROWNIAN | LOG_SONAR | LOG_IONOSPHERE | SEEDS | FUNNEL |
|---|---|---|---|---|---|---|---|---|
| $\Delta t$ | $[0.5, 0.5, 0.5, 0.3]$ | $[0.3, 0.3, 0.2, 0.2]$ | $[0.01, 0.01, 0.008, 0.01]$ | $[0.2, 0.2, 0.05, 0.05]$ | $[0.2, 0.05, 0.2, 0.2]$ | $[0.1, 0.2, 0.2, 0.2]$ | $[0.2, 0.1, 0.05, 0.01]$ | $[0.05, 0.2, 0.2, 0.05]$ |

Table 4: **SMC Results.** ELBO and $\ln Z$ values for a different number of steps $K$ and experiments.

| $\ln Z$ | GMM | LGCP | LORENZ | BROWNIAN | LOG_SONAR | LOG_IONOSPHERE | SEEDS | FUNNEL |
|---|---|---|---|---|---|---|---|---|
| $K = 8$ | $-0.536 \pm 0.042$ | $-364.074 \pm 7.797$ | $-87502.352 \pm 4004.495$ | $-63.32 \pm 8.016$ | $-178.589 \pm 2.784$ | $-204.594 \pm 3.049$ | $-108.676 \pm 1.221$ | $-1.013 \pm 0.116$ |
| $K = 16$ | $-0.255 \pm 0.034$ | $-135.207 \pm 4.665$ | $-42148.287 \pm 1047.478$ | $-28.714 \pm 3.71$ | $-137.691 \pm 1.656$ | $-149.107 \pm 1.088$ | $-88.068 \pm 0.467$ | $-0.65 \pm 0.1$ |
| $K = 32$ | $-0.119 \pm 0.017$ | $86.106 \pm 5.989$ | $-19288.267 \pm 834.52$ | $-12.23 \pm 2.212$ | $-120.557 \pm 0.613$ | $-127.964 \pm 0.394$ | $-79.89 \pm 0.273$ | $-0.408 \pm 0.17$ |
| $K = 64$ | $-0.059 \pm 0.015$ | $269.566 \pm 7.832$ | $-8894.525 \pm 119.723$ | $-4.76 \pm 1.042$ | $-113.835 \pm 0.167$ | $-118.812 \pm 0.192$ | $-76.275 \pm 0.189$ | $-0.359 \pm 0.087$ |
| $K = 128$ | $-0.029 \pm 0.009$ | $390.33 \pm 5.427$ | $-5419.678 \pm 90.362$ | $-1.675 \pm 0.442$ | $-110.901 \pm 0.094$ | $-114.827 \pm 0.307$ | $-74.774 \pm 0.097$ | $-0.255 \pm 0.108$ |
| $K = 256$ | $-0.013 \pm 0.006$ | $477.162 \pm 4.998$ | $-3745.218 \pm 68.342$ | $-0.131 \pm 0.22$ | $-109.562 \pm 0.072$ | $-113.123 \pm 0.172$ | $-74.049 \pm 0.088$ | $-0.211 \pm 0.074$ |
| **ELBO** | | | | | | | | |
| $K = 8$ | $0.002 \pm 0.066$ | $-236.087 \pm 9.623$ | $-56122.917 \pm 5402.094$ | $-10.147 \pm 3.427$ | $-117.499 \pm 4.049$ | $-123.772 \pm 3.689$ | $-75.183 \pm 1.447$ | $-0.417 \pm 0.236$ |
| $K = 16$ | $-0.003 \pm 0.037$ | $-23.219 \pm 7.756$ | $-27397.2 \pm 1987.523$ | $-3.924 \pm 2.114$ | $-110.707 \pm 1.823$ | $-113.476 \pm 1.361$ | $-73.524 \pm 0.543$ | $-0.322 \pm 0.184$ |
| $K = 32$ | $0.003 \pm 0.018$ | $174.797 \pm 7.241$ | $-12110.983 \pm 1204.2$ | $-0.426 \pm 1.415$ | $-108.574 \pm 0.547$ | $-112.048 \pm 0.519$ | $-73.459 \pm 0.29$ | $-0.215 \pm 0.222$ |
| $K = 64$ | $0.001 \pm 0.015$ | $332.187 \pm 9.025$ | $-5360.819 \pm 306.407$ | $0.884 \pm 0.778$ | $-108.424 \pm 0.154$ | $-111.715 \pm 0.184$ | $-73.375 \pm 0.214$ | $-0.267 \pm 0.101$ |
| $K = 128$ | $0.001 \pm 0.009$ | $430.838 \pm 6.441$ | $-3624.167 \pm 168.119$ | $1.008 \pm 0.27$ | $-108.395 \pm 0.087$ | $-111.603 \pm 0.298$ | $-73.436 \pm 0.095$ | $-0.2 \pm 0.124$ |
| $K = 256$ | $0.002 \pm 0.006$ | $453.395 \pm 4.43$ | $-2811.161 \pm 106.68$ | $1.142 \pm 0.125$ | $-108.368 \pm 0.071$ | $-111.611 \pm 0.171$ | $-73.413 \pm 0.087$ | $-0.181 \pm 0.081$ |

### F.5 FURTHER ABLATION WITH NF-STYLE METHODS AND AFT

We further run both flow models (AFT and NFVI) on all possible target distributions (subject to OOM errors). Results can be found in Table 6.

### F.6 WALLCLOCK TIMES FOR $\ln Z$ CALCULATION

In order to calculate the average wall-clock time for $\ln Z$ calculation, we calculate the time it takes to draw 30 seeds of 2000 samples each from the methods below, and use these samples to calculate the mean and standard deviation of $\ln Z$ across 30 seeds.

### F.7 TRAINING TIME COMPARISONS TO SMC

In this section, we explore a total time comparison between our approach CMCD and SMC.

Table 5: $\ln Z$ **comparison.** $\ln Z$ values for a different number of steps $K$, experiments and methods. Not all methods could be evaluated on every $K$/experiment combination due to numerical instabilities or out-of-memory (OOM) problems.

| Dataset | Method | $K = 8$ | $K = 16$ | $K = 32$ | $K = 64$ | $K = 128$ | $K = 256$ |
|---|---|---|---|---|---|---|---|
| funnel | CMCD | -0.3037 ± 0.1507 | -0.223 ± 0.1041 | -0.1805 ± 0.0773 | -0.1085 ± 0.1143 | -0.0573 ± 0.0444 | -0.01928 ± 0.0641 |
| log_ionosphere | VI-DNF | -0.3768 ± 0.2157 | -0.3517 ± 0.1627 | -0.2919 ± 0.0999 | -0.6941 ± 0.6841 | -0.1947 ± 0.1325 | -0.2124 ± 0.0637 |
| | VI-NF | -0.206± 0.079 | -0.206± 0.082 | -0.206± 0.087 | -0.194± 0.101 | -0.182± 0.097 | -0.197± 0.099 |
| | AFT | -0.875± 0.543 | -0.395± 0.351 | -0.348± 0.192 | -0.271± 0.227 | -0.235± 0.139 | -0.196± 0.111 |
| gmm | CMCD | -0.1358 ± 0.0839 | -0.01331 ± 0.1292 | 0.0095 ± 0.0495 | 0.00736 ± 0.0477 | -0.0004 ± 0.0368 | -0.0081 ± 0.0520 |
| | VI-DNF | -0.3676 ± 0.6314 | -0.258 ± 0.412 | -0.4983 ± 0.3878 | -0.4449 ± 0.5379 | -0.4652 ± 0.3223 | -0.204 ± 0.6381 |
| | VI-NF | -0.355± 0.698 | -0.455± 0.258 | -0.064± 0.138 | -0.054± 0.15 | -0.066± 0.188 | -0.045± 0.177 |
| | AFT | -0.336± 0.372 | -0.006± 0.082 | 0.02± 0.068 | 0.02± 0.068 | -0.016± 0.042 | 0.001± 0.026 |
| lgcp | CMCD | 491.059 ± 3.553 | 498.147 ± 2.624 | 502.705 ± 2.482 | 506.045 ± 1.761 | 508.165 ± 1.553 | 509.43 ± 1.242 |
| | VI-DNF | 424.733 ± 5.858 | 424.719 ± 5.855 | 424.714 ± 5.861 | 424.719 ± 5.860 | 424.7 ± 5.869 | 424.705 ± 5.896 |
| | AFT | 126.651± 5.764 | 344.145± 23.95 | 191.613± 173.873 | 420.259± 91.43 | 480.126± 33.059 | 491.028± 8.057 |

Table 6: **ELBO comparison.** ELBO values for a different number of steps $K$, experiments and methods. Not all methods could be evaluated on every $K$/experiment combination due to numerical instabilities or out-of-memory (OOM) problems.

| Dataset | Method | $K = 8$ | $K = 16$ | $K = 32$ | $K = 64$ | $K = 128$ | $K = 256$ |
|---|---|---|---|---|---|---|---|
| seeds | CMCD | $-74.501 \pm 0.049$ | $-74.327 \pm 0.065$ | $-74.142 \pm 0.05$ | $-73.967 \pm 0.038$ | $-73.8 \pm 0.032$ | $-73.684 \pm 0.033$ |
| | VI-NF | $-73.563 \pm 0.013$ | $-73.547 \pm 0.012$ | $-73.574 \pm 0.012$ | $-73.58 \pm 0.014$ | $-73.621 \pm 0.014$ | $-73.675 \pm 0.014$ |
| | AFT | $-147.457 \pm 24.808$ | $-116.134 \pm 8.157$ | $-99.032 \pm 6.321$ | $-87.436 \pm 1.53$ | $-79.847 \pm 0.419$ | $-76.364 \pm 0.188$ |
| | CRAFT | $-146.973 \pm 1.531$ | $-94.2 \pm 0.505$ | $-80.985 \pm 0.344$ | $-76.555 \pm 0.175$ | $-74.979 \pm 0.143$ | $-74.225 \pm 0.133$ |
| log_ionosphere | CMCD | $-113.211 \pm 0.089$ | $-112.643 \pm 0.062$ | $-112.643 \pm 0.062$ | $-112.22 \pm 0.046$ | $-111.98 \pm 0.04$ | $-111.925 \pm 0.046$ |
| | VI-NF | $-111.903 \pm 0.022$ | $-111.902 \pm 0.022$ | $-111.892 \pm 0.017$ | $-111.881 \pm 0.017$ | OOM | OOM |
| | AFT | $-168.174 \pm 21.249$ | $-138.733 \pm 8.374$ | $-123.013 \pm 3.771$ | $-118.644 \pm 0.891$ | $-116.497 \pm 0.495$ | $-114.905 \pm 0.781$ |
| log_sonar | CMCD | $-112.274 \pm 0.124$ | $-110.904 \pm 0.111$ | $-110.459 \pm 0.106$ | $-109.503 \pm 0.075$ | $-109.608 \pm 0.066$ | $-109.25 \pm 0.052$ |
| | VI-NF | $-109.353 \pm 0.035$ | $-109.346 \pm 0.031$ | $-109.441 \pm 0.035$ | $-109.94 \pm 0.044$ | $-109.711 \pm 0.039$ | OOM |
| | AFT | $-203.249 \pm 12.506$ | $-148.357 \pm 8.096$ | $-129.772 \pm 3.057$ | $-121.653 \pm 2.505$ | $-114.911 \pm 0.331$ | $-112.021 \pm 0.182$ |
| lgcp | CMCD | $469.475 \pm 0.259$ | $479.246 \pm 0.237$ | $486.739 \pm 0.249$ | $492.745 \pm 0.239$ | $497.074 \pm 0.267$ | $499.708 \pm 0.236$ |
| | AFT | $75.896 \pm 0.863$ | $265.005 \pm 34.254$ | $62.898 \pm 200.991$ | $340.687 \pm 126.853$ | $417.916 \pm 50.35$ | $424.705 \pm 12.416$ |
| lorenz | CMCD | $-1180.797 \pm 0.184$ | $-1180.797 \pm 0.184$ | $-1176.514 \pm 0.154$ | $-1174.309 \pm 0.148$ | $-1172.453 \pm 0.153$ | $-1170.826 \pm 0.15$ |
| | VI-NF | $-1499.102 \pm 0.84$ | $-1471.798 \pm 0.582$ | $-1439.648 \pm 0.274$ | $-1433.536 \pm 0.316$ | OOM | OOM |
| brownian | CMCD | $-0.753 \pm 0.075$ | $-0.209 \pm 0.059$ | $0.153 \pm 0.045$ | $0.376 \pm 0.038$ | $0.578 \pm 0.046$ | $0.722 \pm 0.032$ |
| | VI-NF | $0.733 \pm 0.019$ | $0.797 \pm 0.018$ | $0.816 \pm 0.018$ | OOM | OOM | OOM |

As both methods are quite inherently different it is not immediately obvious how to carry out an insightful comparison. In order to do so we chose the LGCP which is our most numerically intense target and we phrase the following question:

"For how long do we have to train CMCD to outperform the best-run SMC"

For this, we look at our Figure 1 pane c) and we can see that at $K = 8$ CMCD already outperforms SMC at $K = 256$ with 2000 particles. So we choose these two approaches to compare to. In Table 9 is a brief comparison of total time calculations, note we have included tuning time for SMC which is akin to our training time as without tuning SMCs hyperparameters ELBOs and ln Z estimations were much worse. We can observe that the total runtime for training and sampling CMCD to reach a better $\ln Z$ value does not exceed the time required to tune SMC.

## G  REGULARISED IPF-TYPE EXPERIMENTS

For the purpose of completeness in this section, we empirically explore the regularised IPF-type objectives proposed in the main text. We explore a series of low-scale generative modelling experiments

| Method | Average Time (s) | Min Time (s) | Max Time (s) |
|---|---|---|---|
| CMCD (OD) | 9.665 | 5.592 | 21.475 |
| ULA | 9.204 | 4.673 | 20.721 |
| UHA | 9.427 | 5.588 | 20.263 |
| MCD | 9.204 | 4.673 | 20.721 |

Table 7: Wallclock times for evaluation in seconds.

| Mode | ULA | MCD | CMCD | DDS / PIS |
|---|---|---|---|---|
| Sampling | $\mathcal{O}(K \cdot (d + G(d)))$ | $\mathcal{O}(K \cdot (d + G(d)))$ | $\mathcal{O}(K \cdot (d + G(d) + N(d)))$ | $\mathcal{O}(K \cdot (N_2 + G(d) + N_1(d)))$ |
| ELBO | $\mathcal{O}(K \cdot (d + G(d)))$ | $\mathcal{O}(K \cdot (d + G(d) + N(d)))$ | $\mathcal{O}(K \cdot (d + G(d) + N(d)))$ | $\mathcal{O}(K \cdot (N_2 + G(d) + N_1(d)))$ |

Table 8: Sampling and loss calculation complexity across SDE based methods, $K$ represents the number of integration steps, $G(d)$ represents the cost of evaluating the score of the target and $N(d)$ for evaluating the drift/score networks both quantities are dimension dependant. PIS and DDS have an additional grad network cost $N_2$ which is dimension independant.

| Method | Train + Sample Time (min) | ln Z | ELBO |
|---|---|---|---|
| CMCD | $33.12 \pm 0.12$ | $491.059 \pm 3.553$ | $469.475 \pm 0.2589$ |
| SMC | $62.62 \pm 0.10$ | $477.162 \pm 4.998$ | $453.395 \pm 4.4300$ |

Table 9: Training+ Tuning + Sampling time comparisons for CMCD and SMC at comparable ln Z estimates.

where the goal is to retain generative modelling performance whilst improving the quality of the bridge itself (i.e. solving the SBP problem better).

Across our experiments, we use $D_{\mathrm{KL}}$ and let $\Gamma_0 = \Gamma_T = \mathrm{Leb}$, which can be simplified to the forward-backwards KL objective used in DNF (Zhang et al., 2014), see Appendix E.1.1. We use the Adam optimiser (Kingma & Ba, 2015) trained on 50,000 samples and batches of size 5000 following Zhang & Chen (2021). For the generative modelling tasks we use 30 time steps and train for 100 epochs whilst for the double well we train all experiments for 17 epochs (early stopping via the validation set) and 60 discretisation steps. Finally note we typically compare our approach with $\lambda > 0$ to DNF ($\lambda = 0$), with DNF initialised at the reference process, which we call DNF (EM Init), see Appendix E.4 for further details.

### G.1 2D TOY TARGETS – GENERATIVE MODELLING

Here we consider the suite of standard 2D toy targets for generative modelling explored in Zhang & Chen (2021) In contrast to Zhang & Chen (2021) we consider the SDE $\mathrm{d}\boldsymbol{Y}_t = -\sigma^2 \boldsymbol{Y}_t \, \mathrm{d}t + \sigma\sqrt{2} \, \mathrm{d}\boldsymbol{W}_t$ as the Schrödinger prior across methods. We parametrise DNF and our proposed approach with the same architectures for a fair comparison. Furthermore, we incorporate the drift of the above Schrödinger prior into DNF via parameterising the forward drift as in (75), partly motivated by Corollary E.2.

In order to assess the quality of the bridge we consider three different error metrics. Firstly we estimate $D_{\mathrm{KL}}$ between the Schrödinger prior and the learned forward process (i.e. $\mathbb{E}_{\boldsymbol{Y} \sim \overrightarrow{\mathbb{P}}^{\mu,a}} \left[ \frac{1}{2\sigma^2} \int_0^T \|a_t - f_t\|^2(\boldsymbol{Y}_t) \, \mathrm{d}t \right]$). Secondly, we evaluate $D_{\mathrm{KL}}(\overrightarrow{\mathbb{P}}^{\mu,f+\sigma^2\nabla\phi}, \overleftarrow{\mathbb{P}}^{\nu,f+\sigma^2\nabla\theta})$ to obtain a proxy error between the learned and target marginals. Finally, we estimate the cross entropy between $\overrightarrow{\mathbb{P}}_T^{\mu,a}$ and $\nu$ to assess how well the constraint at time $T$ is met.

In Table 10 we observe that similar values of $D_{\mathrm{KL}}$ are attained across both approaches in the tree, sierpinski, and checkerboard datatsets whilst achieving significantly lower values of the SBP loss across all training sets, and for tree, swirl and checkerboard validation datasets. At the same time, we can see that the cross-entropy errors are effectively the same across both approaches. Overall we can conclude that on the empirical measures over which we train our approach, we obtain a much better fit for the target Schrödinger bridge, and on the validation results we can see that we generalise to 3/5 datasets in improving the bridge quality whilst preserving the marginals to a similar quality.

### G.2 DOUBLE WELL – RARE EVENT

In this task we consider the double well potential explored in (Vargas et al., 2021b; Hartmann et al., 2013) where the Schrödinger prior is specified via the following overdamped Langevin dynamics $\mathrm{d}\boldsymbol{Y}_t = -\nabla_{\boldsymbol{Y}_t} U(\boldsymbol{Y}_t) \, \mathrm{d}t + \sigma \, \mathrm{d}\boldsymbol{W}_t$. The potential $U(\boldsymbol{y})$ typically models a landscape for which it is difficult to transport $\mu$ into $\nu$.

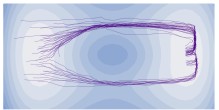 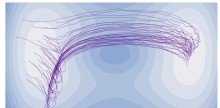 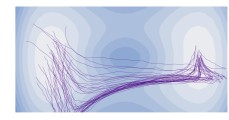 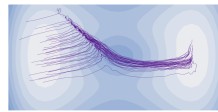

(a) $\lambda = 2$      (b) $\lambda = 0$ (EM Init)      (c) $\lambda = 0$ (Random Init)      (d) DNF (No Prior)

Figure 5: (a) our proposed regularised objective, (b) $\lambda$ set to 0 but using clever EM motivated initialisation, (c) $\lambda$ set to 0 with random initialisation of the forward drift, (d) for reference DNF with $f_t = 0$ (uninformative Schrödinger prior).

This is a notably challenging task as we are trying to sample a rare event and as noted by Vargas et al. (2021a) many runs would result in collapsing into one path rather than bifurcating. In Figure 5 we can observe how our proposed regularised approach (5a) is able to successfully transport particles across the well whilst respecting the potential, whilst both variants of DNF using the EM-Init for $\phi$ (5b) and random init (5c) fail to respect the prior as nicely and do not bifurcate, with the random init in particular sampling quite inconsistent trajectories. Finally for reference we train a DNF model with $f_t = 0$ and $\phi$ (5d) initialised at random to illustrate the significance of the initialisation of $\phi$.

### G.2.1 DOUBLE WELL POTENTIAL

We used the following potential (Vargas et al., 2021a):

$$U\left(\begin{pmatrix} x \\ y \end{pmatrix}\right) = \frac{5}{2}(x^2 - 1)^2 + y^2 + \frac{1}{\delta}\exp\left(-\frac{x^2 + y^2}{\delta}\right), \tag{85}$$

with $\delta = 0.35$, furthermore, we used the boundary distributions:

$$\mu \sim \mathcal{N}\left(\begin{pmatrix} -1 \\ 0 \end{pmatrix}, \begin{pmatrix} 0.0125 & 0 \\ 0 & 0.15 \end{pmatrix}\right), \quad \nu \sim \mathcal{N}\left(\begin{pmatrix} 1 \\ 0 \end{pmatrix}, \begin{pmatrix} 0.0125 & 0 \\ 0 & 0.15 \end{pmatrix}\right).$$

The Schrödinger prior is given by:

$$\mathrm{d}\boldsymbol{Y}_t = -\nabla_{\boldsymbol{Y}_t} U(\boldsymbol{Y}_t)\,\mathrm{d}t + \sigma\,\mathrm{d}\boldsymbol{W}_t, \tag{86}$$

with $\sigma = 0.4$. The terminal time is $T = 1$. Furthermore, we employ the same exponential discretisation scheme as in the generative modelling experiments.

| Target | Method | KL | | SBP Loss | | PINN Loss | | Cross Ent | |
|--------|--------|-----|-------|-----|-------|-----|-------|-----|-------|
| | | Val | Train | Val | Train | Val | Train | Val | Train |
| tree | $\lambda = 0.5$ | 1.67±0.02 | 1.40±0.01 | **47.84±1.58** | **42.31±1.52** | **0.06±0.00** | **0.05±0.00** | 2.87±0.01 | 2.80±0.01 |
| | DNF (EM Init) | 1.63±0.02 | 1.39±0.01 | 55.33±1.79 | 49.60±1.68 | 1.74±0.04 | 1.64±0.04 | 2.88±0.01 | 2.80±0.01 |
| olympics | $\lambda = 0.5$ | 2.95±0.06 | 0.12±0.01 | 39.30±0.90 | **25.24±0.62** | 0.26±0.01 | **0.10±0.00** | 2.49±0.01 | 2.77±0.02 |
| | DNF (EM Init) | **2.70±0.05** | 0.02±0.01 | 40.20±0.77 | 38.30±1.53 | 1.64±0.04 | 2.05±0.08 | **2.54±0.01** | 2.77±0.02 |
| sierpinski | $\lambda = 0.5$ | 2.31±0.01 | 2.20±0.01 | 28.54±1.49 | **26.67±0.90** | **0.04±0.00** | **0.03±0.00** | 2.82±0.01 | 2.83±0.01 |
| | DNF (EM Init) | 2.30±0.01 | 2.20±0.00 | 30.87±1.93 | 29.53±1.18 | 7.25±0.14 | 7.22±0.14 | 2.80±0.01 | 2.82±0.02 |
| swirl | $\lambda = 0.5$ | 15.67±0.29 | 1.95±0.03 | **121.81±1.94** | 40.24±1.74 | 1.01±0.03 | 0.14±0.00 | 2.97±0.01 | 2.69±0.02 |
| | DNF (EM Init) | **13.77±0.38** | 1.92±0.04 | 151.67±3.68 | 67.55±1.86 | 5.89±0.15 | 2.63±0.08 | 2.95±0.02 | 2.74±0.03 |
| checkerboard | $\lambda = 0.5$ | 4.79±0.01 | 4.70±0.01 | **34.47±0.80** | 33.70±0.91 | **0.03±0.00** | **0.02±0.00** | 2.82±0.00 | 2.81±0.01 |
| | DNF (EM Init) | 4.78±0.01 | 4.70±0.02 | 39.76±0.83 | 39.20±1.10 | 3.66±0.07 | 3.68±0.06 | 2.81±0.01 | 2.81±0.02 |

Table 10: Generative Modelling Results comparing DNF (Zhang & Chen, 2021) ($\lambda = 0$) to our PINN regualirsed approach with $\lambda = 0.5$. We observe that PINN regularisation obtains similar KL and Cross entropy losses to DNF whilst achieving lower distances to the prior.

### G.3 IMPLEMENTATION DETAILS

### G.3.1 NEURAL NETWORK PARAMETERISATIONS

Following Zhang & Chen (2021) and the recent success in score generative modelling we choose the following parameterisations:

$$a_t(\boldsymbol{x}) = f_t(\boldsymbol{x}) + \sigma^2 \nabla \phi(t, \boldsymbol{x}), \tag{87a}$$

$$b_t(\boldsymbol{x}) = f_t(\boldsymbol{x}) + \sigma^2 \nabla \phi(t, \boldsymbol{x}) - \sigma^2 s_\theta(t, \boldsymbol{x}), \tag{87b}$$

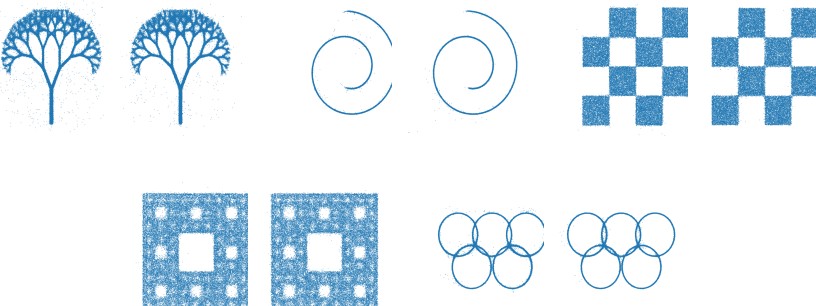

Figure 6: Generated samples trained by our approach ($\lambda = 0.5$) left and DNF ($\lambda = 0$) right. Qualitatively we can observe that both learned models have similarly matched marginals.

where $s_\theta$ is a score network (Song et al., 2021; De Bortoli et al., 2021; Zhang & Chen, 2021) and $\phi(t, \boldsymbol{x})$ is a neural network potential. We adapt the architectures proposed in Onken et al. (2021); Koshizuka & Sato (2023) to general activation functions. Note that these architectures allow for fast computation of $\Delta\phi$ comparable to that of Hutchinson's trace estimator (Grathwohl et al., 2019; Hutchinson, 1989).

Finally, we remark that the parametrisation in (87b) allows us to learn the score of the learned SDE and thus seamlessly adapt our approach to using the probability flow ODE (Song et al., 2021) at inference time.

### G.3.2 PINN Loss

For the PINN loss across all tasks, we sample the trajectories from $\boldsymbol{Y}_{0:T}^{\phi} \sim \overrightarrow{\mathbb{P}}^{\mu, \nabla\phi}$ and thus employ the same discretisation as used in the KL loss. However, we detach the trajectories $\boldsymbol{Y}_{0:T}^{\text{detach}(\phi)}$ before calculating the gradient updates in a similar fashion to Nüsken & Richter (2021).

