# OpenReview forum: "Transport meets Variational Inference: Controlled Monte Carlo Diffusions"
_ICLR.cc/2024/Conference — ICLR 2024 poster_

### Official Review · Reviewer_ndxa · 2023-10-31

**Soundness:** 3 good
**Presentation:** 3 good
**Contribution:** 2 fair
**Rating:** 6
**Confidence:** 4

**Summary:**

This paper present a framework to perform variational inference with a score-based algorithm. Specifically, it introduces HJB-regulariser to the diffusion normalizing flow process and provide some justification for the framework.

**Strengths:**

This connection between score-based algorithm and variational inference is an interesting topic. It remains open to make these score-based model really works in VI literature.

**Weaknesses:**

The idea is relatively direct and the whole motivation looks incremental.

The empirical results are not strong enough from both VI/transport side.

The ablation study of HJB-regulariser is weak.

**Questions:**

In Variational inference literature, the hidden dimension $z$ is a low dimensional manifold that can be interpretable factors? The latent of VAE uses smaller latent dimension than the data. However, in diffusion model/transport-based algorithms. The space of $x$ and $z$ are in the same dimension, which makes $z$ encode all the information of $x$ (and loses the ability to perform inference). How can you justify this compensation? What are the benefits of the (almost) lossless VI?
Moreover, it is not clear that how good is this “VI” compared to other VAEs, such as image generation, latent variable inference. The current setting is relatively toy.


From the transport side, although it is understandable that Figure 2 is more well behaved than non-regularized transport, we still found that Figure 3 is good enough in real practice. Can you find an example that the non-regularized way fail?

In general, I do believe that the HJB-regulariser is useful in some cases, but given the current justifications and evidence, the significance is not clear at this moment.

---

> ### Author Response · Authors · 2023-11-11
> **Review refers to experiments from prior workshop paper, without acknowledging sampling experiments in the main section of this paper.**
>
> Dear Reviewer ndxa,
>
> We thank you for the time and effort taken to review our manuscript and for the many careful and insightful comments presented.
>
> We believe that there might have been a small confusion and you have accidentally reviewed the wrong manuscript. It might be possible that you wrote your review based on a much earlier workshop version of this paper that was publicly available online before the start of the review period. Alternatively it may have been the case that you did not see the main contributions of our paper being a new sampling algorithm, which our extensive experimental section focuses on, instead providing comments on some additional toy 2D experiments that are only in the appendix.
>
> We believe this may be the case since the focus of your feedback is oriented to experiments that are not present in the main manuscript that we have submitted on openreview.
>
> Furthermore, we sincerely request the reviewer to revisit the experimental section of our openreview submission, where we have high dimensional non-toy experiments on sampling from densities, which we believe is the main focus of this work, and quite different from the workshop submission.
>
> To explain why we believe this is the case we would like to point to the following comments and feedback the reviewer has provided:
>
> 1. “The ablation study of HJB-regulariser is weak.”
> 2. “that Figure 2 is more well behaved than non-regularized transport, we still found that Figure 3 is good enough in real practice.”
>
> If you revisit the experiments in the main pdf, you will notice we have experiments on 8 different datasets (target densities) ranging in dimensions from 2 to 1600, and averaging a dimension of ~ 231 which is quite high-dimensional for our focus of sampling from unnormalised densities. We sincerely request that the reviewer draw conclusions based on these 8 experiments in the main text where we compare them to 7 different competing methods.
>
> The reviewer's feedback refers only to the toy 2D experiments that are at the end of the appendix, which we are happy to address but we would reiterate that our main experimental section on sampling experiments is the focus of the paper, not the final appendix section, which we don't mention in the main text and is mainly included as an interesting aside.
>
> We would further like to remark that the HJB-regulariser is not our main proposed method / algorithm and as the title of the paper motivates, the controlled Monte Carlo diffusion sampler CMCD (which we have ablated thoroughly in our experiments) is the main contribution of this work. We apologise if this was a point of confusion. The HJB regulariser is mostly a conceptual/theoretical remark that is in line with our overarching framework whilst our experimental focus is on our proposed method for sampling (CMCD). Therefore, the minor empirical ablations on the HJB regularizer are left to the appendix and are by no means the central contribution of our paper.
>
> You will see in our main manuscript submitted here you can find quite a comprehensive array of experiments for sampling comparing our CMCD method to prior work (note these are not toy experiments):
>
> 1. Ionosphere Logistic Regression Target.       Dimension = 35
> 2. Log Sonar Logistic Regression Target.        Dimension = 61
> 3. Brownian Motion Posterior Target.               Dimension = 32
> 4. Lorenz Model Posterior Target.                    Dimension = 90
> 5. Seeds (Random effect regression) Target.   Dimension = 26
> 6. Log Gaussian Cox Process Target.              Dimension = 1600
> 7. Neals Funnel Target.                                     Dimension = 10
> 8. Toy Difficult GMM [1]                                     Dimension = 2
>
> Note that across all these targets we significantly outperform an array of sampling methods that we compare to:
>
> 1. ULA [2]
> 2. UHA [3]
> 3. MCD-UHA [4,5]
> 4. MCD-UHA [4,5]
> 5. SMC          [6]
> 6. PIS            [7]
> 7. DDS          [8]
>
>
> We would also like to highlight that these experiments are standard baselines for sampling used consistently across many prior works focused on sampling and partition function estimation [1,2,3,4,5,6,7,8]. Note that for sampling, unlike generative modelling, dimensions >= 50 are already considered to be very high dimensional, as many of the provided references suggest.
>
> Finally, to further the extensive nature of these experiments we would like to cite reviewer LiYc’s remark:
>
> “Extensive experiments show that the CMCD method outperforms existing methods in relatively high dimensional examples.”
>
> We hope the reviewer can agree that these experiments are significantly different and much more extensive than the toy 2D experiments from the appendix. We sincerely request the reviewer to reconsider the comprehensive sampling experiments we ran to support our new sampling algorithm as the core contribution, and we would be grateful if they could reevaluate their score accordingly.

---

> ### Author Response · Authors · 2023-11-11
> **Part II - Refferences**
>
> [1] Arbel, M., Matthews, A. and Doucet, A., 2021, July. Annealed flow transport monte carlo. In International Conference on Machine Learning (pp. 318-330). PMLR.
>
> [2] Girolami, M. and Calderhead, B., 2011. Riemann manifold langevin and hamiltonian monte carlo methods. Journal of the Royal Statistical Society Series B: Statistical Methodology, 73(2), pp.123-214.
>
> [3] Geffner, T. and Domke, J., 2021. MCMC variational inference via uncorrected Hamiltonian annealing. Advances in Neural Information Processing Systems, 34, pp.639-651.
>
> [4] Geffner, T. and Domke, J., 2023, April. Langevin Diffusion Variational Inference. In International Conference on Artificial Intelligence and Statistics (pp. 576-593). PMLR.
>
> [5] Doucet, Arnaud, Will Grathwohl, Alexander G. Matthews, and Heiko Strathmann. "Score-based diffusion meets annealed importance sampling." Advances in Neural Information Processing Systems 35 (2022): 21482-21494.
>
> [6] Del Moral, P., Doucet, A. and Jasra, A., 2006. Sequential monte carlo samplers. Journal of the Royal Statistical Society Series B: Statistical Methodology, 68(3), pp.411-436.
>
> [7] Zhang, Q. and Chen, Y., 2021. Path integral sampler: a stochastic control approach for sampling. arXiv preprint arXiv:2111.15141.
>
> [8] Vargas, F., Grathwohl, W. and Doucet, A., 2023. Denoising diffusion samplers. arXiv preprint arXiv:2302.13834.

---

> > ### Comment · Reviewer_ndxa · 2023-11-11
> >
> > Thank you for your quick response.
> >
> > What I am interested in is the comparison with DNF and ablation study about different parameters and settings. Although your main experiments show some advantages against baselines, it is not clear that how does your proposed algorithm help the performance.
> >
> > I do believe that there needs some ablation study to analyse different parts of your algorithms, rather than just running it and get the SOTA conclusion.

---

> ### Author Response · Authors · 2023-11-11
> **Our paper is not about DNF our empirical contribution is a new sampler CMCD (unrelated to the HJB reguralizers)**
>
> Dear reviewer ndxa,
>
> Thank you for the prompt response. It is possible you are looking at a different paper, as DNF is not applicable to our task
>
> Our paper is focused on sampling from unnormalized densities. DNF is a method for generative modeling when you have access to data, this method is not applicable or relevant to the sampling task that we consider (where we do not have access to data or samples).
>
> We cannot compare/do ablations to DNF as DNF is focused on a different task tangential to our paper. Could you please check the experiment section here https://openreview.net/pdf?id=PP1rudnxiW (this is the pdf for this paper on openreview).
>
> Our proposed algorithm CMCD has nothing to do with the HJB regularisation remark, and this is what we explore empirically. We do a careful ablation of its performance across different timesteps, furthermore, in Appendix F.3 you can find further ablations.

---

> ### Author Response · Authors · 2023-11-13
> **Variational Inference extends beyond learning lower dimensional representations, it is also used in the very well-studied application of sampling from densities (e.g. MCMC styled tasks) + Revised Version clarifying task and contributions**
>
> Dear Reviewer ndxa,
>
> We would like to thank you for the time you spent reading our work and the feedback you have provided. In what follows we aim to **address your main question** and detail how our **revised version** clarifies this.
>
> We have uploaded a revised version of the manuscript, which introduces **the sampling problem**, that is the task of sampling from an unnormalised density and estimating its normalising constant. This is a seminal task in both statistics and machine learning [1-8] that predates generative modelling.
>
> We further reiterate in the revised version that our task is focused on **sampling**, not generative modelling. The goal here is as follows; given we have access to a probability density function of the form:
>
> $$p(x) = \frac{e^{-U(x)} }{\int  e^{-U(x)} dx}$$
>
> Where $U(x)$ is a function that we can evaluate and differentiate pointwise, however we are unable to evaluate the normalising constant $\int  e^{-U(x)} dx$.
>
> Given such a distribution $p(x)$, the goal is to learn or design a model that can generate samples that are distributed according to $p(x)$, notice **unlike generative modelling we do not have access to any data or samples**. One way to do so is via minimising KL w.r.t to a parametric distribution $q_{\theta}(x)$:
>
> $$\arg_{\theta}\min D_{KL} (q_{\theta}(x) || p(x))$$
>
> Note that the above KL can only be computed up to a constant in practice ($\int  e^{-U(x)} dx$), and thus an ELBO is what we typically minimise. This family of approaches is typically also **referred to as Variational Inference**, and is a superset of techniques involving approximating intractable distributions [13], VAEs are simply a particular, unrelated sub-instance of VI and are not particularly applicable to our task. Therefore, **there are no low-dimensional latents in this formulation**.
>
> Typically one wants 2 things from $q_\theta$:
>
> 1. It is easy to sample from numerically (not intractable)
> 2. It is a very flexible distribution so that $D_{KL} (q_{\theta}(x) || p(x))$ can be brought very close to $0$.
>
> A common way this is achieved is through [9,14] applying a series of tractable transformations to a simple Gaussian, e.g. :
>
> \begin{align}
> z_0 \sim \mathcal{N}(0,I) \\
> \end{align}
> \begin{align}
> z_1  \sim  \mathcal{N}(f_{\theta_1}(z_0), \sigma_1^2I) \\
> \end{align}
> \begin{align}
> \vdots \\
> \end{align}
> \begin{align}
> x = z_K  \sim  \mathcal{N}(f_{\theta_{K-1}}(z_{K-1}), \sigma_{K-1}^2I) ,
> \end{align}
> the resulting $q_\theta(x)$ is then a very flexible distribution that is easy to sample from.
>
> Notice there is no need for the latents to be lower-dimensional as our goal is to minimise $D_{KL} (q_{\theta}(x) || p(x))$ so that we can sample $p(x)$. We do not care about $z_0$ being an interpretable quantity, we just care about it being a tractable quantity. This is a very common use case of VI [9,10,11,12].
>
> Also, notice that these flavours of deep latent models with equal latent dimensions have been studied quite extensively already in the context of both neural ODEs and SDEs [14,15,16].
>
> Our main contribution is a novel algorithm (CMCD) that allows us to parametrise $q_{\theta}(x)$ with a tractable and learnable SDE that starts at a Gaussian $z \sim \mathcal{N}(0,I)$ and then approximately (with guarantees that we prove) transports the Gaussian to the target distribution $p(x)$.
>
> We hope this answers your question on VI and equal dimensional latents:
>
> > “In Variational inference literature, the hidden dimension is a low dimensional manifold that can be interpretable factors? The latent of VAE uses smaller latent dimension than the data... ”
>
> We look forward to any further doubts you might have and are happy to provide any further information. In particular, we hope now that the task and focus of this paper are clear and you might be able to appreciate our sampling experiments and ablations better, as they are by no means toy examples, and are a comprehensive ablation on standard benchmarks, targets and baselines across sampling literature.
>
>
> [9] Rezende, D. and Mohamed, S., 2015, June. Variational inference with normalizing flows. ICML
>
> [10] Doucet, Arnaud, Will Grathwohl, Alexander G. Matthews, and Heiko Strathmann. "Score-based diffusion meets annealed importance sampling." Nuerips
>
> [11] Geffner, T. and Domke, J., 2023, April. Langevin Diffusion Variational Inference. In AISTATS
>
> [12] Wu, H., Köhler, J. and Noé, F., 2020. Stochastic normalizing flows. Neurips, 33
>
> [13] Blei, D.M., Kucukelbir, A. and McAuliffe, J.D., 2017. Variational inference: A review for statisticians. Journal of the American Statistical Association,
>
> [14] Tzen, B. and Raginsky, M., 2019. Neural stochastic differential equations: Deep latent Gaussian models in the diffusion limit.
>
> [15] Chen, R.T., Rubanova, Y., Bettencourt, J. and Duvenaud, D.K., 2018. Neural ordinary differential equations. Neurips
>
> [16] Xu, W., Chen, R.T.,  et al. 2022, May. Infinitely deep Bayesian neural networks with stochastic differential equations.

---

> > ### Author Response · Authors · 2023-11-14
> > **VI , The Sampling Problem and Generative modelling:  Differences - Summary (Our paper is focused on sampling)**
> >
> > Dear Reviewer
> >
> > To further clarify our contribution, we want to re-emphasize that **our main task and experiments are not generative modeling** and we **do not have access to samples/data from the target distribution**, instead, we are focused on **the sampling problem**. To clarify things a bit further for the reviewer we have made the following ASCII table to contrast these two different (yet related) tasks:
> >
> > |                     | Sampling                                                                        | Generative Modelling                                                  |
> > |---------------------|---------------------------------------------------------------------------------|-----------------------------------------------------------------------|
> > | Has access to       |  Log density $\ln \hat{p}(x)$ (up to a constant)   and $\nabla \ln \hat{p}(x)$        | Samples/dataset from underlying Distribution   i.e. $p_{\text{data}} (x) = \frac{1}{N}\sum_{i=1}^N \delta_{x_i}(x)$                       |
> > | Typical Approaches  | MCMC methods using $\nabla \ln \hat{p}(x)$, VI methods minimising reverse KL i.e. $\min_q KL(q\|\|p)$ | Minimising CE/Forward KL i.e. $\min_p KL(p_{\text{data}}\|\|p)$, VAEs, score matching, GANs, etc |
> > | Tasks / Goals              | Estimating the normalsing constant $\ln Z = \int \hat{p}(x) dx$, generating samples from $p(x)$ | Mostly focused on generating samples
> >
> >
> > Note that sampling as a task still remains much more unsolved than gen-modelling, and the dimensions considered "high" or challenging (e.g. $\geq 50$) are not on the same scale as generative modeling, as this is a more challenging task (due to not having access to samples).
> >
> > Also, notice that **unlike VAEs** (in their most traditional setting) which aim to generate observed data from a lower dimensional latent representation our experimental goal in this work is to train an SDE that can sample from a prespecified density (like in MCMC), and we explore a **very standardized benchmark of target densities many of them arising from real-world applications (e.g. bayesian inference, geospatial inference, ...)**.
> >
> > We hope that this has clarified our task, experiments and setup to the reviewer, and look forward to their response.

---

> > > ### Comment · Reviewer_ndxa · 2023-11-14
> > > **Thank you for your feedback**
> > >
> > > First, we have to highligh that the flow-based algorithm can indeed perform the sampling task as you mentioned [1]. It is important to compare the baseline. Besise, some concurrent work also add similar baselines (VI-NF) [2]. Due to the similar formulation of DNF, it is also important to consider the direct baseline VI-DNF and other VI-NFs. Moreover, some other baselines such as Underdamped Langevin, [3] is missing. Also, it is not fair to compare $K$ with ULA, as you introduce additional parameterization in the model. Even with other methods, the complexity may not directly compared. A wall-clock time and computation overhead comparison would be helpful.
> > >
> > > Finally, as Reviewer yH6G mentioned, the presentation of the paper needs to be revised extensively. Too many unnecessary propositions mislead the readers and have negative impacts for the work.
> > >
> > >
> > > [1] variational inference with normalizing flows
> > > [2] Diffusion Generative Flow Samplers: Improving learning signals through partial trajectory optimization
> > > [3] Continual repeated annealed flow transport monte carlo

---

> ### Author Response · Authors · 2023-11-16
> **Added requested comparisons to NF, DNF-VI , AFT + Significantly Improved Presentation Part I**
>
> Dear Reviewer ndxa,
>
> Thank you so much for engaging with us and for the very helpful comments that are significantly improving our manuscript.
>
> Upon your recommendation, we have added the following additional experiments and ablations (see Appendixes F.2. - F.5. in the revised manuscript https://openreview.net/pdf?id=PP1rudnxiW):
>
> 1. A VI Normalising flow baseline (VI-NF) on the GMM, Funnel and LGCP targets.
> 2. Comparisons to VI-DNF across the GMM, Funnel and LGCP targets. As expected (from our theory but also comparisons to DDS and PIS) **we significantly outperform VI-DNF.**
> 3. Comparisons to AFT, across all datasets with tractable distributions, where outperform on **6/8 targets and remain competitive on 2**.
> 4. Comparisons to **VI-NF** across all datasets , where we outperform on 5/8 datasets and remain competitive on 1. The 2 target densities **VI-NF** performs better are on the more lower-moderate dimensional end and are unimodal.
> 6. We compare to **Underdampened Langevin Annealing** comparisons in the main section of the paper **(UHA)** is the acronym we use for it.
> 7. Wallclock times comparing different methods. Note all methods are trained for the same amount of iterations and parametrised (if needed) by neural networks of the same size across methods.
>
> These ablations can now be found in Appendix F.4. - F.6 colored in teal, **Tables 5,6 and z** (apologies if color is unhelpful). It is **important** to note that for the VI-NF we match $K$ to the number of IAF flows which quickly results in **VI-NF** having much more parameters than our modest drift network used in CMCD.
>
> Now we would like to address your questions:
>
> > Moreover, some other baselines such as Underdamped Langevin, is missing
>
> **We already compare to Unaderdampened Langevin Dynamics**. Underdampened Langevin Dynamics is in fact Hamiltonian dynamics + Brownian noise, and we use the acronym UHA (Unadapted Hamiltonian Annealing) to refer to it. If you look carefully at our plots in Figure 1 a) and b), you will notice there is a column for Underdampened (UD) methods and a column for Overdamped methods (OD). UHA corresponds to Underdampened Langevin dynamics. We further explain the OD and UD acronyms in the experiment section.
>
> > Also, it is not fair to compare $K$  with ULA, as you introduce additional parameterization in the model
>
> I think there is a tiny misconception here. Our algorithm provides us with an optimisation objective that basically allows us to train / improve on ULA in way that variance is reduced and we can obtain better estimators for ln Z . **Notice when $\phi=0$, our approach is in fact exactly the same as ULA**. Our propositions and lemma provide us with an objective to improve on top of ULA. As the reviewer pointed out, the $\phi$ parametrisation will increase the compute time, but correspondingly, there is **significant improvement in performance over ULA** that cannot be achieved otherwise. We trade off some extra compute for much lower variance and better performance.
>
> Rather than an unfair comparison, the ULA line serves as a baseline to see how much our approach improves from its starting point. However, for transparency, we have added wall-clock times for ln Z estimation, to showcase the additional compute that is required for this significant improvement in performance.
>
> We have ensured the other methods that are neural SDEs have the same number of parameters, and their computational complexity is the same as CMCD for sampling, However, we have also added wallclock times for some these (e.g. MCD).
>
> > Finally, as Reviewer yH6G mentioned, the presentation of the paper needs to be revised extensively
>
> We wholeheartedly agree with the reviewer that the initial presentation needed extensive revision, and we have significantly changed our presentation to address reviewer yH6G’s remarks. This can be seen in the revised pdf (https://openreview.net/pdf?id=PP1rudnxiW) ). Additionally, we wrote a 2nd alternative draft for the the introduction with a different presentation flow so that reviewers had an extra option to choose from (https://anonymous.4open.science/r/CMCD-6BF6/rebuttal_version.pdf ), please note this version does not contain the updated experiments the only bit to read here is the different introduction (presented differently).
>
> We would like to remark **that reviewer yH6G was very positive about our new presentation**, and they **increased their scores** from a 3 to an 8, stating that we addressed their remarks:
>
> > I sincerely thank the authors for their comments and their extensive modifications. I think the presentation is much clearer now and I also now finally understand the greater context  … I believe with this modification the paper has reached finally a place where it is broadly significant and I have modified my original rating to reflect this..
>
> We would love to hear your feedback on our revised ablations and presentations, and we hope that this has addressed some of your concerns and questions.

---

> > ### Author Response · Authors · 2023-11-16
> > **Added requested comparisons to NF, DNF-VI , AFT + Significantly Improved Presentation - Part II**
> >
> > To ease discussion with the reviewer we will provide our new ablations here:
> >
> > # DNF + flows Comparisons on GMM, Funnel and LGCP
> >
> >
> > | Dataset | $K$  | 8 | 16 | 32 | 64 | 128 | 256 |
> > | :--- | :--- | :--- | :--- | :--- | :--- | :--- | :--- |
> > | Funnel | CMCD | -0.3037 $\pm$ 0.1507 | -0.223 $\pm$ 0.1041 | **-0.1805 $\pm$ 0.0773** | **-0.1085 $\pm$ 0.1143**| **-0.0573 $\pm$ 0.0444** | **-0.01928 $\pm$ 0.0641**|
> > |  | VI-DNF | -0.3768 $\pm$ 0.2157| -0.3517 $\pm$ 0.1627| -0.2919 $\pm$ 0.0999| -0.6941 $\pm$ 0.6841 | -0.1947 $\pm$ 0.1325| -0.2124 $\pm$ 0.0637 |
> > |  | VI-NF |  **-0.206$\pm$ 0.079** |  **-0.206$\pm$ 0.082** |  -0.206$\pm$ 0.087 |  -0.194$\pm$ 0.101  |  -0.182$\pm$ 0.097 |  -0.197$\pm$ 0.099  |
> > |  | AFT |  -0.875$\pm$ 0.543 |  -0.395$\pm$ 0.351 |  -0.348$\pm$ 0.192 |  -0.271$\pm$ 0.227  |  -0.235$\pm$ 0.139 |  -0.196$\pm$ 0.111  |
> > | GMM | CMCD | **-0.1358 $\pm$ 0.0839** | -0.01331 $\pm$ 0.1292 | **0.0095 $\pm$ 0.0495**| **0.00736 $\pm$ 0.0477**| **-0.0004 $\pm$ 0.0368**| -0.0081 $\pm$ 0.0520|
> > |  | VI-DNF | -0.3676 $\pm$ 0.6314| -0.258 $\pm$ 0.412| -0.4983 $\pm$ 0.3878 | -0.4449 $\pm$ 0.5379| -0.4652 $\pm$ 0.3223 | -0.204 $\pm$ 0.6381|
> > |  | VI-NF |  -0.355$\pm$ 0.698 |  -0.455$\pm$ 0.258 |  -0.064$\pm$ 0.138 |  -0.054$\pm$ 0.15  |  -0.066$\pm$ 0.188 |  -0.045$\pm$ 0.177  |
> > |  | AFT |  -0.336$\pm$ 0.372 |  **-0.006$\pm$ 0.082** |  0.02$\pm$ 0.068 |  -0.016$\pm$ 0.042  |  -0.003$\pm$ 0.029 |  **0.001$\pm$ 0.026**  |
> > | LGCP | CMCD | **491.059 $\pm$ 3.553** | **498.147 $\pm$ 2.624** | **502.705 $\pm$ 2.482** | **506.045 $\pm$ 1.761** | **508.165 $\pm$ 1.553** | **509.43 $\pm$ 1.242** |
> > |  | VI-DNF | 424.733 $\pm$ 5.858| 424.719 $\pm$ 5.855| 424.714 $\pm$ 5.861| 424.719 $\pm$ 5.860| 424.7 $\pm$ 5.869| 424.705 $\pm$ 5.896|
> > | | AFT |  126.651$\pm$ 5.764 |  344.145$\pm$ 23.95 |  191.613$\pm$ 173.873 |  420.259$\pm$ 91.43  |  480.126$\pm$ 33.059 |  491.028$\pm$ 8.057  |
> >
> > # Flows rest of the targets
> >
> > | Dataset | Method |         $K=8$         |         $K=16$        |         $K=32$        |         $K=64$        |        $K=128$        |        $K=256$       |
> > |------------------|-----------------|:---------------------:|:---------------------:|:---------------------:|:---------------------:|:---------------------:|:--------------------:|
> > | seeds            | CMCD            |  $-74.501 \pm 0.049$  |  $-74.327 \pm 0.065$  |   $-74.142 \pm 0.05$  |  $-73.967 \pm 0.038$  |   $-73.8 \pm 0.032$   |  $-73.684 \pm 0.033$ |
> > |                  | VI-NF           |  $-73.563 \pm 0.013$  |  $-73.547 \pm 0.012$  |  $-73.574 \pm 0.012$  |   $-73.58 \pm 0.014$  |  $-73.621 \pm 0.014$  |  $-73.675 \pm 0.014$ |
> > |                  | AFT             | $-147.457 \pm 24.808$ |  $-116.134 \pm 8.157$ |  $-99.032 \pm 6.321$  |   $-87.436 \pm 1.53$  |  $-79.847 \pm 0.419$  |  $-76.364 \pm 0.188$ |
> > | log\_ionosphere  | CMCD            |  $-113.211 \pm 0.089$ |  $-112.643 \pm 0.062$ |  $-112.643 \pm 0.062$ |  $-112.22 \pm 0.046$  |   $-111.98 \pm 0.04$  | $-111.925 \pm 0.046$ |
> > |                  | VI-NF           |  $-111.903 \pm 0.022$ |  $-111.902 \pm 0.022$ |  $-111.892 \pm 0.017$ |  $-111.881 \pm 0.017$ |          OOM          |          OOM         |
> > |                  | AFT             | $-168.174 \pm 21.249$ |  $-138.733 \pm 8.374$ |  $-123.013 \pm 3.771$ |  $-118.644 \pm 0.891$ |  $-116.497 \pm 0.495$ | $-114.905 \pm 0.781$ |
> > | log\_sonar       | CMCD            |  $-112.274 \pm 0.124$ |  $-110.904 \pm 0.111$ |  $-110.459 \pm 0.106$ |  $-109.503 \pm 0.075$ |  $-109.608 \pm 0.066$ |  $-109.25 \pm 0.052$ |
> > |                  | VI-NF           |  $-109.353 \pm 0.035$ |  $-109.346 \pm 0.031$ |  $-109.441 \pm 0.035$ |  $-109.94 \pm 0.044$  |  $-109.711 \pm 0.039$ |          OOM         |
> > |                  | AFT             | $-203.249 \pm 12.506$ |  $-148.357 \pm 8.096$ |  $-129.772 \pm 3.057$ |  $-121.653 \pm 2.505$ |  $-114.911 \pm 0.331$ | $-112.021 \pm 0.182$ |
> > | lgcp             | CMCD            |  $469.475 \pm 0.259$  |  $479.246 \pm 0.237$  |  $486.739 \pm 0.249$  |  $492.745 \pm 0.239$  |  $497.074 \pm 0.267$  |  $499.708 \pm 0.236$ |
> > |                  | AFT             |   $75.896 \pm 0.863$  |  $265.005 \pm 34.254$ |  $62.898 \pm 200.991$ | $340.687 \pm 126.853$ |  $417.916 \pm 50.35$  | $424.705 \pm 12.416$ |
> > | lorenz           | CMCD            | $-1180.797 \pm 0.184$ | $-1180.797 \pm 0.184$ | $-1176.514 \pm 0.154$ | $-1174.309 \pm 0.148$ | $-1172.453 \pm 0.153$ | $-1170.826 \pm 0.15$ |
> > |                  | VI-NF           |  $-1499.102 \pm 0.84$ | $-1471.798 \pm 0.582$ | $-1439.648 \pm 0.274$ | $-1433.536 \pm 0.316$ |          OOM          |          OOM         |
> > | brownian         | CMCD            |   $-0.753 \pm 0.075$  |   $-0.209 \pm 0.059$  |   $0.153 \pm 0.045$   |   $0.376 \pm 0.038$   |   $0.578 \pm 0.046$   |   $0.722 \pm 0.032$  |
> > |                  | VI-NF           |   $0.733 \pm 0.019$   |   $0.797 \pm 0.018$   |   $0.816 \pm 0.018$   |          OOM          |          OOM          |          OOM         |

---

> ### Author Response · Authors · 2023-11-16
> **Added requested comparisons to NF, DNF-VI , AFT + Significantly Improved Presentation - Part III (Wallclock + References)**
>
> # Wallclock Times
>
> Here we provide comparisons for wallclock times for sampling and lnZ estimation (the number of samples kept the same across methods, more precise details can be found in our appendix). Notice how the overhead from our proposed approach whilst higher is still not a large computational overhead.
>
>
> | Method | Average Time (s) | Min Time (s) | Max Time (s) |
> | :--- | :--- | :--- | :--- |
> | CMCD (OD) | 9.665 | 5.592 | 21.475 |
> | ULA | 9.204 | 4.673 | 20.721 |
> | UHA | 9.427 | 5.588 | 20.263 |
> | MCD | 9.204 | 4.673 | 20.721 |
>
> # Method Descriptions + References
>
> * **AFT**: Anealed Flow Transport Montecarlo [1],  **a more modern and scalable flow method than NFVI**
> * **VI-NF**: Stacked Inverse autoregressive flows (IAF) for variational inference (uses the codebase of [1]) . **We match $K$ to the number of stacked IAF layers**.
> * **DNF-VI**: Diffusion Normalising Flow Baseline adapted to VI **suggested by the reviewer.**
>
> **Note:** AFT was unable to train successfully (became unstable and did not converge) on the Lorenz and Brownian targets. We sweeped over learning rates and hyperparameters to try and stabilise this but were not able to produce reasonable results.
>
> [1] Arbel, M., Matthews, A. and Doucet, A., 2021, July. Annealed Flow Transport Monte Carlo. In International Conference on Machine Learning (pp. 318-330). PMLR.

---

> ### Author Response · Authors · 2023-11-20
> **Feedback on revised presentation + new baselines**
>
> Dear reviewer ndxa,
>
> With the revision of our manuscript for which reviewer **yH6G** stated **“has reached finally a place where it is broadly significant“** and **“the presentation is much clearer”**, we hope to have addressed any concerns you may have had regarding the presentation of our manuscript.
>
>  Furthermore, as per your suggestion, we have implemented new baseline algorithms **(NF-VI, DNF-VI, AFT)** and compared our algorithm against them on a suite of target distributions, showing the strong performance of our algorithm.
>
> Finally, we also provide wallclock times of evaluation of our baselines and computational complexity comparisons. We look forward to engaging with you further during the rebuttal period to address any concerns you might have remaining and hope to hear your feedback and review of our revised manuscript.

---

> > ### Comment · Reviewer_ndxa · 2023-11-21
> >
> > Thanks for your feedback. I have reevaluate the rating for the current version. The added VI-DNF/VI-NF experiments look good. However, it looks that some comparisons are missing in the table, so I would suggest to include them for completeness. Moreover, I still think the clarity can be further improved. For example, some important part such as minimising Eq 24 in Algorithm might be further elaborated, which I do think is an important part for this paper.

---

> > > ### Author Response · Authors · 2023-11-22
> > >
> > > Dear Reviewer,
> > >
> > > Thank you for your valuable and continued input, we have expanded Equation 24 and added Algorithm 2 (in the appendix), which details how our approach is trained, we hope this improves clarity around this.
> > >
> > > We are currently running the remaining experiments and will add the results once they are available.

---

### Official Review · Reviewer_nqRv · 2023-10-31

**Soundness:** 4 excellent
**Presentation:** 3 good
**Contribution:** 4 excellent
**Rating:** 8
**Confidence:** 3

**Summary:**

The authors propose a novel framework for generative modelling and Bayesian computation that bridges variational inference and optimal transport using the theory of diffusion models. Starting from the Schrödinger problem and iterative proportional fitting, the authors draw a connection to the EM-algorithm, and finally present a novel regularized diffusion objective with a unique minimizer. They validate their method on several experimental benchmarks where it outperforms competing state-of-the-art methods.

**Strengths:**

- The paper is as far as I (the reviewer) can judge highly original and will be greatly valued by the community.
- The paper is well written and generally clear. The contextualization of their work with respect to other frameworks is well done. It unifies and generalizes several previously introduced frameworks, such as Monte Carlo diffusions or unadjusted Langevin annealing. The motivation and lead-up to their final results is in my opinion very convincing.
- The proposed method outperforms other state-of-the-art-methods on a variety of experimental evaluations. The experimental section is sufficiently exhaustive wrt compared data sets and competing methods.

**Weaknesses:**

I don't have major comments regarding weaknesses other than that the source code for the experimental validation needs to be provided in the supplement and not only via some possibly non-permanent web link.

**Questions:**

- In my opinion the paper could benefit from a shorter introduction and motivation and instead an extended experimental section with method limitations, experimental details, computational trade-offs, etc. for more applied audiences. Since this is likely not possible due to page limits, an extended treatment in the appendix would be very welcome.
- What are the training times of the method, for instance, in comparison to running SMC?
- How were the neural networks architectures chosen and how is the hyperparameter selection and other details impacting the experimental results (e.g., see Table F1).

---

> ### Author Response · Authors · 2023-11-17
> **Thank you for your review !  We have expanded the experimental details and included further computation time ablations.**
>
> Dear Reviewer nqRv,
>
> We thank you so much for the very positive comments on our work, we are thrilled to hear such positive and constructive feedback. We thoroughly agree that more work can be done towards improving the details in the experimental section, and thus we have uploaded some changes to the experimental section within the appendix as requested (we have highlighted your requested changes in orange).
>
> In what follows we will go over each of the points raised:
>
> > In my opinion the paper could benefit from a shorter introduction and motivation and instead an extended experimental section with method limitations, experimental details,
>
> We have moved the HJB remarks which were not central to our story to the appendix. As a result we expanded the practical motivation of the work further and added Algorithm 1 detailing CMCD which should heavily benefit more applied audiences.
>
> > computational trade-offs, etc. for more applied audiences. Since this is likely not possible due to page limits, an extended treatment in the appendix would be very welcome.
>
> We have added Table 8 which details the sampling and loss complexity of the main methods that we compare to. In particular, we have also added wallclock times for inference (sampling + ln Z estimation) comparing several of the ULA + MCD-based methods and ours. We can observe that once trained, sampling time is comparable across approaches.
>
> >  What are the training times of the method, for instance, in comparison to running SMC?
>
> It is not entirely straightforward to compare SMC to our approach taking training into account, as given our fixed training iterations we dramatically outperform SMC across challenging high dimensional targets.
>
> To carry out an insightful comparison, we chose the LGCP target, which is our most numerically intense target, and we phrase the following question:
>
> “For how long do we have to train CMCD to outperform the best-run of SMC”
>
> For this, we look at Figure 1 and see that at $K=8$, CMCD already outperforms SMC at $K=256$ with 2000 particles. So we choose these two approaches to compare to each other.  Below is a brief comparison of training-time calculation, where we have included the tuning time for SMC, which is akin to our training time (as without tuning SMC’s hyperparameters, ELBOs and ln Z estimations were much worse).
>
> | Method | Train + Sample Time (min) | ln Z                | ELBO                  |
> |--------|---------------------------|---------------------|-----------------------|
> | CMCD   | $33.12 \pm 0.12$          | $491.059 \pm 3.553$  | $469.475 \pm  0.2589$ |
> | SMC    | $62.62 \pm 0.1$           | $477.162 \pm 4.998$ | $453.395 \pm 4.43$ |
>
>
> > How were the neural networks architectures chosen and how is the hyperparameter selection and other details impacting the experimental results (e.g., see Table F1).
>
> We did not tune the architectures, we inherited them from the LDVI [1] paper, they are super minimal feedforward architectures, where the only interesting design choice is on the embedding layer used to encode the timesteps.  We have added a diagram for the architecture as well as some additional details like activations (see Figure 2 in the revised version).
>
> >  weaknesses other than that the source code for the experimental validation needs to be provided in the supplement
>
> We have uploaded a zip file to the supplement with our source code as requested. We will also open-source the code on Github at the camera-ready stage (if the paper is accepted), where it will be available publicly.
>
> We hope that with the added figures, discussions, and further ablations we have addressed the reviewer’s remarks and look forward to further feedback!!
>
> [1] Geffner, T. and Domke, J., 2023, April. Langevin Diffusion Variational Inference. In International Conference on Artificial Intelligence and Statistics (pp. 576-593). PMLR.

---

> > ### Comment · Reviewer_nqRv · 2023-11-22
> >
> > I thank the authors for clarifications and answering my questions. I will keep my current rating.

---

### Official Review · Reviewer_2i1s · 2023-11-03

**Soundness:** 3 good
**Presentation:** 3 good
**Contribution:** 3 good
**Rating:** 6
**Confidence:** 3

**Summary:**

The authors propose to revisit sampling through the lens of generative modeling. In particular, they show that the EM algorithm can be formulated as an Iterative Proportional Fitting when there is no restriction on the kernels underlying the generative models. Based on their presentation and observations, they propose a score-based annealing (Controlled Monte Carlo Diffusion) that betters the state of the art on two datasets: funnel and GMM

**Strengths:**

I find the paper very clear and the presentation very interesting. I believe the main novelty to be in the score-based annealing CMCD. I am unsure how novel the restricted correspondence between EM and IPF is; however, presenting such a relation is very interesting.

**Weaknesses:**

I am unsure how novel the restricted correspondence between EM and IPF is; in particular, it seems to me that the correspondence only holds when there is no restriction on $p^\theta(x\vert z)$; in the general setting, EM does not allow for the forward generative model $\pi^{2n+1}(x,z)$ to have as a marginal $\pi_x(x) = \mu(x)$ and the backward to have as a marginal $\pi_z(z) = \nu(z)$. If I am not mistaken, I think it would be very helpful to have some comments on this point in the paper.

**Questions:**

I would be very happy to have more information on the relation between the correspondence between EM and IPF as discussed in limitations.

---

> ### Author Response · Authors · 2023-11-11
> **Thank you for your review, some quick follow up questions, to help us update the manuscript**
>
> Dear Reiveiwer 2i1s,
>
> We would like to thank you for the very helpful comments and for taking the time to read our work with such care.
>
> Before we upload an updated version addressing your feedback, we would like to clarify some of your questions if possible in order to understand how we can improve the EM contributions and their clarity:
>
> 1. “I am unsure how novel the restricted correspondence between IPF and EM is”
>
> To the best of our knowledge, our paper is the first to make this connection. In case this is not true, we would appreciate it if you could point us to any references where this connection is already proved. Is there any other aspect of our contribution where you might be uncertain about the novelty, which we would be happy to clarify?
>
> 2. “EM does not allow for the forward generative model  $\pi^{2 n+1}(x, z)$ to have as a marginal $ \pi_x(x)=\mu(x)$ and the backward to have as a marginal $\pi_z(z)=\nu(z)$ … .”
>
> When you say the general setting, do you mean the case where $p^\theta(x|z)$ and $p^\phi(z|x)$ are restricted? You are completely right that the connection between EM and IPF only holds when these transitions are made to be flexible enough. We mention this briefly in the introduction, but will indeed emphasize this more explicitly. Indeed if the transition densities are restricted it is not possible to match the marginals via EM, however in our paper we focus on the unrestricted setting and thus our construction of EM⇔ IPF  is valid.
>
> 3.  I think it would be very helpful to have some comments on this point in the paper.
>
> We aim to address this in the main text by adding the following remark:
>
> “Note that the correspondance between the EM and IPF iterates can only be made true if the transition densities are unrestricted. This can be achieved by parametrising them as transition densities of SDEs and thus can be carried out approximately via discretising these SDEs in practice. For the setting where one may restrict them to Gaussians as done with VAEs it becomes clear that the EM iterates cannot be minimised fully to enforce the constraint required by the IPF iterates.”
>
> We hope that this remark addresses your concerns, and would be happy to add any additional details that might clarify this further. We would like to highlight that this is mostly a theoretical connection that unifies classical algorithms from VI and OT. We intended this as a bridging element that can allow readers from VI to familiarise themselves with OT through something they are already familiar with and vice-versa.
>
> 4.   they propose a score-based annealing (Controlled Monte Carlo Diffusion) that betters the state of the art on two datasets: funnel and GMM
>
> We thank the reviewer for pointing out the state-of-the-art results that our algorithm achieves, however, we would like to clarify that our method does indeed obtain state-of-the-art performance on 7 of the 8 datasets we consider, and is highly competitive on the remaining dataset (Lorenz, d=90). All of these 8 datasets are highly popular in the literature as standard benchmarks.
>
> Finally in order to further enhance the EM contributions, we have recently been able to show that our EM⇔ IPF correspondence extends beyond KL divergences and to f-divergences (via applying results from Proposition 6 in [1]). We will add a further proposition and proof to the appendix extending our EM⇔ IPF result to f-divergences, and we hope that this further novel connection strengthens the novelty of this connection for the reviewer and the community.
>
> Once again, we thank the reviewer for their insightful comments, and would be very grateful if the reviewer could clarify whether these additions address their concerns regarding the EM contribution, and any further clarifications we could make in case we don’t address their concerns.
>
> [1] Baudoin, F., 2002. Conditioned stochastic differential equations: theory, examples and application to finance. Stochastic Processes and their Applications, 100(1-2), pp.109-145.

---

> > ### Comment · Reviewer_2i1s · 2023-11-11
> > **Thank you very much for your reply**
> >
> > The authors' response addresses my question and provides additional information.

---

> ### Author Response · Authors · 2023-11-11
>
> Dear Reviewer 2i1s,
>
> We thank you for your comment and are glad to see that our response addresses your question. With this additional information and clarification that we have provided, as well as our experimental results establishing **SOTA across 7 datasets**, we humbly ask if the reviewer could consider re-evaluating their score if their major questions have been addressed. Alternatively, we are very happy to address any additional suggestions that could improve our contribution and make it a stronger candidate for acceptance with a higher score and are glad to engage with the reviewer throughout the rebuttal period!
>
> **Update:** We have updated a revised version of our manuscript, including your suggested comments and additional information. These changes have been colored in blue and can be found on pages 6 and 31.

---

> > ### Author Response · Authors · 2023-11-13
> > **Updated draft of the paper incorporating reviewer feedback, request reconsideration of paper!**
> >
> > Dear Reviewer 2i1s,
> >
> > We have uploaded a revised version of the manuscript, which incorporates feedback from the other reviewers and your clarifications (on Page 6 and Page 31), with the text highlighted in blue. We further polished and updated the introduction section along with the mathematical framework. We hope this updated version of our paper reads better to the reviewer, and we are glad that our response "addresses my question". We have incorporated significant improvements to the paper due to the reviewer's engagement, and given that we have addressed any issues from the reviewer's side, we would greatly appreciate any reconsideration regarding our paper's strengths and suitability for acceptance as a meaningful contribution to ICLR.

---

> > > ### Author Response · Authors · 2023-11-21
> > > **Update on revised manuscript, feedback request**
> > >
> > > Dear reviewer 2i1s,
> > >
> > > We hope that our additional comments and revised manuscript address any concerns you may have remaining, and would love to engage further during the rebuttal period to address any pending concerns you may have.
> > >
> > > As before we thank you for your review and insightful comments.
> > >
> > > Best regards,
> > > Authors

---

### Official Review · Reviewer_LiYc · 2023-11-05

**Soundness:** 4 excellent
**Presentation:** 4 excellent
**Contribution:** 3 good
**Rating:** 8
**Confidence:** 4

**Summary:**

This paper is divided in two parts. In the first part, the authors provide a unifying framework for VAEs and diffusion models using stochastic differential equations. In the second part connect the iterative proportional fitting procedure with the expectation maximization algorithm and provide a new objective for solving IPF that avoids the mode forgetting phenomenon that happens when one implements the diffusion schrodinger bridge. The added regularization term is null iff the HJB equation holds. Finally, the authors introduce the MCMD algorithm which makes uses of forward and backward SDEs that admit as marginals a prescribed sequence of distributions $(\pi_t)_t$.

**Strengths:**

- I enjoyed reading this paper; it is highly pedagogical and sheds light on interesting connections between variational inference, diffusion models, stochastic control as well as optimal transport. The connection with the expectation maximization algorithm is particularly interesting.
- The proposed algorithm CMCD generalizes existing methods such as Monte Carlo VAE aswell as MCD. I believe that the adopted point of view here is more rigourous and elucidates what the backward process should "look like", which was not clear in the Monte Carlo VAE paper for example.
- Extensive experiments show that the CMCD method outperforms existing methods in relatively high dimensional examples.

**Weaknesses:**

- It is quite unfortunate that there are no experiments for the loss proposed in Proposition 3.2 to back the fact that the proposed loss does not suffer from mode forgetting. Furthermore, it is not clear how this is implemented in practice since a Laplacian term is involved.

**Questions:**

i have no further questions

---

> ### Author Response · Authors · 2023-11-16
> **Additional Mode collapse experiments + pointer to reference forgetting experiments in Appendix**
>
> Dear Reviewer LiYc,
>
> Thank you for your review and very insightful comments which have led to major improvements in our revised version.
>
> > It is quite unfortunate that there are no experiments for the loss proposed in Proposition 3.2 to back the fact that the proposed loss does not suffer from mode forgetting.
>
> Out of the alternate losses we discuss, it is the log-variance divergence (introduced after Equation 24)  that can handle mode collapse:
>
> $$ \mathcal{L}^{\mathrm{Var}}(\phi)=\mathrm{Var}\left(\ln \frac{\pi_T\left(\boldsymbol{Y}_T\right)}{\pi_0\left(\boldsymbol{Y}_0\right)}+\int_0^T \Delta \phi_t\left(\boldsymbol{Y}_t\right) \mathrm{d} t-\sigma \sqrt{2} \int_0^T \nabla \ln \pi_t\left(\boldsymbol{Y}_t\right) \circ \mathrm{d} \boldsymbol{W}_t-\sigma^2 \int_0^T\left|\nabla \ln \pi_t\left(\boldsymbol{Y}_t\right)\right|^2 \mathrm{~d} t\right)$$
>
> The loss with the HJB/PINN regularizer in Proposition 3.2 does not address the issue of mode collapse, and its application is more relevant to different problems (relevant to molecular dynamics).
>
> > Furthermore, it is not clear how this is implemented in practice since a Laplacian term is involved.
>
> As you have pointed out, there are some Laplacian terms that are not immediately obvious how we could compute. However, as it turns out the term inside the variance is simply the RND between the forward and backwards SDEs, so we can use the exact same discretisation that we used for KL to discretise it and thus:
>
> $$ \mathcal{L}^{\mathrm{Var}}(\phi)  \approx \mathrm{Var}\left[\ln\frac{\pi_0(Y_0)}{\hat{\pi} (Y_T)}\prod_{k=0}^{K-1}\frac{\mathcal{N}(Y_{t_{k+1}} |Y_{t_k}  + (\nabla \ln \pi_{t_{k}}+\nabla \ln \phi_{t_k})(Y_{t_k}) \Delta t_k , 2 \sigma^2 \Delta t_k)}{\mathcal{N}( Y_{t_k} |Y_{t_{k+1}}   + (\nabla \ln \pi_{t_{k+1}}-\nabla \ln \phi_{t_{k+1}})(Y_{t_{k+1}}) \Delta t_k , 2 \sigma^2 \Delta t_k)}\right]$$
>
>
> This makes this loss as scalable as the KL divergence CMCD objective that we ablate. In particular, it is a bit more scalable as the Variance can be taken w.r.t to any measure, thus allowing us to detach the samples and circumvent backpropagating through timesteps.
>
> >  .. There are no experiments for the . ..
>
> To address your main weakness, we explore the log-variance (logvar) loss in the very challenging 40 GMM target [2] for which reverse KL-based methods all mode collapse.
> We compare CMCD KL to CMCD with logvar, and we demonstrate how it significantly outperforms the KL-based loss, both in ELBO but also qualitatively in its ability to overcome mode collapse. The results and plots illustrating mode collapse can be found in **Appendix F.3. (highlighted in olive)** in particular **Tabe 2, Figures 3 and 4**, and here is a table to summarise them
>
> |                     |         ELBO        |        $\ln Z$       |   $\mathcal{W}_2$  |
> |---------------------|:-------------------:|:--------------------:|:------------------:|
> | *log-variance* loss |  -1.279 $\pm$ 0.096 | -0.065 $\pm$ $0.101$ | 0.0143 $\pm$ 0.001 |
> | KL loss             | -2.286 $\pm$ 0.1109 |  -0.244 $\pm$ 0.3309 | 0.0441 $\pm$ 0.012 |
>
>
> > …  for the loss proposed in Proposition 3.2 to back the fact that the proposed loss does not suffer from mode forgetting ..
>
> The loss in proposition 3.2 (now moved to the appendix) solves a different problem to mode collapse. Rather than addressing mode forgetting, it deals with forgetting the reference SDE. This does not solve practical issues (e.g. such as mode collapse) in sampling or generative modelling, but it might be useful in transition path problems in MD. **In appendix G**, we do some small ablations and demonstrate across a series of toy targets (including an MD double well problem) that it does indeed do well in not forgetting the reference process.  To address the Laplacian term computation in this loss in this case, we used specialised architectures from OT [1], which allow for fast calculation of these terms.
>
> Finally, we note we have added a series of further experiments and ablations to address the other reviewers as well as improved the presentation of our work.
>
> We hope that this addresses your main weakness, and we sincerely request any further feedback that you may be able to provide us with regarding our comprehensive changes to the presentation of the paper, as well as additional ablations and baselines. Thank you for your continued feedback!
>
> [1] Onken, D., Fung, S.W., Li, X. and Ruthotto, L., 2021, May. Ot-flow: Fast and accurate continuous normalizing flows via optimal transport. In Proceedings of AAAI
>
> [2] Midgley, L.I., Stimper, V., Simm, G.N., Schölkopf, B. and Hernández-Lobato, J.M., 2022. Flow annealed importance sampling bootstrap. ICLR 2023

---

> > ### Author Response · Authors · 2023-11-21
> > **Feedback on further mode collpase ablations + further experiments and improved presentation**
> >
> > Dear reviewer LiYc,
> >
> > We thank you once again for your insightful comments regarding the loss function that can greatly improve mode-collapse in our sampling algorithm. As you can see in our above response, we implement our CMCD sampler under the log-variance divergence loss, and **find it greatly improves the mode coverage** compared to the default (KL-based) loss function.
> >
> > Our additional experiments on the challenging 40GMM task also demonstrate this visually (see **Figure 3**) as well as quantitively via elbo/lnZ/sample quality metrics (**see Table 2**).
> >
> >  We thank the reviewer for their comment, which has directly led to the addition of an additional experimental section, and we hope that this may have addressed any concerns they may have regarding the paper?
> >
> >  We would be really grateful to get any further feedback the reviewer may have about these additional experiments, as well as the revised version of our manuscript, which adds many new experiments,  a completely revisited introduction section, and the overall improved presentation flow across the paper.

---

> > > ### Comment · Reviewer_LiYc · 2023-11-22
> > >
> > > Dear authors,
> > >
> > > Thank you for your response and your hard work. In my initial comment, I meant reference SDE and not mode forgetting as you pointed out, sorry for that.
> > >
> > > I am satisfied by the numerical experiments and the paper overall. I have chosen to increase my score.

---

### Official Review · Reviewer_yH6G · 2023-11-06

**Soundness:** 4 excellent
**Presentation:** 1 poor
**Contribution:** 2 fair
**Rating:** 8
**Confidence:** 3

**Summary:**

This paper presents the controlled Monte Carlo diffusion sampler (CMCD). CMCD builds upon previous work in the literature by adapting both the forward and the backward dynamics. The authors also posit a regularised iterative proportional fitting for schrodinger bridges that improve upon standard IPF. Finally, the authors link CMCD to Jarzinsky identity and demonstrate its performance through numerical experiments.

**Strengths:**

The technical contributions of this paper is unquestionable. The authors have managed to link key and fundamental ideas from various fields. I especially liked the connection between regularised IPF and EM algorithm, which is illuminating. I think the paper has good contribution and would be helpful to researchers in the field if presented correctly.

**Weaknesses:**

Unfortunately, I think the presentation of the paper needs to be revised extensively. The paper is confusing, and posits ideas without forewarning or motivation. The introduction is especially difficult to read, and not because of its technicality. In general, I believe the authors' math is quite readable. It is the writing and motivation that needs improvements. As an example, nowhere in the introduction the authors explicitly state the problem they are tackling (CMCD). It takes until page 7, eq. (22) for the authors to state the explicit problem that they address.

As another example, the authors never explicitly state the necessity of various connections they draw. For example, the text above Proposition 3.1 explains why IPF fails to perform well. However, there is no explanation afterwards to say why going through the lens of EM solves this issue. Rather, the text proceeds to describe optimality conditions in what seemed to me to be like a tangent. In summary, although I could follow the math, it was extremely hard for me to follow the context of the paper.

It pains me to say that I cannot suggest the authors any single thing that could drastically improve the paper. I appreciate the technical contribution. However, given how difficult it is to follow its context, I don't believe that the paper would be helpful to the wider community in its current state.

At the very least, I think the first two pages of the introduction should be thoroughly revised to focus more on the problem addressed in the paper, rather than overloading on related technical terms and connections. I also think the propositions which are not key contributions of the paper (for instance, 2.1) could be better placed in the supplements.

I also list some minor typos:

1. A bracket has not been closed in page 2, paragraph 3, after "(Section 2.2"
2. Page 7, last paragraph, "Proposition2.2" is missing a white-space.

Post Rebuttal: I thank the authors for the extensive revision. I think it addresses most of my comments and the authors truly went above and beyond. I have revised my scores to reflect this. I think that in its current form the paper can definitely be useful and has interesting insights which are presented clearly. I believe, if accepted this will be an important contribution to the community.

**Questions:**

Is eq. (10) the same as eq (1)? I failed to find the difference.

---

> ### Author Response · Authors · 2023-11-12
> **Revised version of manuscript with significant changes to the presentation structure  - Part I**
>
> Dear Reviewer yH6G,
>
> We thank you for the time and the very detailed feedback, especially since the presentation difficulties you have encountered must have required more time in parsing through our manuscript, and we apologise for the same.
>
> We would also like to thank you for the very positive comments regarding the strength of the technical contribution of our work. We really appreciate your feedback and suggestions for improvement despite the presentation of the paper, while still being able to provide excellent actionable feedback on our technical contribution.
>
> We would like to work with you as much as possible through this discussion period to understand how we can further improve the presentation. We appreciate that you must have many other commitments but we would greatly appreciate it if you could engage with us in improving the presentation of our work.
>
> We have uploaded a revised version of the manuscript where we have attempted to address some of the changes recommended by the reviewer, these changes have been coloured in **magenta/pink**.
>
> The revised manuscript (https://openreview.net/pdf?id=PP1rudnxiW) contains the following changes which are highlighted in the revised version.
>
> 1. We have mentioned that our focus is the sampling task from a very early stage in the introduction and we formally introduce the sampling task in a titled paragraph, connecting it to framework 1 and thus providing motivation to our setup.
> 2. We have reworded the abstract to acknowledge the focus on the sampling task and we have removed the remark on the HJB regularisers from the abstract as this is not our central contribution.
> 4. We have also highlighted that the EM ⇔ IPF correspondence mostly serves a conceptual purpose that connects two adjacent fields rather than an algorithmic one.
> 5. We have changed our contributions to shift the focus to CMCD and the sampling task.
> 6. On page 6, we removed non-central results such as the paragraph on HJB regularizers and the corollary that followed. As before, we highlight that the EM and IPF connection is conceptual rather than practical.
> 7. Also on page 6 we explain more clearly the issues present with EM and IPF, emphasising that an accumulation of errors occurs due to their sequential nature. We then signpost to the next section motivating that our proposed method CMCD can be trained end to end with simultaneous updates and thus solve this sequential error accumulation issue.
> 8. We have also added pseudocode for sampling and $\ln Z$ estimation with CMCD and we hope that this also draws the focus/attention to the task and enhances the readability.
>
> Additionally, we have also tried restructuring the introduction in a different way which can be found in this anonymised link https://anonymous.4open.science/r/CMCD-6BF6/rebuttal_version.pdf
> (same link as the anonymised codebase), here we changed the way the framework is introduced to try and make it more amenable to sampling. We hope that one of these revisions addresses some of the presentation issues raised by the reviewer.
>
> We hope that the revised presentation and rebuttal pave the way to addressing the reviewer's concerns and we are eager to hear the reviewer's feedback plus further points of improvement.

---

> ### Author Response · Authors · 2023-11-12
> **Revised version of manuscript with significant changes to the presentation structure - Part II**
>
> In what follows we will directly address some of your other questions/comments (which have been updated in the revised version of the manuscript):
>
> > Is eq. (10) the same as eq (1)? I failed to find the difference.
>
> The following equations are taken from the unrevised version of the manuscript that you read.
>
> Equation (1):
>
> $$\boldsymbol{z} \sim \nu(\boldsymbol{z}), \quad \boldsymbol{x} \mid \boldsymbol{z} \sim p^\theta(\boldsymbol{x} \mid \boldsymbol{z})$$
>
> Equation (10):
>
> $$\mathcal{L}^{ext}(\phi, \theta) :=D(q^{\mu, \phi} ( y_{0: L} ) \vert\vert p^{\nu, \theta} ( y_{0: L}))$$
>
> Equation (1) specifies a “static” generative model whilst equation (10) details the divergence between two pathwise (many latent) generative models. Is it possible the reviewer was referring to a different pair of equations? Maybe they meant equations 10 and 5:
>
> Equation (5)
>
> $$ \mathcal{L}_D(\phi, \theta):=D\left(q^\phi(\boldsymbol{z} \mid \boldsymbol{x}) \mu(\boldsymbol{x}) \vert\vert p^\theta(\boldsymbol{x} \mid \boldsymbol{z}) \nu(\boldsymbol{z})\right)$$
>
> Both Equations 10 and 5 define the loss as a divergence between backwards and forwards generative models. However, Equation 10 (as the text preceding it details) is a path KL divergence because the generative model has many intermediate latent variables (which becomes an SDE in the limit) whilst Equation 5 does not have any of these intermediate latents. It is a static/non-dynamic setup (you can think of the intermediaries as being marginalised out).
>
> The purpose of having both these equations is, as reviewer LiYc highlights (i.e. “I enjoyed reading this paper; it is highly pedagogical and sheds light … ”), pedagogical. We present our work and framework in a pedagogical manner, first starting from simple static generative models that readers may be familiar with from VAE literature, and then we slowly build up from there to the non-static / dynamic SDE setting.
>
> > Proposition 3.1 explains why IPF fails to perform well. there is no explanation afterwards to say why going through the lens of EM solves this issue.
>
> Indeed this was presented in a very confusing manner and has been changed in the revised version.
>
> The purpose of the connection to EM was not to solve an issue with IPF but to highlight how there exists a correspondence between these two seminal algorithms that have been developed almost in isolation from one another. This provides us a conceptual link between VI and OT which can help when learning one field given knowledge of the other.
>
> Thank you for bringing this point to our attention.
>
> > . Rather, the text proceeds to describe optimality conditions in what seemed to me to be like a tangent.
>
> This is a really good point we were unable to explain this fully due to the page constraint, we have moved this section to the appendix and taken more time to explain the issues with VM and IPF.
>
> > It pains me to say that I cannot suggest the authors any single thing that could drastically improve the paper. I appreciate the technical contribution.
>
> We are very sorry that the initial presentation of our work has made you feel this way, however, we still believe your comments have been very helpful and have led us to make significant restructuring changes across the paper.
>
> > At the very least, I think the first two pages of the introduction should be thoroughly revised to focus more on the problem addressed in the paper
>
> We have added the sampling task (which is the task CMCD solves) to the start of the introduction and modified the signposting and motivational elements in the introduction to refer more to this task.
>
> > I also think the propositions which are not key contributions of the pape … could be better placed in the supplements.
>
> We have moved proposition 3.2 to the appendix as well as the section on HJB reguralisers which are not part of our central algorithmic contribution/experiments. We hope this allows the reader to have a more clear and contextual focus on the story.
>
> > Rather than overloading on related technical terms and connections.
>
> We have not fully removed the connections / related work from the introduction as you requested for the following reasons
>
> 1. Variational inference is effectively the flavour of the task we are solving. We are doing sampling through VI using ideas from optimal transport. This requires us to introduce and discuss VI (Framework 1) in the introduction.
> 2. ELBOs are used for solving the second part of the sampling problem, namely estimating the normalising constant, and thus it is essential for us to introduce these earlier on.
> 3. OT and coupling allow us to highlight some of the pitfalls that doing full-blown VI can have, and as part of the motivation we want to discuss these issues in the introduction since they motivate our proposed improvements.
>
> Instead, we have tried to reinforce their relevance to the sampling problem.

---

> > ### Comment · Reviewer_yH6G · 2023-11-14
> >
> > I sincerely thank the authors for their comments and their extensive modifications. I think the presentation is much clearer now and I also now finally understand the greater context. I believe that both of the revised manuscripts are good, although the second one (4openscience) is slightly clearer.
> >
> > I believe with this modification the paper has reached finally a place where it is broadly significant and I have modified my original rating to reflect this.

---

### Author Response · Authors · 2023-11-22
**General rebuttal + revised version**

Dear Reviewers,


We would like to thank all reviewers for taking the time to review this manuscript and for providing insightful feedback and discussion, which has significantly strengthened the manuscript overall. We have now uploaded the final revised version of the manuscript, which we hope reflects most of the advice provided by the reviewers.

We are delighted to hear that the reviewers found our contributions **“an important contribution to the community.”** (reviewer **yH6G**), **“highly original and will be greatly valued by the community.”** (reviewer **nqRv**),  **“is highly pedagogical and sheds light on interesting connections ..”** (reviewer **LiYc**). Furthermore, we are delighted to hear that reviewer yH6G finds our revised presentation **“much clearer now”** and are thankful for the time and effort put in by the reviewer in helping us revise this manuscript.

In what follows, we will briefly outline the changes in our manuscripts and the remarks they address. For more details, please see the individual responses posted to each reviewer last week.


1. To address reviewers ndxa and yH6G’ concerns on the presentation, we have significantly revised the introduction in our manuscript as well as added structural changes to the flow and story by moving non-central results to the appendix and taking more time to motivate/explain our method, including the addition of pseudocode in Algorithm 1. Reviewer yH6G has already promptly acknowledged this update and has positively updated their review to reflect this.  We have also added Algorithm 2 detailing our numerical training procedure.
2. We have also provided two different restructures of the introduction to assess across reviewers, which one seems more clear. So far, we have had some feedback from reviewer yH6G.
3. We have added 3 additional baseline methods (NF-VI, AFT, DNF) on top of our 7 pre-existing baselines to address reviewer ndxa’s request. We also note that we already have the underdampened langevin dynamics under the acronym UHA. These baselines can be found in the Appendix. Overall, we continue to outperform across the majority of the targets and remain competitive in the ones that we don't.
4. To address the concerns cited by reviewer LiYc, we have added a mode collapse experiment showcasing how the log variance divergence we discussed addresses mode collapse in highly challenging situations where the reverse KL-based loss fails.
5. To address reviewer yH6G, we have added sampling + log Z computation times across methods showcasing how our approach does not significantly slow down compute.
6. We have added more thorough details to the experimental section in the appendix, including diagrams for our architectures, further details on activations, and computational complexities for loss computation and sampling for our proposed approaches.
7. In response to reviewer nqRv, we have also added a further ablation comparing our total training/tuning/inference time to the total training/tuning/inference time for SMC.  These results showcase how our approach can reach much lower values for ELBO/lnZ than SMC within the same compute time.

We would like to take the time to thank reviewers **yH6G**, and **ndxa** for their continued input and engagement with us, especially in helping us improve the overall presentation of this manuscript.

We hope that these significant additions and changes address most of the concerns raised by the reviewers. Friendly reminder that the discussion period ends today, we would be very grateful for additional feedback (given the significant additional work).

---

### Meta-Review · Area_Chair_BdRq · 2023-12-05

**Metareview:**

This paper draws connections between regularised IPF and EM algorithm, as well as forward-backward diffusion process and the celebrated Jarzinsky's equality in statistical mechanics. Draw upon the connections, it proposes a "Controlled Monte Carlo Diffusions" approach for Bayesian inference tasks. The authors addressed the reviewers' comments well and there is no remaining salient concerns for the paper to be accepted.

**Justification For Why Not Higher Score:**

The manuscript can benefit from a more careful comparison against the state of the art methods. Currently we can observe that the number of discretization steps required is small. But compared to MCMC methods, an additional inner loop optimization over $\phi$ is required to implement the method. It is not clear if the proposed method outperforms existing methods in term of the total computation budget.

**Justification For Why Not Lower Score:**

The authors addressed the reviewers' comments well and there is no remaining salient concerns for the paper to be accepted.

---

### Decision · Program_Chairs · 2024-01-16

Accept (poster)